# Respiratory viral infections awaken metastatic breast cancer cells in lungs

Shi B. Chia[1,24], Bryan J. Johnson[1,24], Junxiao Hu[2,3], Felipe Valença-Pereira[4], Marc Chadeau-Hyam[5,6,7], Fernando Guntoro[6,7,8], Hugh Montgomery[9], Meher P. Boorgula[3,10], Varsha Sreekanth[3,11], Andrew Goodspeed[3,10], Bennett Davenport[4], Marco De Dominici[1], Vadym Zaberezhnyy[1], Wolfgang E. Schleicher[1], Dexiang Gao[2,3], Andreia N. Cadar[12,13], Lucia Petriz-Otaño[14], Michael Papanicolaou[14], Afshin Beheshti[15,16,17], Stephen B. Baylin[15,18,19], Joseph W. Guarnieri[20,23], Douglas C. Wallace[20,21], James C. Costello[3,11], Jenna M. Bartley[12,13], Thomas E. Morrison[4], Roel Vermeulen[5,6,22,25], Julio A. Aguirre-Ghiso[14,25], Mercedes Rincon[3,4,25] & James DeGregori[1,2,3,4,25] ✉

Breast cancer is the second most common cancer globally, with most deaths caused by metastatic disease, often following long periods of clinical dormancy[1]. Understanding the mechanisms that disrupt the quiescence of dormant disseminated cancer cells (DCCs) is crucial for addressing metastatic progression. Infections caused by respiratory viruses such as influenza and SARS-CoV-2 trigger both local and systemic inflammation[2,3]. Here we demonstrate, in mice, that influenza and SARS-CoV-2 infections lead to loss of the pro-dormancy phenotype in breast DCCs in the lung, causing DCC proliferation within days of infection and a massive expansion of carcinoma cells into metastatic lesions within two weeks. These phenotypic transitions and expansions are interleukin-6 dependent. We show that DCCs impair lung T cell activation and that CD4+ T cells sustain the pulmonary metastatic burden after the influenza infection by inhibiting CD8+ T cell activation and cytotoxicity. Crucially, these experimental findings align with human observational data. Analyses of cancer survivors from the UK Biobank (all cancers) and Flatiron Health (breast cancer) databases reveal that SARS-CoV-2 infection substantially increases the risk of cancer-related mortality and lung metastasis compared with uninfected cancer survivors. These discoveries underscore the huge impact of respiratory viral infections on metastatic cancer resurgence, offering new insights into the connection between infectious diseases and cancer metastasis.

Breast cancer is the most diagnosed cancer in women and is the second most common cause of cancer-associated deaths in the United States[1]. After initial remission, DCCs can stay dormant for years to decades before metastatic relapse, most commonly in lung, bone and liver[4]. Both cell-intrinsic factors and the tumour microenvironment determine whether metastatic cells stay dormant or progress[5]. Importantly, microenvironmental perturbations, such as those caused by increased inflammation, can be sufficient to increase metastasis[5].

Viral respiratory infections are common. Seasonal influenza affects more than 1 billion people each year[6] and, by May 2025, SARS-CoV-2 infection had caused nearly 778 million recorded cases of COVID-19 (ref. 7). Viral respiratory infections are typically associated with pulmonary inflammation, with a concomitant increase in pulmonary inflammatory

[1]Department of Biochemistry and Molecular Genetics, University of Colorado Anschutz Medical Campus, Aurora, CO, USA. [2]Department of Pediatrics, University of Colorado Anschutz Medical Campus, Aurora, CO, USA. [3]University of Colorado Comprehensive Cancer Center, University of Colorado Anschutz Medical Campus, Aurora, CO, USA. [4]Department of Immunology and Microbiology, University of Colorado Anschutz Medical Campus, Aurora, CO, USA. [5]Division of Environmental Epidemiology, Institute for Risk Assessment Sciences, Utrecht University, Utrecht, The Netherlands. [6]MRC Centre for Environment and Health, Imperial College London, London, United Kingdom. [7]Department of Epidemiology and Biostatistics, School of Public Health, Imperial College London, London, United Kingdom. [8]MRC Centre for Global Infectious Disease Analysis, School of Public Health, Imperial College London, London, United Kingdom. [9]Department of Medicine, University College London, London, United Kingdom. [10]Department of Biomedical Informatics, University of Colorado Anschutz Medical Campus, Aurora, CO, USA. [11]Department of Pharmacology, University of Colorado Anschutz Medical Campus, Aurora, CO, USA. [12]UConn Center On Aging, University of Connecticut School of Medicine and UConn Health, Farmington, CT, USA. [13]Department of Immunology, University of Connecticut School of Medicine and UConn Health, Farmington, CT, USA. [14]Department of Cell Biology, Cancer Dormancy Institute, Montefiore Einstein Comprehensive Cancer Center, Albert Einstein College of Medicine, Bronx, NY, USA. [15]COVID-19 International Research Team, Medford, MA, USA. [16]Department of Surgery, McGowan Institute for Regenerative Medicine – Center for Space Biomedicine, University of Pittsburgh, Pittsburgh, PA, USA. [17]Stanley Center for Psychiatric Research, Broad Institute of MIT and Harvard, Cambridge, MA, USA. [18]Oncology and Medicine Departments, The Sidney Kimmel Comprehensive Cancer Center, Johns Hopkins University School of Medicine, Baltimore, MD, USA. [19]Van Andel Research Institute, Grand Rapids, MI, USA. [20]Center for Mitochondrial and Epigenomic Medicine, Division of Human Genetics, The Children's Hospital of Philadelphia, Philadelphia, PA, USA. [21]Department of Pediatrics, Division of Human Genetics, Perelman School of Medicine, University of Pennsylvania, Philadelphia, PA, USA. [22]Julius Centre for Health Sciences and Primary Care, University Medical Centre, Utrecht University, Utrecht, The Netherlands. [23]Present address: Blue Marble Space Institute of Science, Seattle, WA, USA. [24]These authors contributed equally: Shi B. Chia, Bryan J. Johnson. [25]These authors jointly supervised this work: Roel Vermeulen, Julio A. Aguirre-Ghiso, Mercedes Rincon, James DeGregori. ✉e-mail: james.degregori@cuanschutz.edu

cytokines, such as interleukin-6 (IL-6) and interferons (IFNs), and an expansion of immune cells, including neutrophils, macrophages and T lymphocytes[2,3]. Such inflammatory mechanisms, specifically involving IL-6 and STAT3 signalling[8,9], neutrophils and neutrophil extracellular traps[10], as well as the CD4+ cell–macrophage axis[11], have been identified as regulators of metastatic processes in cancer.

The observation that death rates from cancer rose in the first two years of the COVID-19 pandemic[12], which is not fully accounted for by COVID-19 deaths or delayed screening and treatment, prompts an important hypothesis: that pulmonary viral infections increase cancer deaths by triggering the development of metastases from dormant DCCs. We sought to test this hypothesis through a dual approach: examining the effects of viral respiratory infections (influenza virus and SARS-CoV-2) on breast cancer dormancy in mouse models and correlating SARS-CoV-2 infection among cancer survivors to metastatic progression and cancer mortality.

## Influenza virus infection awakens DCCs

To study the effects of influenza virus infection on the awakening of dormant breast DCCs already lodged in the lung, we used the well-established MMTV-ErbB2/Neu/Her2 (hereafter MMTV-Her2) mouse model of breast cancer metastatic dormancy, in which mice overexpress rat *Neu* (*Erbb2*, a paralogue of human *HER2*) in epithelial mammary gland cells[13,14]. HER2+ early lesion cells in the mammary glands seed the lungs and other organs with DCCs within 10–14 weeks of life, where they remain largely as dormant single cells for up to one year before progressing to overt metastatic disease[15]. Thus, this model recapitulates the persistence of dormant DCCs in lungs and bone marrow in individuals who remain in remission for years to decades.

MMTV-Her2 mice (FVB background) were infected with a sublethal dose of the influenza A virus (IAV) (Fig. 1a). Infected mice lost weight and recovered by 11–12 days post-infection (dpi) (Extended Data Fig. 1a), and wild-type and MMTV-Her2 mice showed a similar inflammatory response with increased cellularity of bronchoalveolar lavage (Extended Data Fig. 1b). The kinetics of viral clearance were similar between wild-type and MMTV-Her2 mice, in which IAV RNA copies peaked around 6 dpi, with a 100–1,000-fold reduction in viral load from 9–15 dpi (Extended Data Fig. 1c).

The lungs of MMTV-Her2 mice were taken at 3, 6, 9, 15, 28 and 60 dpi (Fig. 1a) and examined for the abundance of HER2+ cells (Fig. 1b,c), as reported previously[14]. Consistent with previous work[14,16], we observed a small number of isolated DCCs or small clusters (fewer than 10 cells) in lungs before IAV infection. Strikingly, metastatic burden increased 100–1,000-fold between 3 and 15 dpi; the number of pulmonary HER2+ cells remained elevated even at 28 days, 60 days and 9 months after infection (Fig. 1b–d). IAV-mediated expansion of HER2+ DCCs was similarly observed in lungs of MMTV-Her2 mice in the C57BL/6J background at 15 dpi (Fig. 1e). Notably, the resultant expanded HER2+ cells exhibited a diffuse non-epithelial-like architecture, unlike the epithelial-like clusters and metastasis (more than 100 cells per cluster) of DCCs observed in lungs of MMTV-Her2 mice that are more than 10 months old[14,16] (Extended Data Fig. 1d). Notably, we did not observe changes in the number of Ki67+HER2+ cells in mammary glands (Extended Data Fig. 1e,f). Furthermore, quantitative PCR (qPCR) analysis of transgenic rat *Erbb2* expression in haematopoietic lineage-depleted cells from peripheral blood showed no changes with infection (Extended Data Fig. 1g). Thus, the increased number of HER2+ cells in the lungs does not seem to derive from increased seeding of cancer cells from mammary glands.

We tested MMTV-PyMT mice with mammary-specific expression of polyoma middle T-antigen oncoprotein, which display early dissemination but shorter lung DCC dormancy[16]. MMTV-PyMT mice demonstrated an increased number of small tumour clusters in lungs after IAV infection (Fig. 1f). We also tested the effect of IAV infection using EO771 breast cancer cells (C57BL/6) implanted in the mammary gland,

which seed the lungs and undergo a dormancy phase[17]. C57BL/6 mice with orthotopically implanted EO771 cells infected with IAV as above exhibited increased lung metastatic burden by 17–18 dpi compared with non-infected mice, in which EO771 cells remained largely dormant (Fig. 1g and Extended Data Fig. 1h,i) Taken together, these findings show that IAV infection promotes DCC expansion in multiple models of breast DCC dormancy.

## IAV infection induces DCC phenotypic transitions

When we examined the proliferation of DCCs in the lungs, we found a significant increase in the percentage of HER2+ cells expressing Ki67 (a marker of all cycle phases except G0) beginning at 3 dpi and peaking at 9 dpi (Fig. 2a,b). Similar results were obtained through in vivo incorporation of the thymidine analogue EdU (Fig. 2b). Although the fraction of HER2+ cells that express Ki67 decreased by 15 dpi, the total number of HER2+ cells expressing Ki67 remained highly elevated relative to baseline even at 60 dpi, given the overall increase in DCC burden in the lungs (Fig. 2b). These results show that IAV infection triggers DCC awakening in the lungs, increasing the metastatic burden.

Dormant DCCs in HER2+ and PyMT models maintain a ZFP281-driven mesenchymal-like state (vimentin+) until ZFP281 loss triggers an epithelial shift (EpCAM/E-cadherin+) during dormancy exit[16]. Consistent with previous results, most dormant DCCs present in uninfected lungs expressed vimentin and not EpCAM (Fig. 2c,d). The percentage of HER2+ cells expressing vimentin was not significantly affected early after infection (3–6 dpi). However, at 9 dpi the percentage of HER2+ cells expressing vimentin was decreased to around 50%, with a further decrease to less than 20% at 28 dpi (Fig. 2c). By contrast, early during IAV infection (3 dpi), a substantial fraction of HER2+ cells acquired EpCAM expression, associated with the awakening of DCCs (Fig. 2d). Most HER2+ cells were EpCAM-negative after 6 dpi, although the percentage of EpCAM+HER2+ cells remained elevated (Fig. 2d). Thus, IAV infection drives sustained mesenchymal marker loss and a transient epithelial shift, with a persistent mixed or hybrid population over time, creating a hybrid phenotype that enables the awakening of dormant cells. To better understand the effect of IAV infection on DCCs, we performed flow cytometry sorting for HER2+ cells from the lungs of uninfected and IAV-infected mice at 9 dpi and performed bulk RNA-seq. As expected, IAV infection induced inflammatory, IFNα, IFNγ, TNF and IL-6–JAK–STAT3 signalling pathways (Fig. 2e and Extended Data Fig. 2a–c). Notably, IAV infection also activated pathways in DCCs including collagen-containing extracellular matrix and angiogenesis (Fig. 2e), with increased expressions of many collagens and collagen-crosslinking genes (*Lox*, *Loxl1* and *Loxl2*) (Fig. 2f), metalloproteinases (*Mmp8*, *Mmp11*, *Mmp14*, *Mmp15* and *Mmp19*) (Fig. 2g) and genes implicated in angiogenesis (*Vegf-a*, *Vegf-c*, *Vegf-d*, *Vcam1*, *Icam1* and *Icam2*) (Fig. 2h). Collagen-1 abundance and crosslinking into fibrillar collagen has been linked to dormant DCC awakening[18] that could be sustained by an angiogenic switch to maintain tumour growth. Indeed, angiogenesis and metalloproteinases have been shown to have a role in sustaining dormant cancer-cell awakening[19,20]. Notably, we observed striking changes in the expression of genes involved in mesenchymal or epithelial fate (Extended Data Fig. 2d). Although previous studies have shown conversion of DCCs from a dormant mesenchymal state to a more epithelial state after awakening[16], we observed a unique and, to our knowledge, previously unrecognized hybrid and proliferative pattern after influenza virus infection, with increased expression of both mesenchymal markers (such as *Cdh2*, *Cdh11*, *Fn1*, *Eng* and *Vim*) and epithelial markers (such as *Cdh1*, *Cldn2*, *Cldn5*, *Krt19*, *Klf4* and *Ovol2*) (all $P_{adj} \leq 0.05$; Supplementary Data 1). Notably, expression of *Zfp281*, which is a key mediator of the dormant mesenchymal state[16], actually increased by 9 dpi in DCCs, further highlighting the hybrid state adopted by DCCs after IAV infection that seems to bypass the pro-dormancy function of ZFP281. We also observed increases in *Cd274* (which encodes PDL1) and decreases in *B2m* (which is required for antigen presentation by major

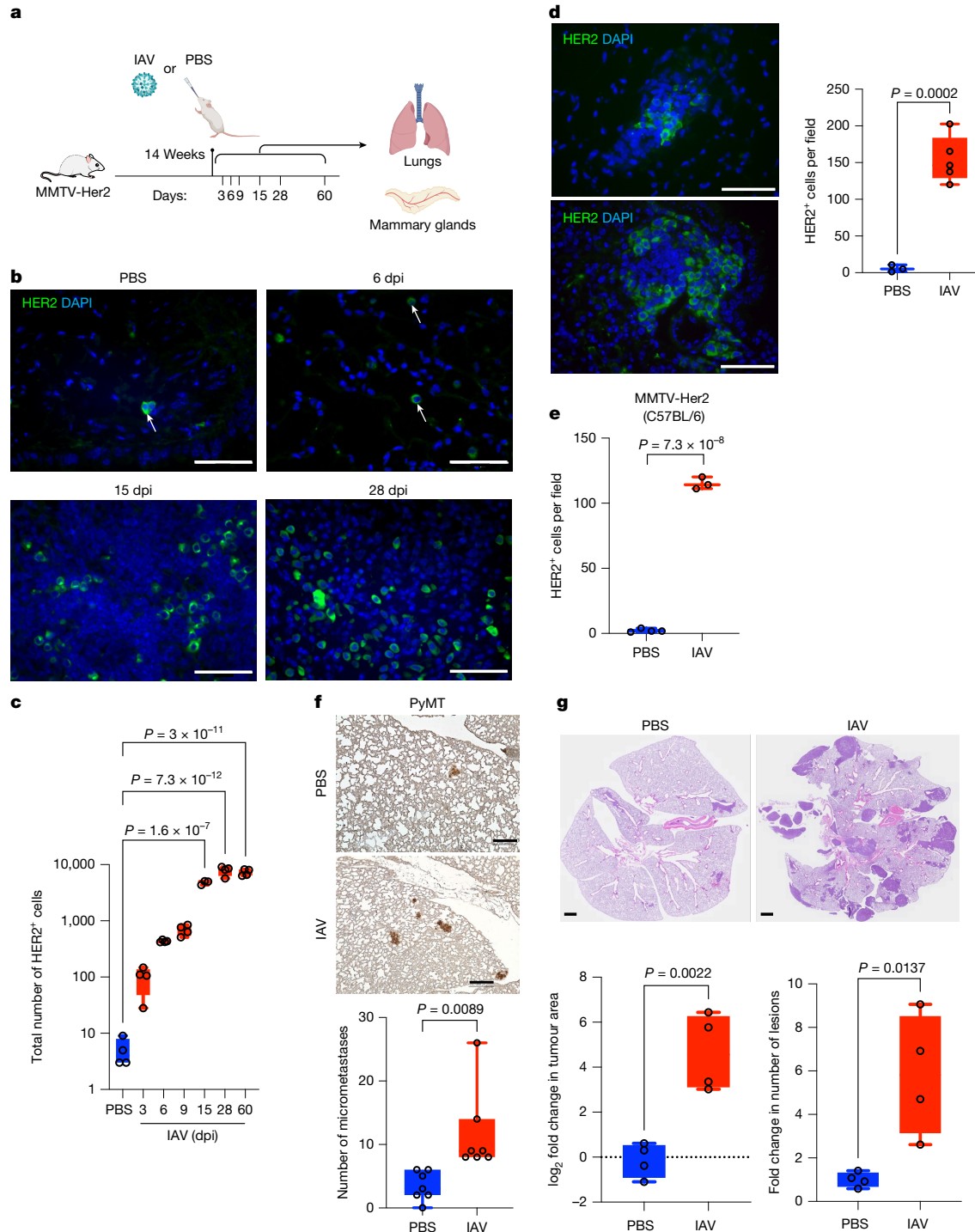

**Fig. 1 | Influenza A virus infection increases DCC in lungs. a**, MMTV-Her2 female mice in an FVB background were infected with a sublethal dose of Puerto Rico A/PR/8/34 H1N1 IAV by intranasal administration. Lungs and mammary glands were taken for analysis at the time points indicated after infection. **b,c**, Immunofluorescence (**b**) and quantification of HER2$^+$ cells in lungs (**c**) at 3, 6, 9, 15, 28 and 60 dpi. The total number of HER2$^+$ cells from three sections of the whole lung were quantified ($n = 4$ per group, $n = 3$ at 15 dpi). Lung sections were stained with DAPI (blue) and HER2 (green) as a marker for DCCs (**b**). **d**, Immunofluorescence and quantification of HER2$^+$ cells in lungs 9 months after an influenza infection ($n = 3$ PBS, $n = 5$ IAV). **e**, Quantification of HER2$^+$ cells in C57BL6/J MMTV-Her2 mouse lungs at 15 dpi with IAV ($n = 4$ PBS, $n = 3$ IAV). **f**, Immunohistochemistry and quantification of PyMT$^+$ micrometastases defined by lesions with an area of less than 0.03 mm$^2$ ($n = 7$ per group). **g**, EO771 mammary tumour cells were implanted into the mammary fat pads of C57BL/6

mice ($n = 4$ per group) and infected with IAV or PBS control after 31 days (experiment 1) and 20 days (experiment 2). The mice were implanted with $2 \times 10^5$ (experiment 1) or $1 \times 10^6$ EO771 (experiment 2) cells across two experiments, and combined results are shown. Lungs were taken for analysis 18 days (experiment 1) and 17 days (experiment 2) after infection and stained with H&E, and the tumour area and the numbers of lesions were quantified. For each experiment, the average of the quantification of the PBS-treated mouse lungs was set to 1, so that a fold change could be calculated. Significance was determined by one-way analysis of variance (ANOVA). All box-and-whisker plots are presented as maximum value (top line), median value (middle line) and minimum value (bottom line), with all data points shown as dots. Scale bars: **a** and **d**, 25 μm; **f**, 200 μm; **g**, 1 mm. Illustration in **a** created using BioRender (De Dominici, M., https://BioRender.com/i40c047; 2025). All replicates are biological.

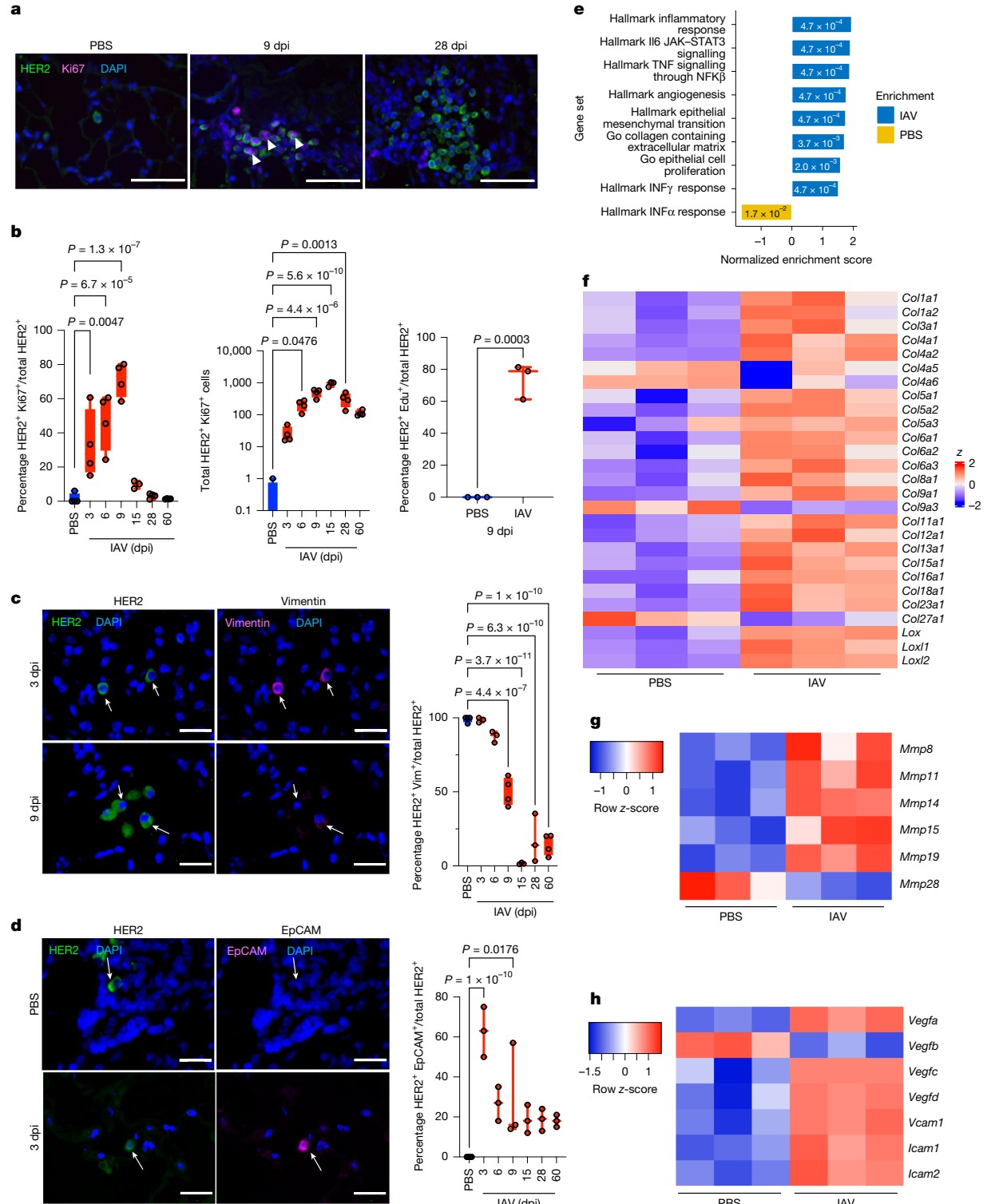

**Fig. 2 | Influenza A virus infection promotes dormant DCC proliferation and phenotypic change. a,b**, Immunofluorescence (**a**) and quantification (**b**) of Ki67⁺ HER2⁺ cells in lungs after IAV infection. Lung sections from naive and IAV-infected mice were stained with antibodies against HER2 (green), Ki67 (magenta) and DAPI (blue) (**a**). Percentage of Ki67⁺ HER2⁺ cells (**b**, left), absolute number of Ki67⁺ HER2⁺ cells across three lung sections (middle, *n* = 4 per group, *n* = 3 at 15 dpi) and detection of EdU incorporation (right, *n* = 3 per group). **c,d**, Immunofluorescence and quantification of vimentin⁺ (Vim⁺) (**c**) and EpCAM⁺ HER2⁺ (**d**) cells in lungs after influenza infections, in which lung sections from naive and IAV-infected mice were stained with HER2 (green) and vimentin (magenta) (*n* = 3 per group, *n* = 4 PBS, 9 dpi, 60 dpi) or HER2 (green) and EpCAM (magenta) (*n* = 3 per group). Graphs show the percentage of vimentin⁺ HER2⁺ (**c**)

or EpCAM⁺ HER2⁺ (**d**) cells. In **a**–**d**, statistical significance relative to PBS samples is shown, as determined by one-way ANOVA. **e**, GSEA analyses comparing DCCs from lungs of uninfected (PBS) and IAV-infected MMTV-Her2 mice at 9 dpi. See Supplementary Fig. 2 for the gating strategy used for sorting. **f**–**h**, Heatmaps of significantly differentially expressed collagen (Col) isoforms/lysyl oxidase (**f**), metalloproteinase (Mmp) (**g**) and vascular endothelial growth factor (Vegf) and intercellular adhesion molecule (Icam)/vascular cell adhesion molecule-1 (*Vcam1*) genes (**h**). All box-and-whisker plots are presented as maximum value (top line), median value (middle line) and minimum value (bottom line) with all data points shown by dots. Scale bars: **a**, 25 μm; **c** and **d**, 10 μm. All replicates are biological.

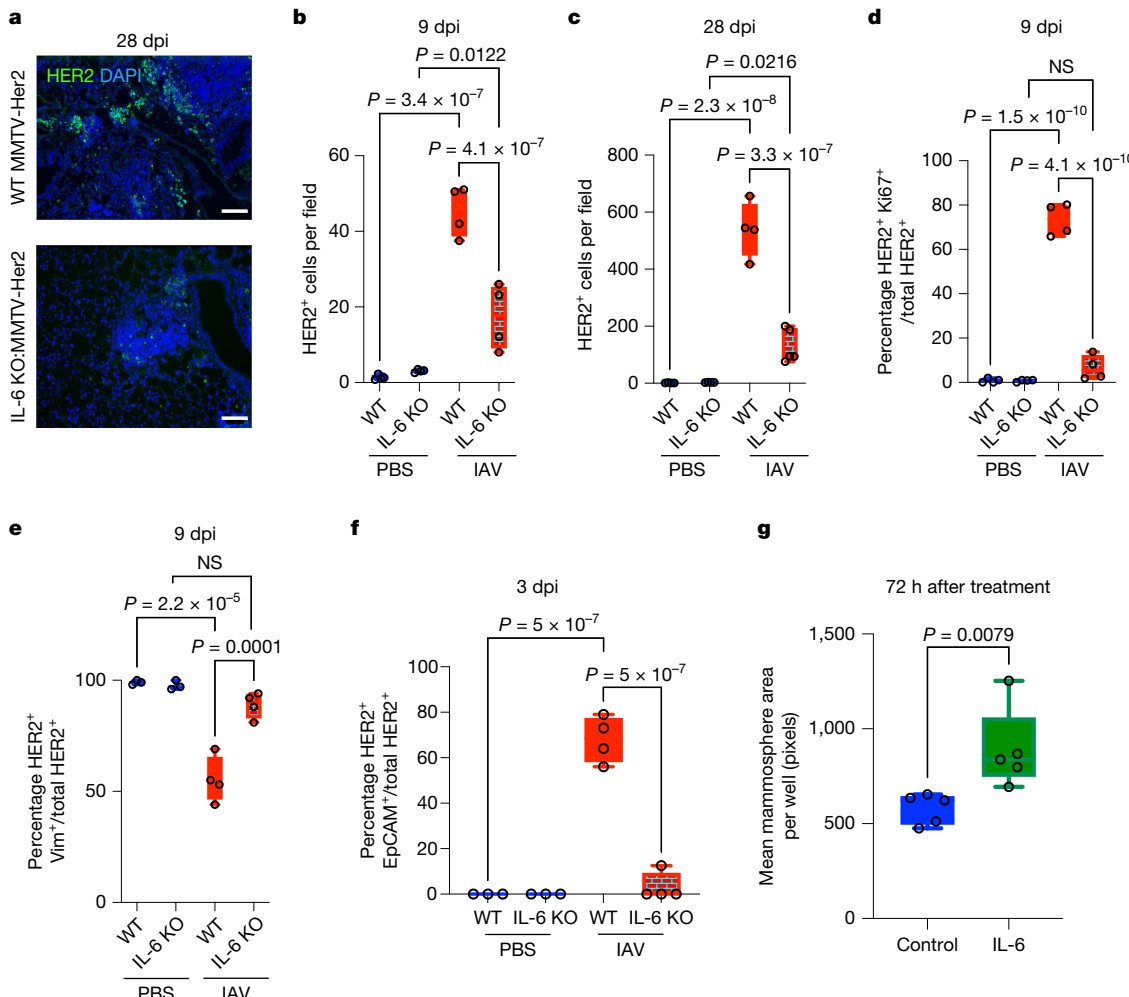

**Fig. 3 | IL-6 contributes to the awakening of dormant DCCs and proliferation.** **a**–**c**, Lung sections of MMTV-Her2 or IL-6 KO:MMTV-Her2 mice at 9 and 28 dpi with IAV (or PBS) were stained for HER2 (green) and DAPI (blue) (**a**) and quantified at 9 dpi (**b**) and 28 dpi (**c**). Scale bars: 50 μm. **d**, Quantification of the percentage of HER2⁺ Ki67⁺ cells in MMTV-Her2 and IL-6 knockout:MMTV-Her2 at 9 dpi (*n* = 4 per group). NS, not significant. **e**,**f**, Quantification of the percentage of vimentin⁺ HER2⁺ (**e**) and EpCAM⁺ HER2⁺ (**f**) cells in MMTV-Her2 and IL-6 knockout: MMTV-Her2 mice at 9 dpi (vimentin) and 3 dpi (EpCAM) (*n* = 3 for PBS, *n* = 4 IAV). Significance was determined by one-way ANOVA. **g**, Mean mammosphere area per well for HER2⁺ organoids after treating with 10 ng ml⁻¹ IL-6 (*n* = 5 per group) (**g**); significance was calculated by Mann–Whitney test. All box-and-whisker plots are presented as maximum value (top line), median value (middle line) and minimum value (bottom line) with all data points shown by dots. All replicates are biological.

histocompatability complex class I), which could contribute to the avoidance of immune elimination (Supplementary Data 1).

## IAV-induced DCC awakening requires IL-6

Inflammatory cytokines such as IL-6 and IL-1 are known to promote cancer malignancy and metastases[21–23]. Furthermore, IL-6 produced during acute inflammation resulting from biopsy or chemotherapy contributes to the development of lung metastatic outgrowth of disseminated mammary tumour cells[9,10]. IL-6 is abundantly produced during IAV infection, partly because of the replication of the virus in lung epithelial cells[24]. Similarly, we also detected high levels of IL-6 in bronchoalveolar lavage fluid (BALF) from wild-type and MMTV-Her2 mice after IAV infection, with very low levels of IL-1β (Extended Data Fig. 2e,f). Moreover, we observed clear activation of the IL-6 signalling pathway in DCCs after IAV infection (Fig. 2e and Extended Data Fig. 2c). Furthermore, in vitro IAV infection of primary mouse tracheal epithelial cells induced significantly increased secretion of IL-6 (Extended Data Fig. 2g), indicating that epithelial cells are a main source of IL-6 after IAV infection.

To determine whether IL-6 production triggered by IAV infection contributes to the awakening of dormant DCCs, we used MMTV-Her2

mice crossed with *Il6*-knockout (KO) mice[25]. MMTV-Her2 and *Il6*-KO:MMTV-Her2 mice were infected with IAV and lungs taken for analysis at 9 and 28 dpi (at this dose of IAV, all mice recovered without excessive weight loss). Before infection, there was no difference in the number of dormant HER2⁺ cells between *Il6*-KO:MMTV-Her2 and MMTV-Her2 lungs (Fig. 3b,c), and these mice developed primary tumours requiring euthanization with similar timing at older ages (Extended Data Fig. 3a). Thus, IL-6 is not required for primary tumour growth or for early cancer-cell dissemination to the lungs. Strikingly, the number of HER2⁺ cells in lungs of IAV-infected *Il6*-KO:MMTV-Her2 mice was markedly decreased compared with infected MMTV-Her2 mice at both 9 and 28 dpi (Fig. 3a–c), with substantial reductions in Ki67⁺HER2⁺ cells, indicative of maintained DCC dormancy (Fig. 3d). Furthermore, in the MMTV-PyMT mouse model of breast cancer metastasis, IAV-induced proliferation of PyMT⁺ small lesions and formation of micro-metastases in the lungs was dampened by IL-6 deficiency (Extended Data Fig. 3b,c). Similar IL-6 dependency was found for the increased lung metastatic burden after IAV infection in the EO771 cell model (Extended Data Fig. 3d,e). Staining for vimentin and EpCAM demonstrated that most HER2⁺ cells in the lungs of *Il6*-KO:MMTV-Her2 mice retain vimentin expression and maintain EpCAM-negative status, which together with

the failure to enter the cell cycle, as shown through immunofluorescence for Ki67, supports an IL-6 requirement for infection-induced DCC conversion from dormancy to awakening (Fig. 3e,f).

We next digested mammary glands from 3-month-old MMTV-Her2 mice, grew them under organoid conditions and treated them with vehicle or IL-6. Mammary mammospheres treated with IL-6 showed significant increases in overall size (Fig. 3g and Extended Data Fig. 3f). Similar increases in organoid size were observed for EO771 cells (Extended Data Fig. 3g). Although in vivo solitary DCC dormancy in the lung alveoli[16,26] could not be fully replicated in vitro, mammosphere initiation assays from single cells to proliferative clusters have been used to study how perturbations affect single-cell growth initiation or growth arrest[14,16]. Despite its limitations, this assay confirmed that IL-6 directly affects HER2+ mammary tumour cells in a solitary state, which reproduced the IL-6 response observed in vivo. Overall, these results indicate that IAV infection-triggered IL-6 has a key role in mediating the awakening of dormant DCCs.

## CD4+ T cells maintain IAV-awakened DCCs

Although IL-6 was essential for the awakening and the initial marked expansion of DCCs, minimal levels of IL-6 were detected in BALF of MMTV-Her2 mice 15 dpi (Extended Data Fig. 2e), indicating the presence of other factors that promote persistence after the expansion of DCCs at later times post-infection. Whereas recruitment of neutrophils to the lung occurs by 3 dpi with IAV, CD4+ T cells, CD8+ T cells and B cells accumulate in the lung from around 9 dpi in both wild-type and MMTV-Her2 mice (Extended Data Fig. 4a–d). Infection with IAV has also been shown to trigger the formation in the lungs of inducible bronchus-associated lymphoid tissues (iBALTs), which are lymphoid organizations that include primarily CD4+ and B cells. iBALTs can be detected in the lungs long after infection (up to 100 dpi)[27]. Accordingly, we also detected these CD4+ cell and B cell-enriched lymphoid organizations in the lung sections of wild-type and MMTV-Her2 mice 28 dpi (Fig. 4a and Extended Data Fig. 4e–h,k). As expected, B cells in these iBALTs are also positive for the germinal centre B cell marker GL7. In contrast to CD4+ cells, very few CD8+ cells were present in iBALT in either wild-type or MMTV-Her2 mice (Fig. 4a and Extended Data Fig. 4i). Interestingly, co-staining of CD4 with HER2 revealed the selective presence of DCCs in proximity to high-density clusters of CD4+ cells and the near absence of HER2+ cells in regions lacking CD4+ cells (Fig. 4b and Extended Data Fig. 4j). Moreover, consistent with the upregulation of many collagen genes in DCC after IAV infection (Fig. 2f), we observed substantial increases in collagen deposition selectively in iBALT after IAV infection in MMTV-Her2 mice relative to wild-type mice (Extended Data Fig. 4l,m). Collagens have been shown to limit T cell infiltration and activity[28].

To determine whether CD4+ cells maintain awakened DCCs later after IAV, CD4+ cell depletion was done before infection (−1 dpi) and HER2+ cells were examined in lung sections (Extended Data Fig. 5a,b). CD4+ cell depletion reduced the numbers of awakened DCCs at 28 dpi (Fig. 4c–e). However, CD4+ cell depletion did not affect DCC numbers or proliferation at 9 dpi (Extended Data Fig. 5c,d), consistent with the accumulation of CD4 cells late during infection (Extended Data Fig. 4b). Furthermore, the number of lung DCCs 28 dpi was also decreased when CD4+ cell depletion was initiated at 10 dpi (Fig. 4d). Thus, CD4+ cells contribute to the maintenance of awakened DCCs later after IAV infection. Together, these data show that IL-6 (but not CD4+ cells) contributes to the initial awakening and expansion of dormant DCCs, but that later during the infection, following the recruitment of T cells, CD4+ cells are required for the maintenance of the awakened DCCs.

Previous studies have shown how neutrophil extracellular traps produced during inflammation can awaken DCCs in lungs[29]. However, in contrast to CD4+ cell depletion, the depletion of neutrophils with an anti-Ly6G antibody at the time of IAV infection did not alter the numbers or Ki67 positivity of HER2+ cells in the lungs (Extended Data Fig. 5b,e–g).

Similarly, depletion of CD8+ cells before infection had no effect on the presence of DCCs (Fig. 4e), consistent with the paucity of CD8+ cells in lungs 28 dpi (Fig. 4a). Thus, maintenance of the awakened lung DCCs following IAV infection is selectively dependent on the presence of CD4+ cells.

Interestingly, although only a low number of dispersed CD8+ cells were present in lungs 28 dpi in MMTV-Her2 mice, we found an increased accumulation of CD8+ cells in lungs of infected mice when CD4+ cells were depleted (Fig. 4f). These results indicated that CD4+ cells may repress the recruitment of CD8+ cells to the lung, potentially compromising immune surveillance against awakened DCCs. We therefore tested the effect of depleting both CD4+ and CD8+ cells on the maintenance of lung DCCs following IAV infection. Although CD4+ cell depletion resulted in a marked reduction of HER2+ cells in the lungs of infected mice, the dual depletion of CD8+ cells and CD4+ cells partly restored the numbers of HER2+ cells in the lungs (Fig. 4e). Thus, CD4+ cells maintain awakened lung DCCs after IAV infection, partly by suppressing the CD8+ immune response.

We then examined whether the presence of awakened DCCs in the lungs following IAV infection could reprogram recruited T cells to a more suppressive or suppressed state by performing single-cell RNA-seq (scRNA-seq) of lungs of wild-type and MMTV-Her2 mice 9 and 15 dpi with IAV, at points in which the accumulation of T cells in the lungs is high (Extended Data Fig. 4b,c). As expected, multiple immune cell types, such as macrophages, natural killer cells, B cells, effector and memory CD4+ cells and effector and memory CD8+ cells, were present in the lung at both 9 and 15 dpi (Extended Data Fig. 6a–c). We had two replicates for most conditions, which exhibited very similar gene expression patterns (Extended Data Fig. 6d). IAV infection induced type I and II interferon responses and other innate immune pathways across these cell types, as expected (Extended Data Fig. 7a). Interestingly, there was substantially increased expression of a selective subset of genes in effector CD4+ cells from MMTV-Her2 mice relative to effector CD4+ cells in wild-type mice. In particular, *Tnfaip3*, *Zfp36l2*, *Dusp5*, *Dusp1*, *Cxcr4*, *Klf6*, *Pdcd4* and *Ctla4* were highly upregulated in effector CD4+ cells from MMTV-Her2 mice relative to wild-type mice (Fig. 4g and Supplementary Data 2). The differential expression of some of these genes was validated at the protein level either by flow cytometry (*Cxcr4*) or by western blot analysis (*Dusp5*) (Extended Data Fig. 7b). *Tnfaip3*, which encodes the E3 ubiquitin ligase A20, suppresses the antitumour activity of CD8+ cells[30], whereas *Zfp36l2*, which regulates RNA stability, restrains CD8+-cell activation and expansion[31,32]. *Dusp5*, which encodes a dual phosphatase, suppresses T cell proliferation and promotes their survival[33]. *Klf6* and *Dusp1* are markers of central and resident memory T cells[34,35]. *Ctla4* induces *Pdcd4* in cytotoxic T cells, and *Pdcd4* deficiency enhances their antitumour effector functions[36]. These results indicated an impaired effector-cell phenotype in CD4+ cells from MMTV-Her2 mice relative to wild-type mice. Furthermore, comparing memory CD4+ cells from MMTV-Her2 mice versus wild-type mice at day 15 after IAV infection revealed similar changes in gene expression, with increased *Tnfaip3*, *Klf6*, *Dusp1*, *Dusp5* and *Dusp10* (Extended Data Fig. 7c). Similar changes were also observed in CD8+ cells isolated from MMTV-Her2 versus wild-type mice at 15 dpi (Extended Data Fig. 7d,e). Overall, these data indicated a bias to a more memory phenotype and increased survival with less effector function in CD4+ and CD8+ cells of infected MMTV-Her2 mice. Further analysis confirmed the increase in the ratio of memory to effector cells for both CD4+ and CD8+ cells in infected MMTV-Her2 mice relative to wild type (Extended Data Fig. 7f), consistent with DCC-mediated suppression of T cell effector function.

Moreover, CD4+ cell expression of genes important for T cell activation, including *Gadd45b* and *Slfn2* (refs. 37–39), was reduced in CD4+ effector cells in infected HER2+ mice relative to wild-type mice (Fig. 4g and Supplementary Data 2). Interestingly, the expression of multiple mitochondrial genes (including *mt-Atp6*, *mt-Nd1*, *mt-Co3* and *mt-Nd3*) was greatly reduced in CD4+ cells from MMTV-Her2 mice, indicating

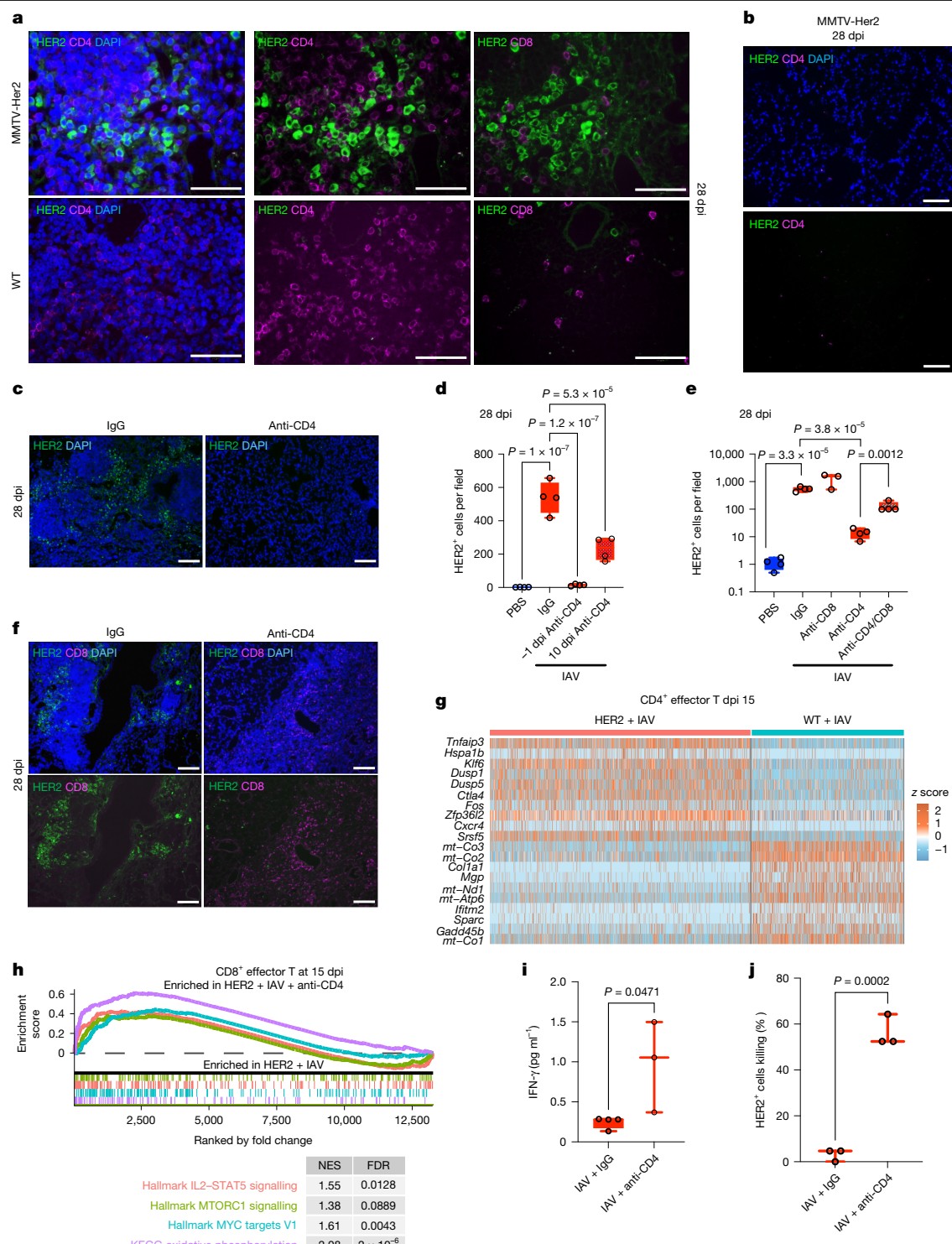

**Fig. 4 | CD4+ cells are required to maintain expanded HER2+ DCCs after IAV infection. a**, Adjacent lung sections of IAV-infected mice at 28 dpi were stained for HER2 (green) and CD4 (magenta, left and middle) or HER2 (green) and CD8 (magenta, right). **b**, As in **a** for IAV-infected MMTV-Her2 mice at 28 dpi, but for a region lacking CD4+ cells. **c**,**d**, Lung sections of MMTV-Her2 mice without or with CD4 depletion starting at −1 dpi or 10 dpi and taken for analysis at 28 dpi were stained for HER2 (green) and DAPI (blue) (shown for CD4 depletion starting at −1 dpi) (**c**). The number of HER2+ cells was quantified (n = 4 per group) (**d**). **e**, Quantification of HER2+ cells from MMTV-Her2 mice with CD4, CD8 or CD4/CD8 depletion (on −1 dpi) at 28 dpi (n = 4 per group, n = 3 CD8 depletion). **f**, Lung sections of MMTV-Her2 and CD4-depleted MMTV-Her2 mice at 28 dpi were stained for HER2 (green), CD8 (magenta) and DAPI (blue). **g**, Heatmap of the top 20 differentially expressed genes from scRNA-seq

comparing CD4+ effector T cells from MMTV-Her2 + IAV versus wild type + IAV mice at 15 dpi. **h**, GSEA analysis showing pathway enrichment in effector CD8+ T cells in CD4-depleted MMTV-Her2 + IAV versus control MMTV-Her2 + IAV mice at 15 dpi. **i**, Concentration of IFNγ in a supernatant of enriched CD8+ cells enriched from lungs of MMTV-Her2 mice with or without CD4 depletion at 15 dpi and activated by anti-CD3/CD28 antibody (n = 4 IAV + IgG, n = 3 IAV + anti-CD4). **j**, Ex vivo CD8+ cytotoxic assay in which HER2+ cells were incubated with the same CD8+ cells (n = 3 per group). Significance was determined by one-way ANOVA (**d** and **e**) or two-tailed Student's t-test (**i** and **j**). All box-and-whisker plots are presented as maximum value (top line), median (middle line) and minimum (bottom line) with all data points shown as dots. Scale bars: **a**, 25 μm; **b**, **c** and **f**, 50 μm. All replicates are biological.

a reduced mitochondrial content in these cells (Fig. 4g). The reduced mitochondrial content could result from increased autophagy mediated by the upregulation of *Tnfaip3*, because *Tnfaip3* deficiency has been shown to increase mitochondrial content in CD4[+] cells by promoting autophagy[40]. Reductions in mitochondrially and nuclear-encoded transcripts crucial for oxidative phosphorylation were observed across T cell types, including CD8 cells, in infected MMTV-Her2 mice by 15 dpi (but not at 9 dpi, before full DCC expansion), consistent with reduced T cell effector function (Extended Data Fig. 8 and Supplementary Note 1). We validated the reduction in mitochondria content in CD4[+] and CD8[+] cells from infected MMTV-Her2 mice relative to wild-type mice by Mitotracker staining and flow cytometry (Extended Data Fig. 7g,h).

Pathway analysis further supports a compromised effector function of CD4[+] cells from MMTV-Her2 mice. Over-representation analysis (ORA) of scRNA-seq comparing MMTV-Her2 and wild-type mice after IAV infection showed reduced interferon responses in MMTV-Her2 mice across multiple immune cell types (Extended Data Fig. 9a,b), as shown for macrophages (Extended Data Fig. 9c,d) and T cells (Extended Data Fig. 10a), consistent with the lower levels of type I and II IFNs produced in the lungs of infected MMTV-Her2 mice (Extended Data Fig. 10b; see Supplementary Data 3 for all significant gene set enrichment analysis (GSEA) results). Moreover, we observed an increase in the ratio of M2 to M1 macrophages in infected MMTV-Her2 mice, accompanied by significant changes in gene expression (Extended Data Fig. 9e). We also observed significant suppression of the oxidative phosphorylation pathway in macrophages and T cells from infected MMTV-Her2 mice relative to infected wild-type mice by 15 dpi (Extended Data Figs. 9c,d and 10a). Together, these results indicate that the presence of DCCs skews the macrophage phenotype and impairs CD4[+] and CD8[+] cell activation in response to IAV infection, favouring tumour-cell persistence.

Crucially, GSEA comparing CD8[+] T cells from MMTV-Her2 mice with IAV infection (15 dpi) and CD4 depletion revealed significant increases in pathways involved in CD8[+] cell activation, such as IL-2–STAT5, MTORC1 and type 1 and 2 interferon signalling pathways, indicating that both effector and memory CD8[+] cells are more proliferative and activated when CD4[+] cells are depleted (Fig. 4h and Extended Data Fig. 10c). Mitochondrial and respiration phenotypes were also restored in CD8[+] cells by CD4[+] cell depletion by 15 dpi (Extended Data Fig. 8). Thus, the effect of CD4[+] cell depletion on eliminating DCCs seems to be mediated by enhanced CD8[+] cell responses against DCCs.

To further determine whether CD4[+] cells in MMTV-Her2 mice are suppressing the activation of antitumour CD8[+] cells, we isolated lung CD8[+] cells from MMTV-Her2 mice or CD4[+] cell-depleted MMTV-Her2 mice at 15 dpi with IAV. Increased production of IFNγ was specifically detected in lung CD8[+] cells isolated from CD4-depleted infected MMTV-Her2 mice following ex vivo activation with anti-CD3/anti-CD28 antibodies (Fig. 4i). Moreover, we investigated the ability of these CD8[+] cells to kill mammary cancer cells ex vivo using cultures of primary HER2[+] cells isolated from the mammary glands of MMTV-Her2 mice. Lung CD8[+] cells from CD4-depleted MMTV-Her2 mice after IAV infection had superior killing activity against HER2[+] tumour cells compared with lung CD8[+] cells from IAV-infected Her2 mice (Fig. 4j and Extended Data Fig. 11a). By contrast, CD8[+] cells from IAV-infected wild-type mice had minimal killing activity against MMTV-Her2 tumour cells. Lung CD8[+] cells from CD4-depleted Her2 mice were also superior in killing MET-1 cancer cell lines generated from MMTV-PyMT mice that lacked HER2 (Extended Data Fig. 11b). Together, these data demonstrate the specific anti-mammary tumour activity of CD8[+] cells from IAV-infected MMTV-Her mice and the suppressive effect that lung CD4[+] cells have on antitumour CD8[+] cells.

## SARS-CoV-2 infection awakens DCC

To determine whether SARS-CoV-2 infection of lungs can promote the reawakening of dormant DCCs, we performed studies (analogous to those in Fig. 1 with IAV) using mouse-adapted SARS-CoV-2 that recognizes mouse ACE2, termed MA10, obtained by genetic modification of the spike gene[41] followed by serial passages in mice[42]. This SARS-CoV-2 strain induces a COVID-19-like disease in mice including acute lung injury characterized by impaired pulmonary function, diffuse alveolar damage and infiltration of immune cells[42]. Similar to IAV infection, MA10 infection induced the production of high levels of IL-6 and IFNα in lungs (Extended Data Fig. 11c,d). Lower levels of IFNβ, IFNγ and IL-1β were also detectable in the BALF. Importantly, infection of MMTV-Her2 mice with MA10-SARS-CoV-2 resulted in a striking increase in HER2[+] cells by 28 dpi (Fig. 5a and Extended Data Fig. 12a). Analyses of earlier time points (3 and 9 dpi) demonstrated a stepwise increase in the number of HER2[+] cells and Ki67[+] HER2[+] cells after MA10 infection (Fig. 5b), with transient increases in EpCAM positivity and reductions in vimentin positivity (Extended Data Fig. 12b), as observed after IAV infection. Notably, we demonstrated that these changes in HER2[+] DCC proliferation, expansion and phenotypic transitions require IL-6, because MA10 infection-dependent changes are significantly reduced in *Il6*-KO:MMTV-Her2 mice, with no detectable change in MA10 virus replication (Fig. 5c and Extended Data Figs. 11f and 12c).

## COVID-19 associates with cancer progression

The COVID-19 pandemic presents a unique opportunity to study the effect of pulmonary virus infections on cancer progression because, unlike influenza, data on virus infections and resulting outcomes were systematically collected in the first years of the pandemic. First, we analysed data from the UK Biobank to determine whether a SARS-CoV-2 positive test result, among a population of cancer survivors, was associated with an increased risk of all-cause, non-COVID-19 and cancer mortality (Extended Data Fig. 12d). To reduce potential confounding from vaccination and the use of at-home SARS-CoV-2 tests, we limited the analysis to subjects with PCR tests conducted before December 2020. Stratification of results based on primary tumour type and metastatic disease was not possible, owing to an insufficient number of observations.

In the full study population with follow-up till 31 December 2022, which included 4,837 participants with a cancer diagnosis before 1 January 2015 (indicating inferred remission), we observed 413 deaths (298 in the test negatives and 115 in the test positives), yielding an odds ratio of 4.50 (95% confidence interval: 3.49–5.81) (Fig. 5d). When we excluded 120 deaths directly attributed to COVID-19, SARS-CoV-2-positive cases still showed increased mortality with an odds ratio of 2.56 (95% CI: 1.86–3.51). Based on the 128 cancer-related deaths as an outcome, we estimated a nearly twofold increase in cancer mortality in those who tested positive compared with those who tested negative (odds ratio, 1.85; 95% CI: 1.14–3.02).

Analyses of participants diagnosed with cancer at least 10 years before the pandemic (before 1 January 2010) showed results consistent with the main analyses, although there was some loss of power owing to the reduced number of cases. SARS-CoV-2-positive participants had increased risks for all-cause mortality (odds ratio, 5.24; 95% CI: 3.66–7.48), non-COVID-19 mortality (odds ratio, 2.58; 95% CI: 1.64–4.06) and cancer mortality (odds ratio, 1.80; 95% CI: 0.85–3.80) compared with SARS-CoV-2-negative participants.

When we reduced the follow-up period from 31 January 2022 to 1 December 2020, the odds ratio increased from 1.85 (1.1.4–3.02) to 8.24 (3.43–19.77) (Fig. 5d and Extended Data Table 2), with decreasing odds ratios across follow-up times reduced by subsequent 6-month intervals. This trend indicates that the strength of the association is greatest shortly after infection and diminishes over time. Some individuals in the test-negative group may have become infected during follow-up. These evolving differences in infection risk over time may have contributed to a weakening of the observed association. Even so, our findings indicate that the increased risk of cancer mortality is greatest in the first few months after SARS-CoV-2 infection. Together,

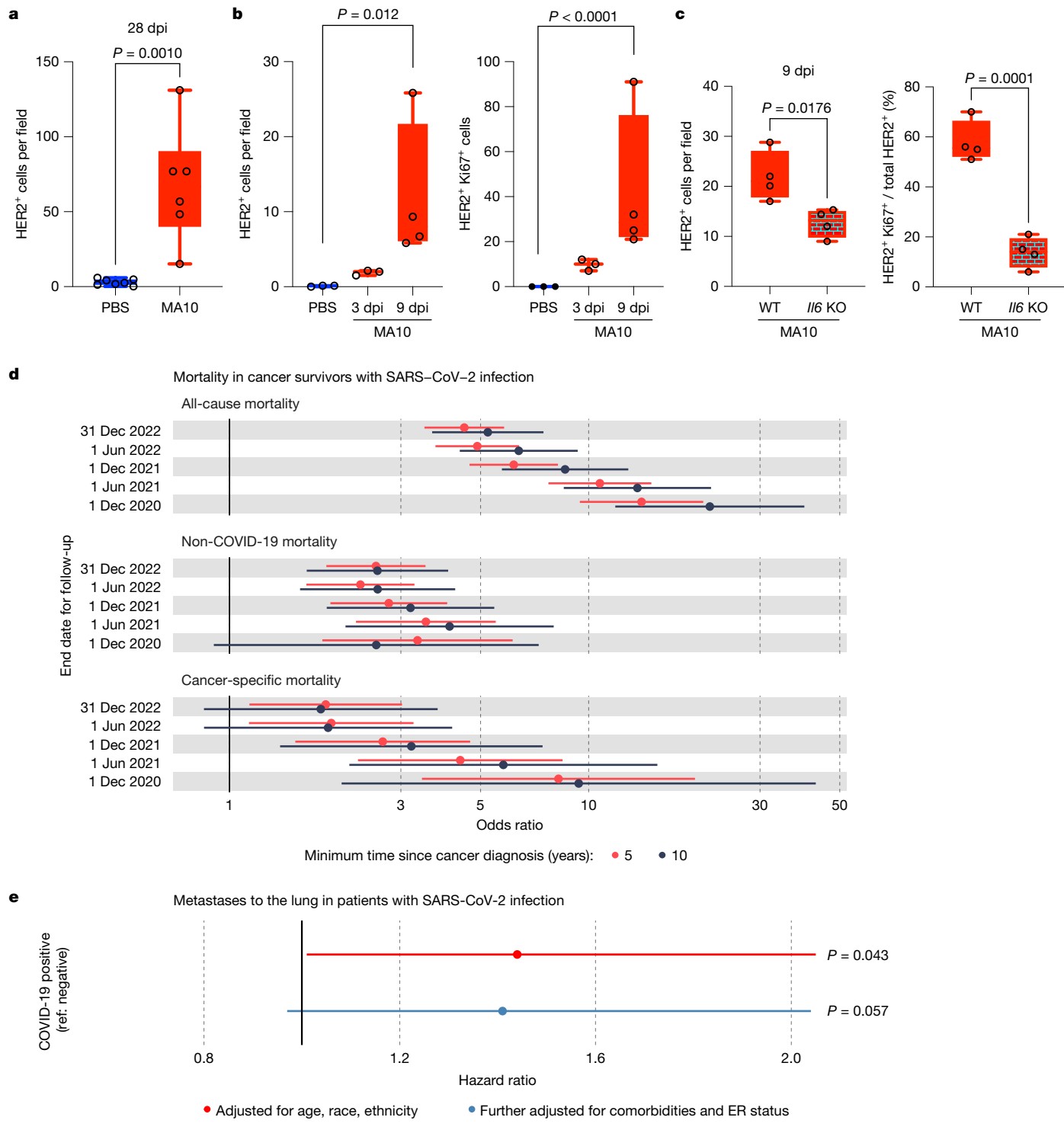

**Fig. 5 | SARS-CoV-2 infection increases cancer progression, metastasis to lungs and mortality. a**, Quantification of HER2⁺ cells across three lung sections in C57BL6/J MMTV-Her2 mouse lungs at 28 dpi with MA10 SARS-CoV-2 (*n* = 6) or PBS control (*n* = 7). **b,c**, Quantification of HER2⁺ cells and percentage of Ki67⁺ HER2⁺ cells at 3 dpi and 9 dpi with MA10 in the lungs of FVB MMTV-Her2 mice (*n* = 3 PBS at 3 dpi and *n* = 4 at 9 dpi) (**b**) and comparing MMTV-Her2 mice without (WT) or with *Il6* knockout (*n* = 4 per group) (**c**). Significance was determined by two-tailed Student's *t*-test (**a,c**). For **b**, we applied a negative binomial model for HER2⁺ cells per field comparing 9 dpi and PBS control (to accommodate the potential overdispersion); for HER2⁺Ki67⁺ cells per field, we determined whether cells per field in infected groups were significantly higher than 0 (all PBS samples were 0) using a negative binomial model. All replicates are biological. **d,e**, Epidemiological studies. **d**, Analyses from the UK Biobank examining the association between a SARS-CoV-2 test being positive or negative and the risk of all-cause, non-COVID-19 and cancer-related mortality in cancer survivors with cancer diagnoses more than 5 (red) or 10 (black) years before the start of the COVID-19 pandemic. The analyses compared mortality risks between positive-test and negative-test participants, using censoring dates for death events from 1 December 2020 to 31 December 2022. **e**, Analyses from the Flatiron Health database evaluating the hazard ratio for the risk of progression to metastatic lung disease among patients with breast cancer who developed COVID-19 disease versus those who did not, adjusted for age, race and ethnicity (red) and multivariate analyses after also including co-morbidities, breast cancer subtype (for example, ER status) and other potential confounding factors (blue). All box-and-whisker plots are shown as maximum value (top line), median (middle line) and minimum (bottom line) with all data points shown as dots.

the data indicate a markedly increased risk of death from cancer for cancer survivors who contract a SARS-CoV-2 infection.

We used the Flatiron Health database with 36,845 female patients with breast cancer with complete information to specifically determine whether women with a primary diagnosis of breast cancer experienced an increased risk of progression to metastatic disease in the lungs after COVID-19 (Extended Data Table 3 and Extended Data Fig. 12e). Crucially, female patients with breast cancer who experienced COVID-19 disease after their initial breast cancer diagnosis exhibited an age, race and ethnicity-adjusted hazard ratio of 1.44 (95% CI: 1.01, 2.05; $P = 0.043$) for subsequent diagnosis of metastatic breast cancer in the lungs (red in Fig. 5e and Extended Data Tables 3–5). Because we lack COVID test-negativity data in the Flatiron Health dataset, the COVID-19 negative group may include undiagnosed cases, probably underestimating the hazard ratio. Further adjusting our model for comorbidities and breast cancer subtype (oestrogen receptor (ER) status), we estimated a consistent, although slightly attenuated and not statistically significant, effect of COVID-19 on metastatic progression to the lungs with a hazard ratio of 1.41 (95%CI: 0.97, 2.04) (blue in Fig. 5e). These results indicate that the initial findings from the parsimonious model are robust and that potential confounding by additional covariates has a negligible effect in the Flatiron Health dataset. In all, these data show that COVID-19 increases lung metastasis risk in female patients with breast cancer.

## Discussion

Our results indicate that respiratory virus infections promote the awakening and expansion of previously seeded dormant cancer cells. This process unfolds in two distinct phases. First, an IL-6-dependent switch of DCCs from a mesenchymal phenotype to a hybrid state promotes expansion. Second, this expansion is followed by a return to quiescence and establishment of CD4+ cell niches that inhibit DCC elimination, partly through the suppression of CD8+ cells (Extended Data Fig. 12f). We further show how the presence of HER2+ tumour cells results in a suppressive phenotype for CD4+ cells, and that depletion of CD4+ cells leads to the elimination of influenza virus infection-expanded DCCs, dependent on CD8+ cells with restored effector activity. Importantly, we show that a mouse-adapted SARS-CoV-2 virus similarly leads to IL-6-dependent DCC expansion in lungs.

Since the onset of the COVID-19 pandemic, the potential influence of SARS-CoV-2 on cancer progression has been a crucial question in the research community[43–45]. Although species differences warrant caution in interpreting mouse data, UK Biobank analyses show that cancer survivors had increased cancer mortality after SARS-CoV-2 infection. This risk peaked in the months after infection, paralleling mouse models showing greater than 100-fold DCC expansion into metastatic lesions within two weeks. Analyses based on the Flatiron Health database further reveal a marked rise in metastatic lung disease among breast cancer survivors following COVID-19. Collectively, these findings underscore the substantial metastatic risk COVID-19 posed to cancer survivors, with dormant DCC reactivation potentially driving this phenomenon.

Our studies highlight the importance of developing interventions to minimize the risk of lung DCC awakening and metastatic disease in the millions of cancer survivors who experience respiratory virus infections. As well as primary prevention strategies, treatments for managing severe COVID-19 that have been approved by the US Food and Drug Administration include antagonistic antibodies against IL-6R[46] and orally available JAK1/2 inhibitors[47], raising the prospect of interventions that could reduce the risk of infection-induced metastatic cancer progression. The effectiveness and safety of these interventions, and the timing of their application to avoid impeding the resolution of the infection, will need to first be tested in preclinical and clinical studies.

In conclusion, our studies reveal how respiratory virus infections can increase cancer recurrence risk and underscore the need for public health and clinical strategies to mitigate the increased risk of metastatic progression associated with SARS-CoV2 and other respiratory virus infections.

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

## Methods

### Mouse strains, influenza infection and antibody treatments

Transgenic mouse models of breast cancer, using mouse mammary tumour virus (MMTV) long terminal repeats, are widely used. In brief, MMTV-PyMT and MMTV-erbB2/neu/Her2 (MMTV-Her2) mice express the oncogenes polyoma virus middle T antigen (PyMT) and rat *Erbb2* (encoding HER2), respectively, upstream of the MMTV promoter, which confers expression in the mammary epithelium, as described elsewhere[10,13,48]. The MMTV-PyMT transgene is congenic in the FVB mouse background (a gift from William Muller). Given that the MMTV-PyMT mice exhibit substantial lung tumour burden within a few months of life, we limit our analyses to newly awakened DCCs in this model (forming micrometastases, defined as lesions with an area of less than 0.03 mm²). The MMTV-Her2 transgene is congenic in the FVB (Jackson Laboratory, 002376) and C57BL/6 (a gift from Ramon Parsons, congenic in C57BL/6J by backcrossing from the FVB background[26]) backgrounds. MMTV-Her2 mice (FVB) were crossed with IL-6-knockout (KO) mice[9,25]. For an orthotopic model of breast cancer, EO771 breast cancer cells[49] (a gift from Diana Cittelly) were injected into the fourth right and left mammary fat pads with $2 \times 10^5$ or $1 \times 10^6$ cells per fat pad.

Eight-week-old MMTV-PyMT and 12- to 14-week-old MMTV-Her2 female mice were infected with 500 EIU Puerto Rico A/PR/8/34 H1N1 IAV through intranasal administration in 50 µl PBS. For viral administration, mice were anaesthetized using 5% induction isoflurane and 2% maintenance, performed with a SomnoFlo Low-Flow electronic vaporizer machine in an induction chamber. After ensuring adequate anaesthesia with slow and deep breathing, droplets of viral fluid were placed on the mouse's nostrils. The mouse inhaled the fluid through the nostrils. Once the fluid had been inhaled, the mouse was placed on a heating pad to recover.

For immune-cell depletion experiments, mice were injected intraperitoneally with rat IgG as a control (MP Biochemicals, MPBio 0855951), 100 µg anti-CD4 (Bio X cell, clone GK1.5, BP003-1), or 100 µg anti-CD8 (Bio X cell, clone2.43) 1 day before IAV infection and every 6 days afterwards, or 200 µg anti-Ly6G (Bio X cell, clone 1A8, BP0075-1) on the day of the influenza virus infection, then 24 h and every other day afterwards, until being euthanized. For 5-Ethynyl-2′-deoxyuridine (EdU) incorporation, mice were injected with 50 mg per kg EdU (Sigma Aldrich, BCK488-IV-FC-S) 4 h before euthanasia.

For both MMTV-PyMT and MMTV-Her2 mice, as a humane end point, mice were euthanized when the tumour reached 20 mm in any one dimension, tumours were ulcerated or infected, or if there was a major sign of discomfort, as determined by the institutional veterinarian. Mice were monitored every other day during the first week or until the tumour was palpated, and daily afterwards until the mice needed to be euthanized. Veterinary technicians in the institutional facility monitored the mice daily.

All mice were co-housed in specific pathogen-free animal facilities, maintained at 21 °C (±1 °C) and 35% humidity with a 14 h:10 h light:dark cycle (light 06:00–20:00). All the mice were backcrossed in the C57Bl/6J background for more than 10–12 generations. Only female mice were used for the studies. The average age of the mice was 12–24 weeks. An approved measure of $CO_2$ followed by cervical dislocation was used for euthanasia.

The University of Colorado Institutional Animal Care and Use Committee (IACUC) reviewed and approved all animal experiments (including humane end points described above), which were conducted in accordance with the NIH Guidelines for the Care and Use of Laboratory Animals.

### SARS-CoV-2 MA10 propagation

Mouse-adapted SARS-CoV-2 MA10 (BEI Resources, NR-55329) was propagated in Vero E6 cells (ATCC CRL-1586) as previously described[42]. In brief, low-passage Vero E6 monolayers were inoculated at a multiplicity of infection of 0.01 with SARS-CoV-2 MA10. When Vero E6 monolayers exhibited 70–75% cytopathic effect (2–3 dpi), supernatants were collected, clarified by centrifugation, supplemented with an additional 10% FBS, aliquoted and stored at −80 °C. SARS-CoV-2 titres were determined by plaque assay on Vero-E6 cells. Vero-E6 cells were maintained at 37 °C in Dulbecco's Modified Eagle medium (DMEM, HyClone 11965-084) supplemented with 10% fetal bovine serum (FBS), 10 mM HEPES (pH 7.3) and 100 U ml⁻¹ of penicillin-streptomycin.

### SARS-CoV-2 MA10 infection of mice

MMTV-Her2 female mice (in both C57BL/6J and FVB backgrounds) at 14–19 weeks of age were anaesthetized by intraperitoneal injection of a mixture of ketamine (80 mg per kg) and xylazine (7.5 mg per kg) in a volume of 100–200 µl. Fully anaesthetized mice were inoculated intranasally with $10^4$ PFU of SARS-CoV-2 MA10 diluted in PBS supplemented with 1% bovine calf serum by administration of 25 µl of inoculum in each nostril for a total volume of 50 µl. Mouse weights were collected daily for 15 days, and mice inoculated with SARS-CoV-2 MA10 exhibited weight loss beginning at 2 dpi, with greatest loss achieved at 3-4 dpi, as previously reported[42]. As controls, MMTV-Her2 mice were mock inoculated with 50 µl of PBS/1% bovine calf serum.

### SARS-CoV-2 MA10 viral titre from lungs

MA10 viral titre was determined as previously described[50]. Lung superior lobes were homogenized, serially diluted in DMEM with 2% FBS, HEPES, penicillin-streptomycin and incubated on Vero E6 cells for 1 h at 37 °C. Cells were then overlaid with 1% (w/v) methylcellulose in MEM with 2% FBS at 37 °C for 3 days. Overlays were removed afterwards, and the plates were fixed with 4% paraformaldehyde for 20 min at room temperature. Fixed plates were stained with crystal violet (0.05% w/v) in 20% methanol for 10 min. Infectious viral titres were determined by manually counting the plaques formed.

### Immunohistochemistry and immunofluorescence staining

Lungs and mammary glands were collected and fixed in 10% neutral buffered formalin overnight, transferred to 70% ethanol the next day and then embedded in paraffin. Tissues were sectioned (5 µm) and used for immunohistochemistry (IHC) and immunofluorescence. Slides were deparaffinized in three incubations of 15 min in Histo-clear (Fisher Scientific, 50-899-90147) then descending 10-min ethanol incubations: three at 100%, followed by 95% and 70% followed by 10 min of $H_2O$ incubation. Heat-induced antigen retrieval was done for 10 min in a pressure cooker in citrate buffer (10 mM citric acid, pH 6.0). For IHC, samples were incubated in 1% $H_2O_2$ for 15 min to block endogenous peroxidase activity. Permeabilization was done using 0.1% normal goat serum in 0.4% Triton-X 100 in PBS for 30 min. Sections were blocked for 1 h at room temperature with blocking solution (Abcam, AB64226) containing MOM blocking reagent (Vector Laboratories, MKB2213-1), incubated with primary antibodies (Supplementary Information Table 1) at 4 °C overnight in antibody diluent (Abcam, 64211), then washed 3 times for 30 min each in 0.1% triton-X 100 in PBS. For IHC samples, sections were incubated in ImmPRESS HRP goat anti-rabbit or rat IgG polymer detection kit (Vector Laboratories, MP-7451/MP7404) and ImmPACT DAB substrate, peroxidase HRP (Vector Laboratories, SK4105) according to the manufacturer's instructions. The IHC slides were mounted using micromount mounting medium (StatLab, MMC0126). For immunofluorescence, sections were incubated with secondary antibodies for 1 h at room temperature in antibody diluent (Abcam, 64211). Sections were then washed in 0.1% Triton-X 100 in PBS 3 times for 30 min each and were mounted using fluoroshield mounting media with DAPI (Abcam, 104139). Immunofluorescence images were collected using a Zeiss Axiovert 200-m fluorescence microscope. IHC images were collected using a Keyence BZ-X800 microscope. Section staining, image capturing and image analysis were done manually using ImageJ and were carried out by a researcher who was blinded to sample identities. Subsequent

grouping and graphing were done by a different lab member who was unblinded after image analyses and quantification were completed.

## Assessment of collagen deposition

Collagen deposition was assessed using Masson's Trichrome stain. The intensity of the stained areas was assessed using FRIDA software as described elsewhere[51].

## BALF processing

Bronchoalveolar lavage was done using 1 ml PBS (ThermoFisher, 14190-144) after mice were euthanized. BALF was collected and centrifuged at 500$g$ for 5 min at 4 °C. Supernatant was flash frozen in liquid nitrogen and stored at −80 °C until analysis. Red blood cells were lysed using haemolytic buffer (150 mM $NH_4Cl$, 1 mM $NaHCO_3$, 1.1 mM $Na_2EDTA$) for 3 min, flow buffer (PBS with 2% FBS and 2 mM EDTA) was added and cell suspensions were centrifuged at 500$g$ for 5 min at 4 °C. Cells were resuspended in flow buffer and counted manually.

## Cytokine detections

Cytokines in the BALF were measured using custom-made high-sensitivity multiplex assays from Meso Scale Discovery according to the manufacturer's instructions.

## Flow-cytometric analyses

Cells recovered from BALF were stained with antibodies (Supplementary Information Table 1). Alternatively, whole lungs were taken and digested using a method described elsewhere[52]. In brief, lung digestion mix (1.5 mg ml$^{-1}$ collagenase A (Sigma Aldrich, COLLA-RO), 0.4 mg ml$^{-1}$ deoxyribonuclease I (Worthington, LS002139), 10 mM HEPES pH 7.2, 5% FBS) was injected into the lungs through cannulae and lungs were incubated in a shaking incubator at 37 °C for 30 min followed by vigorous vortexing. Digested lungs were passed through a 50 μm cell strainer and red blood cells were lysed using haemolytic buffer for 3 min, flow buffer were added and cell suspensions were centrifuged at 500$g$ for 5 min at 4 °C. Single cells were resuspended in flow buffer and stained with antibodies (Supplementary Information Table 1) for flow cytometry. For mitochondrial mass analysis, lung cell suspensions were stained with Mitotracker green (Invitrogen, M7514) for 30 min at 37 °C. Staining for CD4 and CD8 was performed for the last 5 min of the incubation at 37 °C, and cells were immediately washed for flow cytometry analysis. Data were collected on an LSR II flow cytometer (BD Biosciences) or Aurora (Cytek) and analysed using FlowJo software v.10. CD4 and CD8 cell populations were well defined (Extended Data Fig. 7h). For cell sorting of DCCs, lung cell homogenates were obtained from PBS or IAV (9 dpi)-infected MMTV-Her2 mice using a Lung Dissociation Kit Mouse according to the manufacturer's protocol (Miltenyi, 130-095-927). The single-cell suspensions were treated for red blood cell lysis. Single-cell suspensions were pre-incubated (5 min) with anti-CD16/CD32 Fc-Block (BD Biosciences, 553141) followed by staining for CD45 and HER2 and sorting using an Astrios EQ flow cytometer (Beckman Coulter). DCCs were gated on CD45$^{neg}$ HER2$^+$. Sorted DCCs were used for bulk RNA-seq (described below).

## Ex vivo analysis of lung CD8$^+$ and CD4$^+$ cells

CD8$^+$ cells were isolated from digested lungs using positive selection with CD8α (Ly-2) microbeads (Miltenyi, 130-117-044) according to the manufacturer's protocol. For CD8$^+$ cell-mediated cytotoxicity experiments, Her2 cells isolated from mammary glands of MMTV-Her2 mice and expanded in culture, or immortalized PyMT cells (MET-1) isolated from mammary glands of MMTV-PYMT mice, were plated 2 days before the killing assay. Lung CD8$^+$ cells were isolated 15 days after IAV infection from wild-type mice, MMTV-Her2 mice or MMTV-Her2 mice treated with anti-CD4 antibodies (starting the day before infection). Lung CD8$^+$ cells (pooled from 3–4 mice) were added to the cancer cell cultures at a 1:1 effector:target ratio. Then, 48 h later, co-cultures were washed

(removal of CD8$^+$ cells), trypsinized and live cancer cells were counted. Isolated lung CD8$^+$ cells were restimulated using anti-CD3/anti-CD28 coated beads[53], and 20 h later, supernatant was collected and used for detection of IFNγ by ELISA, as previously described[54], using anti-mouse IFNγ capture antibody (Biolegend, 505702) and biotinylated anti-mouse IFNγ antibody (Biolegend, 505804).

CD4$^+$ cells were isolated from IAV-infected wild-type and Her2 mice lung cell homogenate using positive selection with CD4 Microbeads (L3T4) (Miltenyi, 130-117-043) according to the manufacturer's protocol. Cell pellets were used for whole-cell lysates for western blot analysis.

## Western blot analysis

Whole-cell extracts were prepared from CD4$^+$ cells isolated from the lungs of wild-type and MMTV-Her2 mice (FVB) infected with IAV, following methods described elsewhere[55]. For western blot analysis, the following antibodies were used: β-actin monoclonal antibody (AC-15) (Invitrogen, AM4302), anti-DUSP5 (Invitrogen, PA5-85961), anti-rabbit HRP (Jackson ImmunoResearch Laboratories, 111-035-144) and anti-mouse HRP (Jackson ImmunoResearch Laboratories, 115-035-166).

## Fixed single-cell RNA-seq

Single cells were generated as described in the section Flow-cytometric analyses. Cells exhibiting greater than 80% viability were fixed in a 4% formaldehyde solution using the Chromium Next GEM Single Cell Fixed RNA Sample Preparation Kit (10X Genomics). The whole-transcriptome probe pairs (10X Genomics) were added to the fixed single-cell suspensions to hybridize to their complementary target RNA during an overnight incubation at 42 °C. After hybridization, unbound probes were removed by washing. The fixed and probe-hybridized single-cell suspensions were loaded onto a Chromium X (10X Genomics) microfluidics instrument to generate partitioned nanolitre-scale droplets in oil emulsion. The target was for each droplet to contain a barcoded gel bead, a single cell and enzyme Master Mix (10X Genomics) for probe pair ligation and gel bead primer barcode extension. The droplets in oil emulsion were placed in a thermal cycler for 60 min at 25 °C, 45 min at 60 °C and 20 min at 80 °C. The single-cell barcoded, ligated probe products underwent library preparation using standard 10X Genomics protocols in preparation for Illumina next-generation sequencing. The gene expression library derived from single-cell barcoded, ligated probe product were sequenced as paired-end 150-base pair reads on an Illumina NovaSeq 6000 (Illumina) at the University of Colorado Genomics Shared Resource at a target depth of 20,000 reads per cell for all samples.

## Data processing for scRNA-seq analysis

The scRNA-seq fastq files were processed using Cell Ranger software (v.7.1.0, 10X Genomics)[56] to assign reads to genes based on Cell Ranger's Chromium mouse transcriptome probe set (v.1.0.1). The counts were analysed using the Seurat R package[57]. Genes found in fewer than 10 cells were excluded. Cells were excluded if they contained fewer than 201 genes, more than 7,500 unique molecular identifiers (UMIs) or greater than 2.5% of mitochondrial UMIs. The R package scDblFinder[58] was used to identify and subsequently remove doublets from the data. As well as removing cells identified as doublets, preliminary clustering was used in sequential fashion to remove clusters with greater than 50% of cells being identified as doublets. After downstream processing, clusters were filtered if they contained canonical markers from multiple cell types. The data were then depth-normalized followed by natural-log transformation. The top 2,000 most variable genes were used to scale the data while regressing out cell cycle S/G2M difference, total UMI and percentage of mitochondrial UMIs.

Principal component analysis was performed using the top 2,000 variable genes. Principal components ($n = 30$) that captured most of the variation were then included in further data-processing steps.

Clusters were identified (at a resolution of 1.5) using the K-nearest neighbours algorithm. Clusters were annotated to cell types using enriched canonical markers and ORA[59] with gene sets from the MSigDB[60] and the PanglaoDB[61]. Broad T lymphocytes were identified and subclustered separately to increase cell-type resolution. Differentially expressed genes were identified using the Wilcoxon rank sum test within each of the cell types identified for the indicated comparisons. GSEA was done using the clusterProfiler R package (v.4.0.5)[62] and the Benjamini–Hochberg method was used to calculate the adjusted $P$ values. ORA[62] was performed on the top 200 differentially expressed genes using the Hallmark, KEGG and GO Biological Processes gene set collections of the MSigDB[60]. Plots were produced using the Seurat[57], ggplot2[63], ggpubr[64] and pheatmap[65] R packages.

### Mitochondrial-specific scRNA-seq analysis
We analysed the $\log_2$(fold change), adjusted $P$ values and raw $P$ values generated from scRNA-seq data to compare the following experimental groups: HER2 + IAV versus HER2 + PBS, HER2 + IAV versus wild type + IAV, and HER2 + IAV +anti-CD4 versus HER2 + IAV. To focus on mitochondrial functions, we used our custom mitochondrial pathway gene lists, originally published in ref. 66. Specifically, we examined overlaps between mitochondrial OXPHOS genes and our curated innate immune pathways associated with mitochondrial activity. The results were visualized as heatmaps using the pheatmap package (v.1.0.12). Pathway analysis was done using fast GSEA[67] with custom gene-set files previously curated in ref. 66. All samples were compared with controls, and the ranked list of genes was defined using the $-\log_{10}(P\text{value}) \times \log_2(\text{fold change})$. Statistical significance was assessed through 1,000 permutations of the gene sets[60]. Results are reported with a false discovery rate (FDR) threshold of less than 0.25 and visualized as heatmaps generated with the pheatmap package (v.1.0.12).

### RNA-seq analysis of DCCs
RNA was isolated from sorted DCCs (flow sorting described above) with the RNeasy plus micro kit (Qiagen, 74034) and libraries were prepared using the SMART-Seq mRNA LP kit (Takara Bio, 634762) following the manufacturer's instructions. Pooled libraries were sequenced on the NovaSeq X (Illumina). The fastq files were processed using the nf-core rnaseq pipeline (v.3.12.0)[68]. Reads were trimmed with Cutadapt[69] and aligned to the mouse transcriptome (GRCm38, Ensembl release 102) using STAR (v.2.7.9a)[70] and quantified using Salmon (v.1.10.1)[71]. Differential expression analysis was done using limma[72] with the voom method followed by GSEA as described above.

### Influenza virus RNA quantification
Whole lung tissue was homogenized and RNA was isolated using TRIzol/chloroform extraction following the manufacturer's protocol (ThermoFisher and MilliporeSigma, respectively). RNA (1 μg) was reverse transcribed with an iScript cDNA synthesis kit (Bio-Rad Laboratories) and the viral load was determined by qPCR for the PR8 acid polymerase gene compared with a standard curve of known PR8 acid polymerase gene copy numbers as previously described[73].

### HER2+ mammospheres and EO771 organoid culture
FvB-MMTV-Her2/Neu female mice 14–18 weeks old were used as early ('premalignant') stage mice. Mice were euthanized using isoflurane and cervical dislocation. Whole mammary glands were minced and digested in 0.15% Collagenase 1A (Sigma, C-9891), 2.5% bovine serum albumin and 200 U DNAse I (Stemcell Technologies, NC9007308) solution at 37 °C with agitation for 30 min. Red blood cell lysis buffer (eBioscience, 4333-57) was used for 2 min at room temperature to remove blood cells. Cells were filtered through a 40-μm filter. Then $3 \times 10^5$ cells per well were seeded in six-well ultralow-adhesion plates in 1 ml mammosphere media (DMEM/F12 (Gibco, 11320-082), 1× B27 supplement (Gibco, 17504-044), 10 ng ml$^{-1}$ EGF (Peprotech, AF-100-15-A),

50 U penicillin-streptomycin (Thermo Fisher, 15070-063)). An additional 1 ml of mammosphere medium was added 24 h after seeding. At day 4 after seeding, cells were treated with either PBS or 10 ng ml$^{-1}$ IL-6 (R&D Systems, 406-ML-005) for 3 consecutive days. Using a Nikon Eclipse Ti-S microscope, mammospheres were imaged at 4× magnification with two images taken per well at the end of the treatment. The size and number of mammospheres were analysed using QuPath software.

EO771 cells were seeded in a poly-HEMA-coated 12-well low-adhesion culture dish at a density of $1.5 \times 10^5$ cells per well in 1 ml organoid medium (DMEM/F12, 5% FBS, 1% penicillin-streptomycin 5,000 U ml$^{-1}$, 20 ng ml$^{-1}$ FGF2, 10 ng ml$^{-1}$ EGF, 5 μM Y-27632, 4 μg ml$^{-1}$ heparin plus 5% Matrigel). Cells were treated with either PBS or 10 ng ml$^{-1}$ IL-6 (R&D Systems, 406-ML-005) for 3 days. Using an EVOS M7000 microscope, EO771 organoids were imaged at 4× magnification with five images taken per well every other day. The size and number of organoids were analysed using FIJI (ImageJ).

### Measuring transgenic *Her2* mRNA in leukocyte-depleted peripheral blood
MMTV-Her2 mice 12–14 weeks old were infected with 500 EIU Puerto Rico A/PR/8/34 H1N1 IAV or PBS as described above and euthanized using $CO_2$ at 9 dpi. Blood was collected by intracardiac puncture and placed in heparin solution on ice. Following red blood cell lysis, lineage depletion was performed using a Miltenyi Direct Lineage cell depletion kit, mouse (Miltenyi, 130-110-470) following the manufacturer's instructions. After lineage depletion, RNA was extracted using an RNeasy Plus Micro Kit (Qiagen, 74034) following the manufacturer's instructions. Quantitative PCR with reverse transcription (RT–qPCR) was done using the iTaq Universal SYBR Green One-Step RT–qPCR (Bio-Rad, 172-5150) with primers for the MMTV-Her2 rat transgene; forward, 5′-CCCGAGTGTCAGCCTCAAA-3′; reverse, 5′-GCAGGCTGCACACT GATCA-3′. The RT–qPCR was run on a Bio-Rad thermocycler (CFX Opus 384).

### Quantification and statistical analyses (mouse models)
Statistical analyses were done using Prism 10.2.1 software (GraphPad). Investigators were not blinded to allocation during virus (IAV or SARS-CoV-2) inoculation or antibody treatment. Quantification and image analysis were done in a blinded manner; $n$ indicates the number of mice per group. A minimum of three slides per mouse were used for image analysis. Total HER2+ cell counts (Figs. 1c and Fig. 2b), HER2+ cells and HER2+ Ki67+ cells were counted manually using ImageJ. Three lung sections at least 50 μm apart per mouse were counted and summed. We collected and analysed PBS groups at each time point; because no differences in DCC expansion or phenotype were observed at different time points, results for PBS samples were pooled. For other image quantifications, whole-lung images were divided into fields using the ImageJ grid function and 8–10 fields were selected at random per image and counted. For experiments with two groups, a two-tailed Student's $t$-test was used; for experiments with more than one group, one-way ANOVA tests were used unless otherwise stated. Data were expressed as mean ± s.d. and $P$ values ≤ 0.05 were interpreted as evidence against the null hypothesis (that is, no effect, no difference). Replicates represent different mice or different cultures, not repeated measures of the same sample. Graphs are presented as box and whiskers with dots representing individual values; the three lines represent the maximum (top line), median (middle line) and minimum (bottom line) values of the dataset.

### Human observational data
We selected SARS-CoV-2 infections as the driver virus owing to the mandatory reporting of infections and COVID-19 disease during the early stages of the pandemic, allowing the use of real-world data to test the hypothesis that respiratory viral infections promote metastatic disease. Two complementary datasets from different regions of the world were analysed: the UK Biobank, which is a population-based

study including 502,356 adult volunteers aged 40–69 years at recruitment from 2006 to 2010 (refs. 74,75), and the Flatiron Health electronic health record (EHR) database, which contains longitudinal data from about 280 US cancer clinics (around 800 sites of care) on patients with cancer and survivors[76,77].

### Population-based analyses of the UK Biobank

Study 1 was an analysis of UK Biobank data including lifestyle, anthropometric, medical history, SARS-CoV-2 testing and mortality data linked to national registries. Previous cancer diagnoses were obtained through consented linkage to the national cancer registry and SARS-CoV-2 test status through linkage to national registers. Mortality data were obtained from the national death registries (NHS Digital, NHS Central Register and National Records of Scotland). We considered all-cause mortality (including both primary and secondary causes), non-COVID-19 mortality (by excluding deaths with ICD codes U07.1 and U07.2 (ref. 78) or any death within one month of the latest recorded positive test result) and cancer mortality (considering cause of death with ICD codes listed in Extended Data Table 1).

To evaluate whether SARS-CoV-2 test positivity affected all-cause, non-COVID-19 or cancer mortality, we implemented a rigorous matching strategy. Cancer survivors with a primary cancer diagnosis at least five years before the start of the pandemic and a positive COVID-19 test result were matched to cancer survivors with negative test results with a similar risk profile.

Of the 502,356 UK biobank participants, we excluded two groups: first, those with missing information on sex, age, body mass index, ethnicity, smoking status, alcohol consumption, education, employment status, household income, self-reported comorbidities, date of SARS-CoV-2 testing when the primary cause of death was COVID-19 and cancer diagnosis date if the primary cause of death was cancer ($n = 65,245$); and second, participants without any SARS-CoV-2 PCR test record ($n = 195,559$) (Extended Data Fig. 12d).

This left 241,552 participants, of whom 48,958 had been diagnosed with cancer at the latest follow-up (18 December 2022). From this group, we excluded five groups: participants with inconsistent dates of death ($n = 8$); those diagnosed with multiple cancers ($n = 4,421$); participants with a primary cancer diagnosis after the start of the pandemic (defined as 1 January 2020; $n = 7,650$); those who tested positive for COVID-19 after the UK vaccination rollout (1 December 2020; $n = 13,274$); and participants with cancer diagnoses less than five years before the pandemic onset ($n = 9,969$); this was to ensure that participants were, in all likelihood, in full remission and thus any residual metastatic cancer cells were likely to be dormant.

After these exclusions, the final cohort included 13,636 participants, of whom 531 tested positive for SARS-CoV-2, and 13,105 who tested negative before the vaccination rollout (Extended Data Fig. 12d).

We used a non-parametric matching approach (without replacement)[79] to identify (up to) ten negative-test participants for each positive-test participant. Matching was performed in two steps. We performed an exact matching based on cancer type and sex. Then, we matched for age, ethnicity, smoking status, alcohol consumption, education, employment status, household income and cancer diagnosis date (with a maximum allowable difference in cancer diagnosis of five years) using the nearest-neighbour method, an algorithm based on propensity score matching. The resulting matched population included 487 with positive tests (that is, we could not find good matches for 44 of those with positive tests) matched to 4,350 with negative tests.

Using test positivity as the predictor, we ran a series of unconditional logistic regression models for all-cause, non-COVID-19 and cancer mortality. Models were adjusted for all matching factors to account for potential residual confounding. We also repeated the analyses for patients with cancer diagnosed at least ten years before the start of the COVID-19 pandemic in the United Kingdom to further increase the likelihood that patients were in remission. This was achieved by excluding the positive-test participants who were diagnosed with cancer between 1 January 2010 and 31 December 2019 and re-running the matching procedure, resulting in 266 with positive tests and 2,228 matched individuals with negative tests.

Sensitivity analyses were conducted by varying censoring dates in six-month intervals from 1 June 2020 to 31 December 2022. Longer follow-up periods included more events, whereas shorter periods minimized potential bias from missing infection data and vaccination.

### Flatiron Health EHR-based analyses

**Data source.** Study 2 used Flatiron Health's nationwide EHR-derived database, including de-identified data from about 280 US cancer clinics (around 800 sites of care). The database is longitudinal, comprising de-identified patient-level structured and unstructured data, curated by technology-enabled abstraction[76,77]. Most patients in the database originate from community oncology settings, although the community and academic proportions may vary, based on study cohorts. The data were subject to obligations to prevent re-identification and protect patient confidentiality. Institutional Review Board approval of the protocol was obtained before the study was done and included an informed-consent waiver.

Included in our study were women aged at least 18 years old at the time of initial cancer diagnosis, and who had:

i. early breast cancer; the cohort includes a probabilistic sample of patients with a diagnosis of stage I–III breast cancer on or after 1 January 2011, including those who presented with non-metastatic disease but who subsequently developed recurrent or progressive disease, with at least two visits occurring on or after January 1, 2011;

ii. metastatic breast cancer; the cohort includes a probabilistic sample of patients diagnosed with stage IV breast cancer on or after 1 January 2011 and those who presented with earlier-stage breast cancer but who subsequently developed metastatic disease on or after 1 January 2011, and who had at least two clinic encounters evident in the database occurring on or after 1 January 2011; and

iii. adult female patients aged 18 years or more at the initial diagnosis.

**Real-world data source.** The index date was defined as the date of the initial diagnosis of breast cancer. The COVID-19 status was defined as positive if any COVID-19 diagnosis (ICD codes B97.29, B97.21, J12.81, B34.2 and U07.1) was made after the index date and before the diagnosis of lung metastases or the last follow-up date. The data cut-off date was 31 August 2023. The start date of COVID-19 positivity status was the earliest COVID-19 diagnosis date. Baseline characteristics of gender, race, ethnicity and age at index date were obtained from structured data.

**Analyses.** Baseline characteristics were summarized using descriptive statistics. Cause-specific analysis was conducted (death was censored). Univariable and multivariable Cox proportional hazard models were used to evaluate the effect of COVID-19 diagnosis on the risk of metastasis to the lungs, in which COVID-19 diagnosis status was treated as a time-varying covariate. The multivariable model was adjusted for patient characteristics considered relevant, including age, race and ethnicity. There were 36,216 COVID-19-negative patients and 532 COVID-19-positive patients (all 532 COVID-19-positive patients were COVID-19 negative at the index date) included in the multivariate analysis (Extended Data Fig. 12e). The median follow-up, the corresponding interquartile range (IQR) and the total number of accumulated person-years, for all patients, were 4.36 years, 6.21 years and 277,788 person-years, respectively. The median follow-up, the corresponding IQR and the total number of accumulated person-years, for patients' COVID-negative period, were 4.35 years, 6.21 years and 277,115 person-years, respectively. The median follow-up, the corresponding IQR and the total number of accumulated person-years, for patients' COVID-positive period, were 0.98 years, 1.08 years and 673 person years, respectively. The unadjusted and adjusted hazard ratio with the corresponding two-sided 95% confidence interval

was reported. The two-sided likelihood ratio tests were conducted. The significance level was 0.05. The time to metastases to the lungs was defined as the time from the index date to the date of metastases to the lungs. Patients without a date of pulmonary metastases were censored at the last confirmed activity date or death. Last confirmed activity was defined as the latest date of vitals record, medication administration or reported laboratory tests or results. We performed additional multivariate analyses (MVA) to control for additional potential confounding factors, including comorbidities and breast cancer subtypes as sensitivity analysis. Comorbidity scores using the Elixhauser comorbidity index were computed using ICD-9-CM or ICD-10 codes as previously reported[80]. The diagnosis codes were included if the diagnosis dates were on or within 365 days after the initial diagnosis date. Cancer subgroups were based on the most recent test results recorded in the Flatiron Health database and the subgroups were defined as follows:

**Triple-negative.** Evidence of an ER-negative, progesterone receptor-negative and HER2-negative test result, in which HER2-negative is defined as negative with the cancer type not otherwise specified (NOS), next-generation sequencing (NGS) negative (ERBB2 not amplified), fluorescence in situ hybridization (FISH) negative/not amplified, IHC negative (0–1+) or IHC equivocal (2+);

**HER2+.** Defined as one or more of the following: positive NOS, IHC positive (3+), FISH positive/amplified, NGS positive (ERBB2 amplification);

**ER+.** ER-positive and/or progesterone receptor-positive test result(s).

A stratified Cox proportional hazard model with stratification factors stage, year of diagnosis, age group and cancer subgroup was used to evaluate the effect of COVID-19 diagnosis on the risk of metastases to the lungs while adjusting important covariates (age, race, ethnicity and comorbidity) at initial diagnosis. There were 23,876 COVID-19-negative patients and 359 COVID-19-positive patients included in this multivariate analysis. The adjusted hazard ratio with the corresponding two-sided 95% confidence interval was reported. The assumption of proportionality was assessed using the method outlined in ref. 81, indicating that there was no statistically significant evidence suggesting a violation of the proportional hazard assumption.

### Reporting summary

Further information on research design is available in the Nature Portfolio Reporting Summary linked to this article.

## Data availability

Raw and processed scRNA-seq data have been deposited in the Gene Expression Omnibus (GSE264175). For RNA-seq of DCCs, raw and processed RNA-seq data have been deposited in the Gene Expression Omnibus (GSE282438). The bulk RNA-seq was aligned to Ensembl GRCm38, release 102, and the scRNA-seq was processed using the Cell Ranger Chromium mouse transcriptome probe set (v.1.0.1). Data availability for the UK Biobank and Flatiron Health analyses is described in the sections above. This study used the UK Biobank data under application number 69328 to M.C.-H. The UK Biobank received ethical approval from the North West Multi-centre Research Ethics Committee (REC reference11:/NW/0382) (http://www.ukbiobank.ac.uk/ethics/). The UK Biobank data are accessible on approval from the UK Biobank access committee. Preprocessing and recoding and analytical scripts are available on request to allow the replication of findings by researchers with active UK Biobank access. Data that support the findings of this study were originated by and are the property of Flatiron Health, which has restrictions prohibiting the authors from making the dataset publicly available. Requests for data sharing by licence or by permission for the specific purpose of replicating results in this manuscript can be submitted to PublicationsDataAccess@flatiron.com. The UK Biobank and Flatiron Health statistical analyses were conducted using R v.4.1.0 (ref. 82). All other data are available from the corresponding author upon reasonable request. Source data are provided with this paper.

## Code availability

The code used for the scRNA-seq analysis is available without restrictions on GitHub at https://github.com/Aeg22/dcc_flu. Source data are provided with this paper.

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

**Acknowledgements** We acknowledge funding from grant 1 I01 BX004495 from the Department of Veterans Affairs to J.D.; the Courtenay C. and Lucy Patten Davis Endowed Chair in Lung Cancer Research to J.D.; the Kay Sutherland and Monika Weber Research Fund to J.D.; and an Across the Finish Line grant from the University of Colorado School of Medicine to J.D., M.R. and T.E.M. M.R. is supported by a National Institutes of Health (NIH)/National Cancer Institute (NCI) award R01CA260909. J.D., M.R. and J.A.A.-G. are supported by NIH/NCI award R01CA301643. J.M.B. is supported by R01AG081226, R01AI173305 and P30 AG067988. This work was also supported by NIH/NCI awards R01CA109182 and P30CA013330, the Samuel Waxman Cancer Research Foundation Tumor Dormancy Program to J.A.A.-G., who is also a Samuel Waxman Cancer Research Foundation Investigator. S.B.C. was supported by grant AWD-232405-SC from the Cancer League of Colorado. B.J.J. was supported by T32 NIGMS/NIH 5T32GM141742-02. A.N.C. is supported by a NIGMS/NIH predoctoral fellowship (T32AR079114). F.G. acknowledges funding from the MRC Centre for Global Infectious Disease Analysis (reference MR/X020258/1), funded by the UK Medical Research Council (MRC); this UK-funded award is carried out in the frame of the Global Health EDCTP3 Joint Undertaking. H.M. is supported by the UK National Institute for Health Research's Comprehensive Biomedical Centre at University College London Hospitals. M.P. was funded by the Cancer Research Institute Irvington Postdoctoral Fellowship (CRI5290). D.C.W. was supported by NIH grants R01CA259635 and R01AG078814, DOD grant W81XWH-21-1-0128 and the Bill & Melinda Gates Foundation grant INV-046722. We thank the Rocky Mountain Regional VAMC Flow Core; the Biostatistics and Bioinformatics, Genomics, Human Immune Monitoring, and Flow Cytometry Shared Resources supported by National Cancer Institute grant P30CA046934 to the University of Colorado Cancer Center; and the Gates Institute Histology Core at the University of Colorado. This work was supported by the Alpine HPC system, which is jointly funded by the University of Colorado Boulder, the University of Colorado Anschutz, Colorado State University and the National Science Foundation (award 2201538). We thank G. Geno, W. Wang and M. Wang and J. Figura for guiding our studies with the Flatiron Health database; C. Swanton for connecting us with the team that did the UK Biobank analyses; and D. Merrick and P. Jedlicka for advice on the evaluation of DCC pathology.

**Author contributions** S.B.C. and B.J.J.: conceptualization of the study; designed, performed and analysed experiments; compiled data and graphed figures; and initial draft and editing of the manuscript. J.H. and D.G.: analysed Flatiron Health data for patients with breast cancer and generated relevant figures; and edited the manuscript. R.V., M.C.-H., F.G. and H.M.: analysed UK Biobank data for patients with cancer and generated relevant figures; and edited the manuscript. B.J.J., M.P.B., V.S. and A.G.: processing and analysing scRNA-seq data and generating the relevant figure; and edited the manuscript. B.D.: did mouse experiments involving SARS-CoV-2 MA10 infection; and performed the SARS-CoV-2 MA10 plaque assay. M.D.D.: isolated RNA from flow-sorted HER2$^+$ cells and prepared the library for the bulk RNA-seq run. F.V.-P.: analysed experiments pertaining to MMTV-PyMT and assisted in tissue processing for flow cytometry; and did the isolation of DCC for RNA-seq, ex vivo CD8$^+$ T cell cytotoxic assay, mitochondrial mass analyses and western blot for Dusp5. V.Z.: did all the influenza PR8 infections and intraperitoneal injections of antibodies. W.E.S.: performed and analysed some MMTV-PyMT experiments. A.N.C. and J.M.B.: processed influenza-infected lungs and calculated the influenza viral load. L.P.-O.: performed experiments to determine the response of early lesion mammospheres to IL-6. M.P.: provided protocols and expertise for the immunofluorescence stains. A.B.: analyses of scRNA-seq data with generation of figures showing differential expression across cell types for innate immune and mitochondrial metabolism genes. S.B.B.: experimental guidance, particularly for scRNA-seq results. J.W.G. and D.C.W.: interpreted scRNA-seq differential expression results for mitochondrial and oxidative phosphorylation-related genes and wrote Supplementary Note 1. A.G. and J.C.C.: scRNA-seq data processing and analyses; and edited the manuscript. T.E.M.: provided expertise on and supervised the SARS-CoV-2 MA10-related work and provided the SARS-CoV-2 virus; edited the manuscript. J.A.A.-G.: directed experimental design and provided mouse strains and protocols; edited the manuscript; and supervised the studies. M.R.: conceptualization, design and analysis of experiments; writing and editing of the manuscript; and supervised the studies. J.D.: overall lead of the studies; initial conceptualization of the studies; designed and analysed experiments; wrote and edited the manuscript; and supervised the studies.

**Competing interests** J.A.A.-G. is a co-founder, advisory board member and equity holder in HiberCell, a Mount Sinai spin-off that develops cancer recurrence-prevention therapies. He consults for HiberCell and Astrin Biosciences, serves as chief mission advisor for the Samuel Waxman Cancer Research Foundation and has ownership interest in patent number WO2019191115A1/ EP-3775171-B1. J.D. and M.R. are on the scientific advisory board for Mitotherapeutix. J.C.C. is a cofounder and chief scientific officer of OncoRx Insights. M.C.-H. holds shares in the O-SMOSE company and has no conflict of interest to disclose; consulting activities conducted by the company are independent of the present work. H.M. has consulted for Astra Zeneca relating to the use of monoclonal antibodies in the prevention and treatment of SARS-CoV-2 infection. All other authors declare no competing interests.

**Additional information**
**Correspondence and requests for materials** should be addressed to James DeGregori.

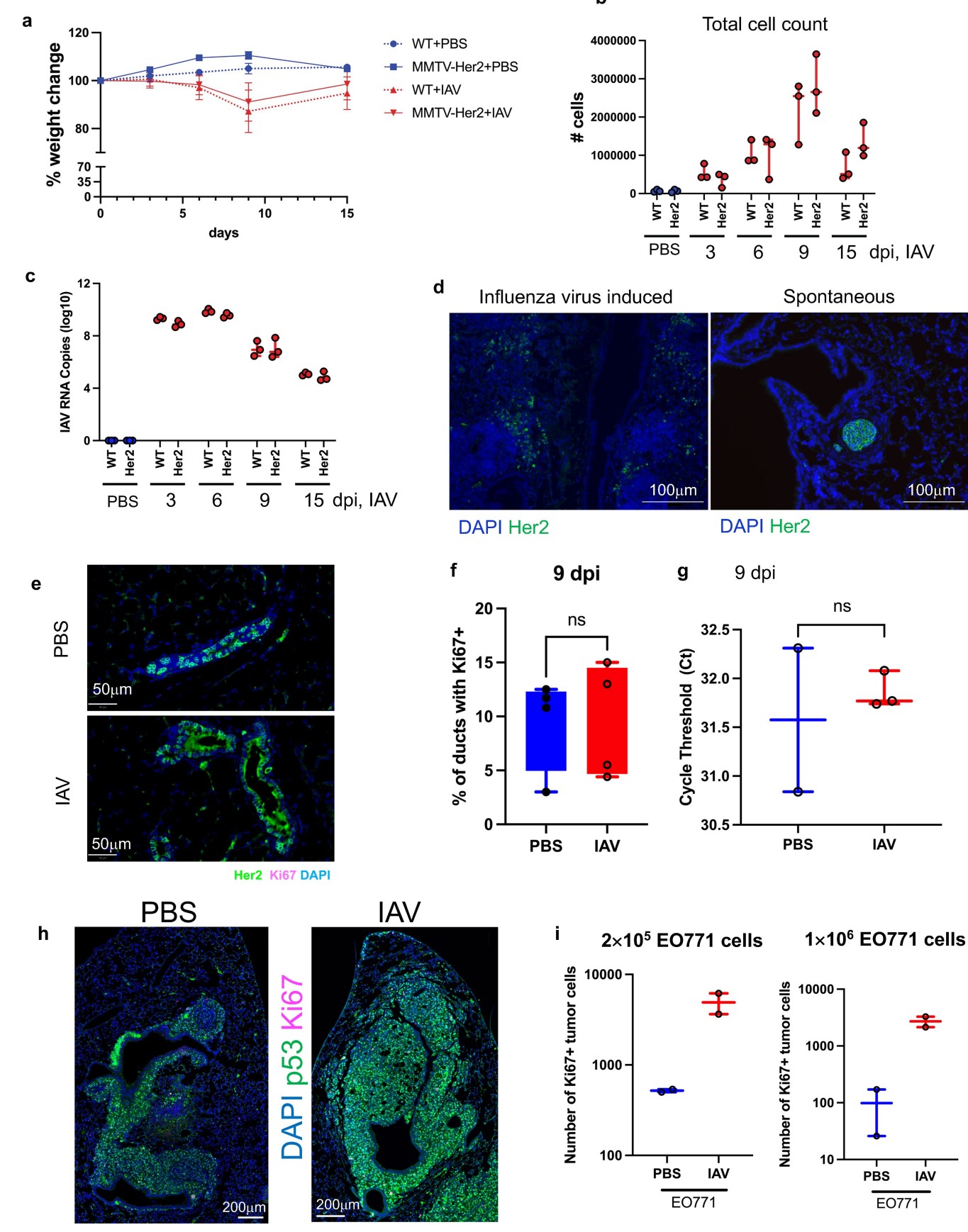

**Extended Data Fig. 1** | See next page for caption.

**Extended Data Fig. 1 | Comparison of MMTV-Her2 and WT response to IAV infection and analysis of mammary glands, circulating blood.** Weight change of MMTV-Her2 and WT mice post-IAV infection, error bars denote standard deviation (a) (n = 3/group). Total cell counts in BALF 3, 6, 9, 15 dpi (b) (n = 3/group). IAV lung viral load at 3, 6, 9, 15 dpi or for vehicle control mice (c) (n = 3/group). There were no significant differences between BALF cell counts or IAV viral load between MMTV-Her2 and WT mice. Lungs of an 18-week-old mouse 28 dpi with IAV and a naïve 9-month-old mouse stain with Her2 (green) (d). IF detection of Her2 and Ki67 in mammary glands (e) and quantification of Ki67$^+$ ducts 9 dpi (f) (n = 4/group). $C_t$ values from real-time RT-PCR for transgenic rat Erbb2 mRNA in CD45$^{neg}$ cells in circulating blood from PBS or IAV infected MMTV-Her2 mice at 9 dpi (g). IF for p53 and Ki67 (with DAPI) (h) and quantification (i) for Ki67 of EO771 tumor lesions ($2\times10^5$ EO771 cells implanted – left; $1\times10^6$ EO771 cells implanted - right) (n = 2/group). Significance is determined by one-way ANOVA test. All box-and-whisker plots are presented as maximum value (top line), median value (middle line), minimum value (bottom line) with all data points shown (dots).

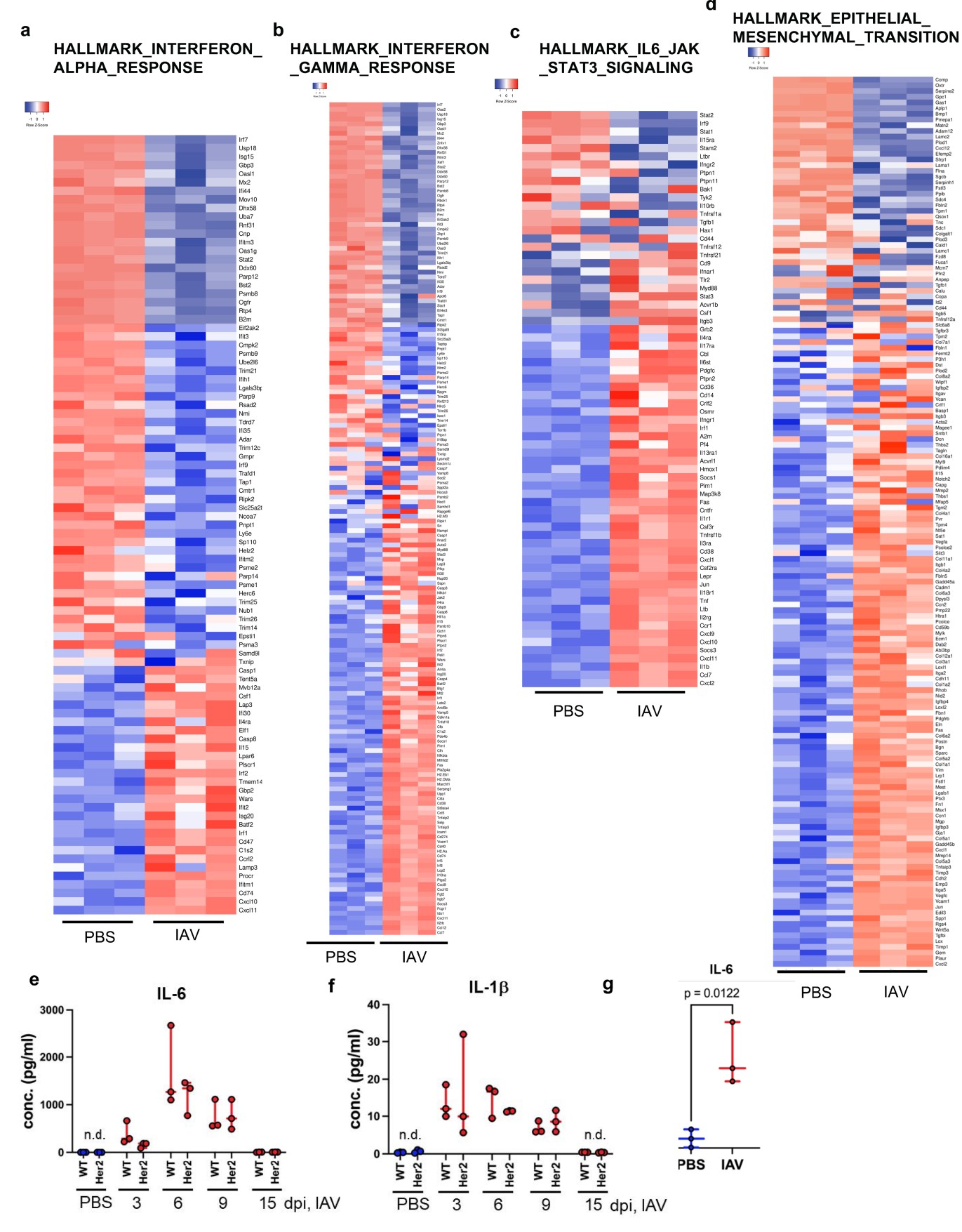

**Extended Data Fig. 2** | See next page for caption.

**Extended Data Fig. 2 | Inflammatory and EMT gene expression changes in DCCs 9 dpi and IL-6 and IL-1β concentration post IAV infection.** Heatmaps of genes for Interferon Alpha Response (a), Interferon Gamma Response (b), IL6/JAK/STAT3 Signaling (c), and Epithelial Mesenchymal Transition (d) pathways from RNA-seq for DCC comparing PBS to IAV at 9 dpi. Concentrations of IL-6 and IL-1β in BAL from MMTV-Her2 mice treated with PBS or infected with IAV at 3, 6, 9, 15 dpi (e, f). Concentration of IL-6 in supernatant of mouse tracheal epithelial cells (MTECs) treated with PBS or 24 h post IAV infection (g) (n = 3/group). Significance is determined by two tailed Student's t test. All box-and-whisker plots are presented as maximum value (top line), median value (middle line), minimum value (bottom line) with all data points shown (dots).

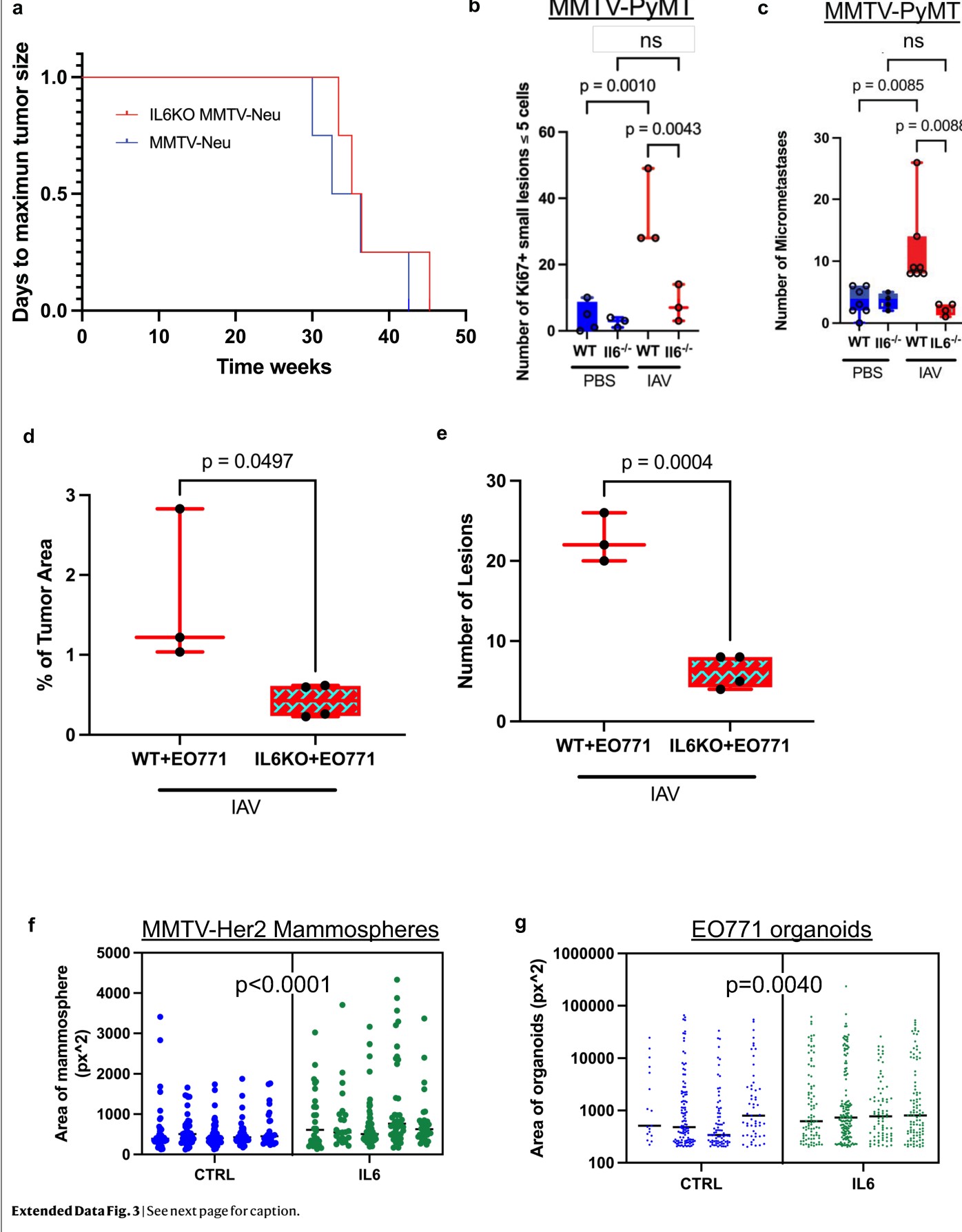

**Extended Data Fig. 3** | See next page for caption.

**Extended Data Fig. 3 | Primary tumor burden of MMTV-Her2 and IL-6 KO/ MMTV-Her2 mice and further characterization of IL6KO metastasis post IAV.** Kaplan-Meier curve for time to sacrifice due to primary tumor burden of MMTV-Her2 and IL-6 KO/MMTV-Her2 mice (n = 4/group) (a). Mice were sacrificed when primary tumors reached 2 cm in diameter; there was not a significant difference between the two groups. Quantification of PyMT+ lesions (≤ 5 cells) with at least one Ki67+PyMT+ cell in lungs 21 dpi with IAV (b) (n = 3/ group, n = 4 for PBS). PyMT+ micrometastases (defined by lesions with an area <0.03 mm$^2$) were quantified (c) (n = 7 WT, 4 IL6$^{-/-}$); the WT groups are the same as in Fig. 1f, and are included to allow comparison with the IL-6 KO. EO771 cells (2 × 10$^5$) were implanted into WT or (n = 3) IL-6 KO C57BL/6 mice (n = 4), infected with IAV, and harvested at 12 dpi, stained with H&E, and tumor area and the numbers of lesions quantified (d and e). Significance is determined by one-way ANOVA test (b,c) or two tailed Student's t test (d,e). Size of mammospheres from MMTV-Her2 mice per well 72 h post daily treatment with either PBS (CTRL) or IL-6 (10 ng/mL) (f). Size of EO771 derived organoids per well 72 h post daily treatment with either PBS (CTRL) or IL-6 (10 ng/mL) (g) (n = 5/group). Significance by Mann-Whitney tests. All box-and-whisker plots are presented as maximum value (top line), median value (middle line), minimum value (bottom line) with all data points shown (dots).

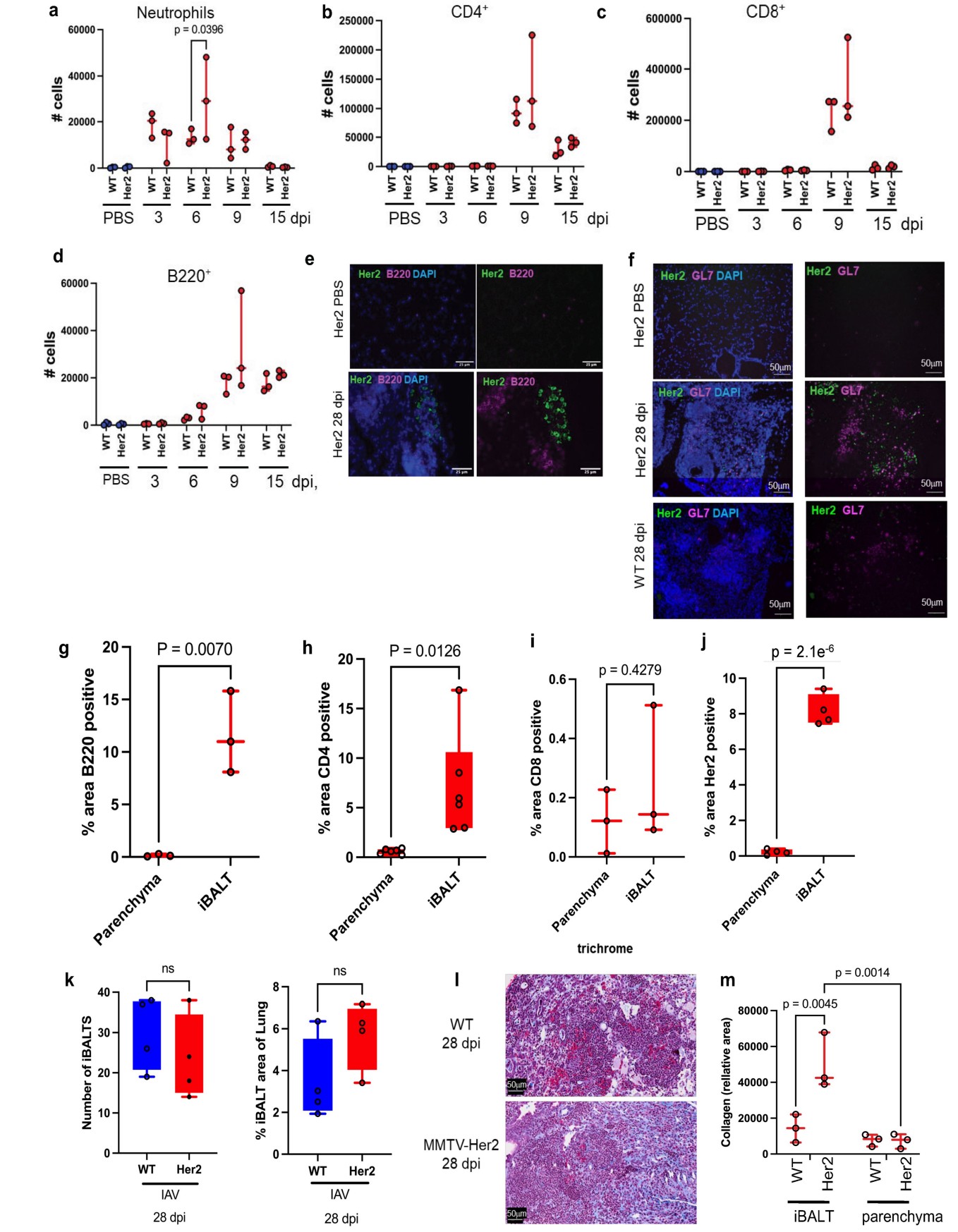

**Extended Data Fig. 4** | See next page for caption.

**Extended Data Fig. 4 | Immune infiltration and cell type and collagen analysis of iBALTs post IAV infection.** Counts of immune cells in bronchoalveolar lavage (BAL). Numbers of neutrophils, CD4$^+$, CD8$^+$, and B cells in BAL for mice treated with PBS or 3, 6, 9, 15dpi after IAV infection (a-d) (n = 3 per group. See Supplementary Fig. 3 for flow cytometry gating strategy. Representative IF stain of Her2 (green), B220 (magenta) (e), and GL7 (magenta) (f) in lungs of MMTV-Her2 mice treated with PBS or 28dpi post IAV infection (lung sections from 3 mice per group). Percentages of B220$^+$ area (g) (n = 3/group), CD4$^+$ area (h) (n = 6/group), CD8$^+$ area (i) (n = 3/group), and Her2$^+$ area of parenchyma and iBALT (j) (n = 4/group) at 28 dpi in lungs of IAV-infected mice. Quantitation of the number of iBALT and the fraction of lung occupied at 28 dpi (k) (n = 4/group). Significance is determined by two-way ANOVA test. Representative Masson's Trichrome stain of lungs 28 dpi (lung sections from 3 mice per group) (l) and quantification (m) (n = 3/group) of the positively-stained area (blue) in parenchyma and iBALT for IAV infected WT and MMTV-Her2 mice. Significance is determined by two-way ANOVA test. All box-and-whisker plots are presented as maximum value (top line), median value (middle line), minimum value (bottom line) with all data points shown (dots).

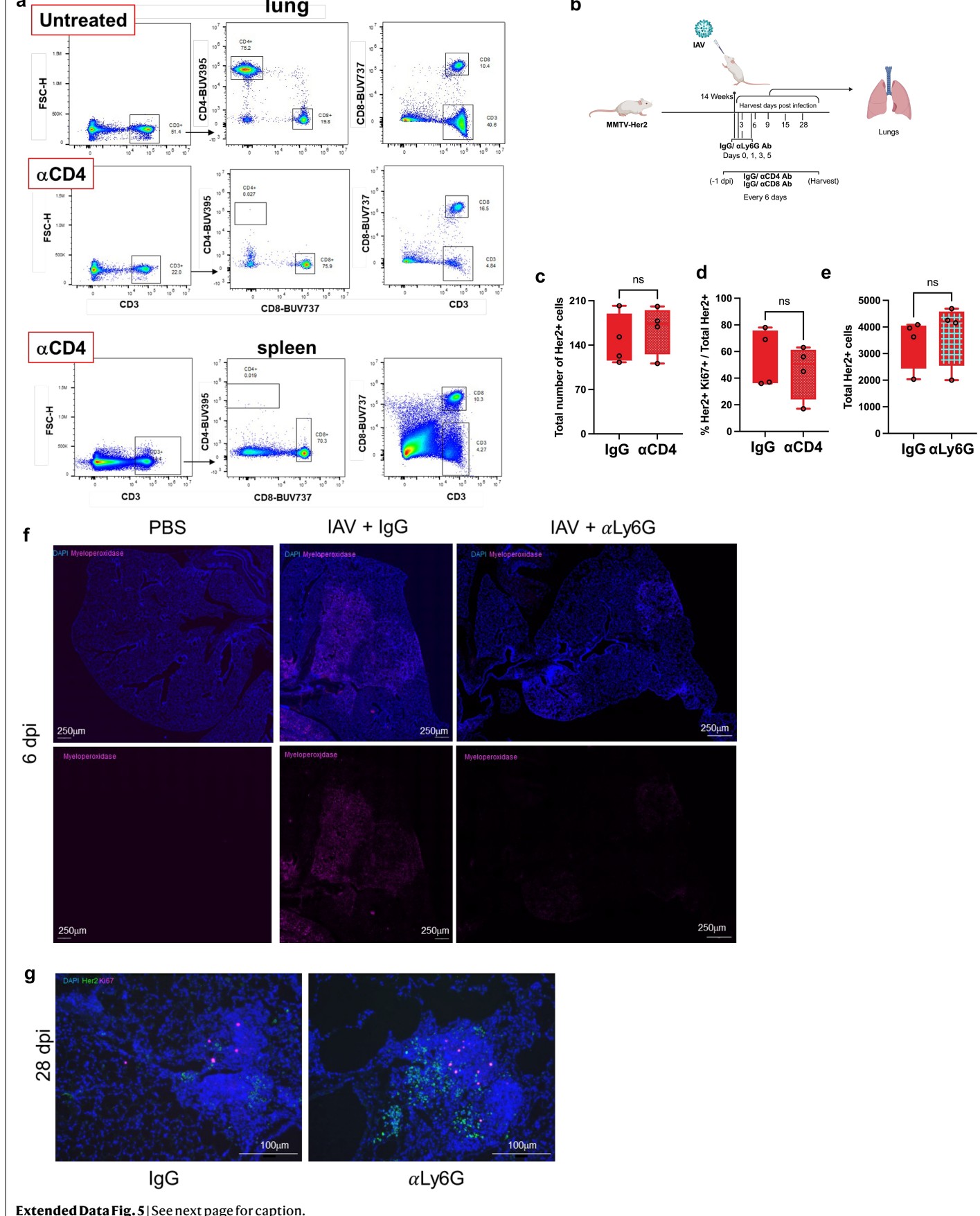

**Extended Data Fig. 5** | See next page for caption.

**Extended Data Fig. 5 | Validation of CD4 and Ly6G depletion.** Flow cytometric analysis of CD3$^+$, CD4$^+$ and CD8$^+$ cells in lungs and spleens of WT mice 6 days post injection with 100 µg αCD4 antibody clone GK1.5 (a). The third column shows that the few CD3$^+$ cells that are negative for CD8 persist following αCD4 treatment (demonstrating that the antibody was not simply preventing detection of CD4) (a). Experimental design for antibody-mediated depletion of CD4$^+$ cells, CD8$^+$ cells, or neutrophils (b). Quantification of Her2$^+$ and Ki67$^+$/Her2$^+$ cells in lungs of MMTV-Her2 (IgG) and CD4-depleted MMTV-Her2 mice 9 dpi with IAV (c, d) (n = 4/group). For e-g, mice were injected with anti-Ly6G to deplete neutrophils or IgG control. Quantification of Her2$^+$ cells 28 dpi with IAV is shown in (e) (n = 4/group). IF stains for myeloperoxidase (magenta) (f; 6 dpi) and Her2 (g; 28 dpi). Representative images show neutrophil (myeloperoxidase$^+$) depletion (f) and maintenance of Her2$^+$ cells despite this depletion (g). Significance is determined by two-tail Student's t test. All box-and-whisker plots are presented as maximum value (top line), median value (middle line), minimum value (bottom line) with all data points shown (dots). Illustration in b created using BioRender (De Dominici, M., https://BioRender.com/i40c047; 2025).

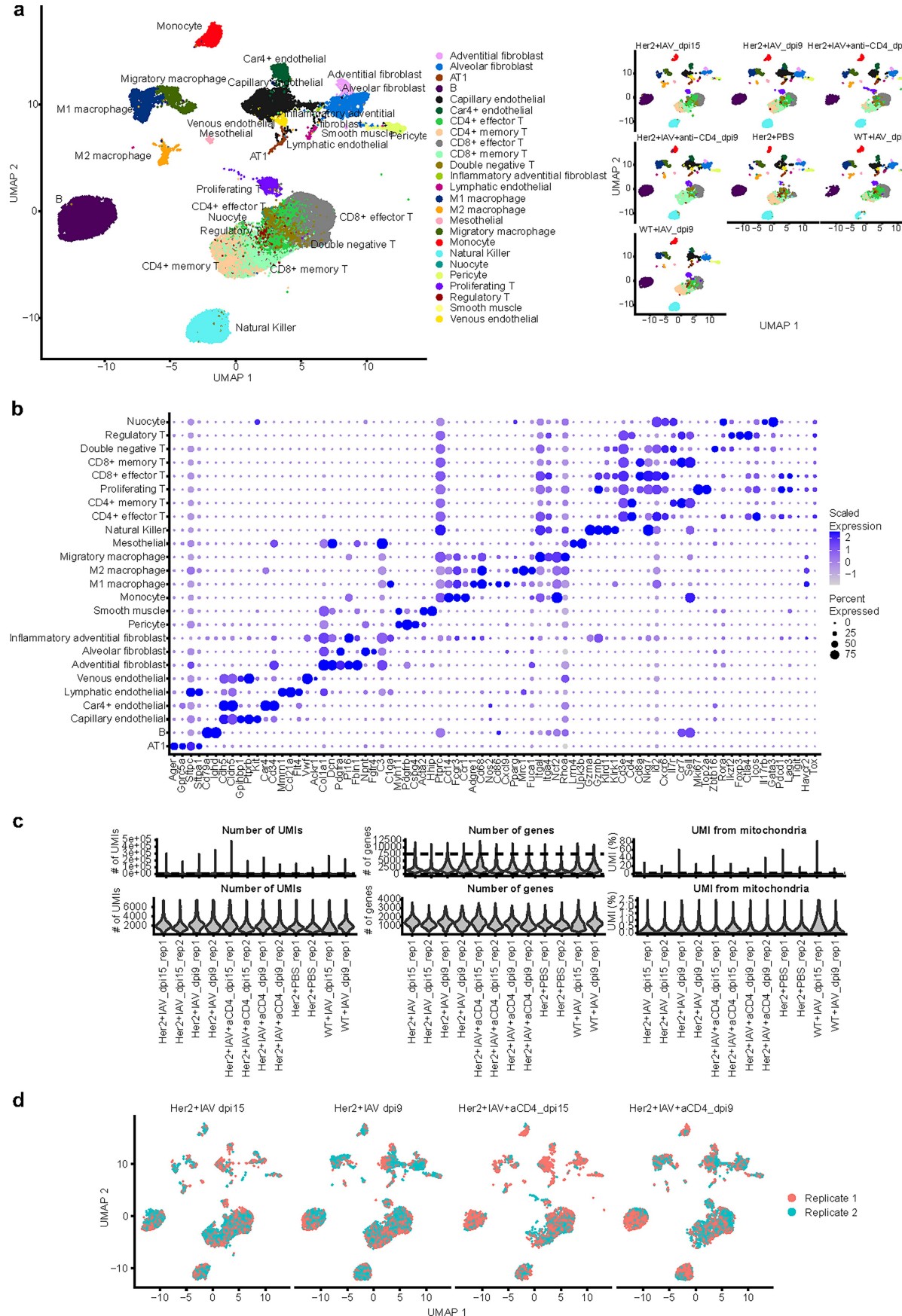

**Extended Data Fig. 6** | See next page for caption.

**Extended Data Fig. 6 | scRNAseq cell type markers and quality control.** UMAP plot labelled by cell type (a, left). UMAP plots of each experimental group (a, right). Dot plot showing the expression of canonical marker genes used to identify the cell types (b). Quality control of the scRNAseq data set: distribution of UMIs, genes and mitochondrial transcripts within each sample pre-filtering (upper panel) and post-filtering (lower panel) with dashed lines indicating the filtering thresholds (c). Mice at 9 and 15 dpi with the replicates within each group highlighted (demonstrating reproducible patterns) (d).

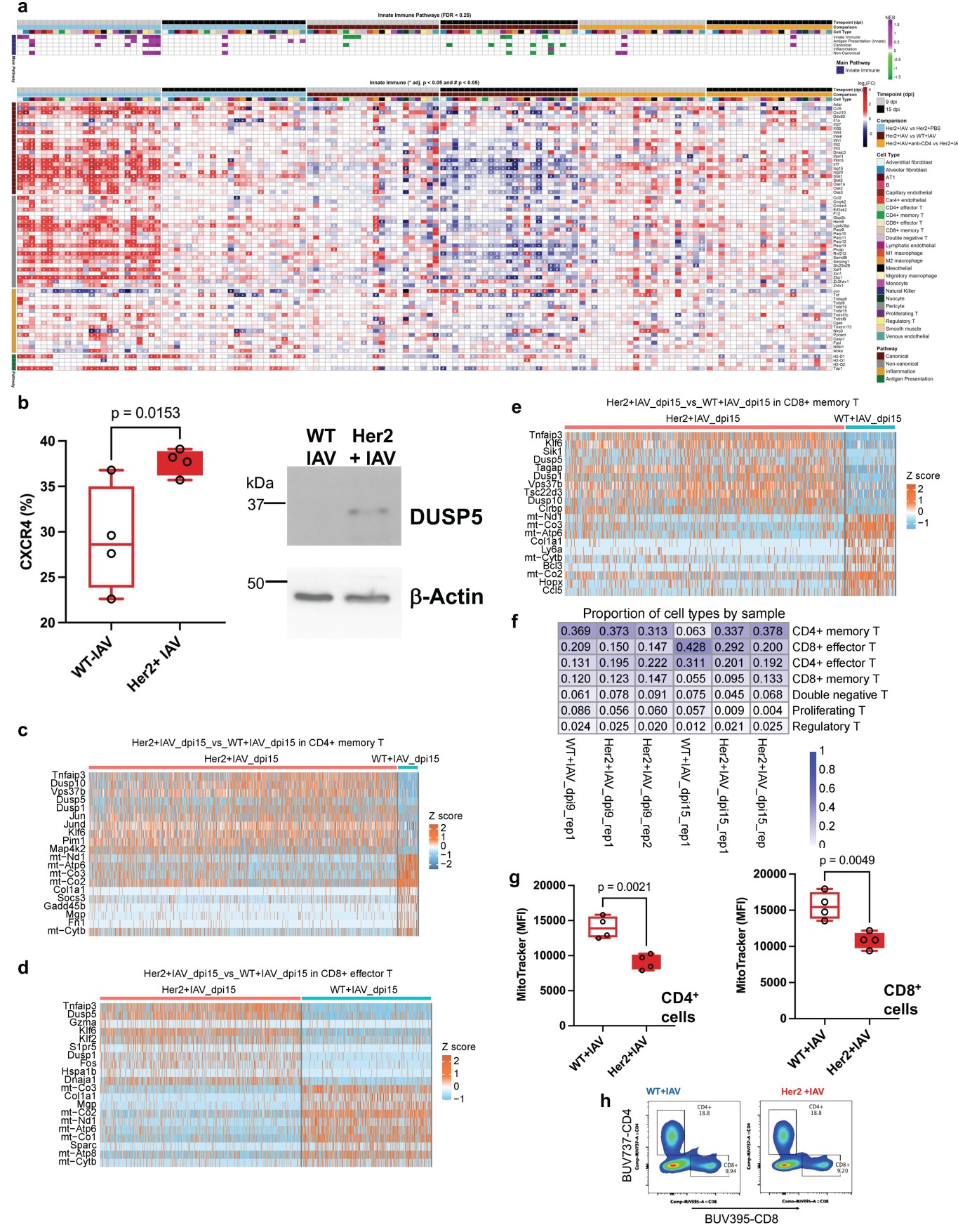

**Extended Data Fig. 7** | See next page for caption.

**Extended Data Fig. 7 | Mitochondrial OXPHOS pathway changes across all cell types and analysis of T-cells following IAV infection with or without DCCs or CD4 depletion.** The top heatmap in (a) displays statistically significant changes in custom mitochondrial OXPHOS pathways, ranked by normalized enrichment score (NES) and identified through fGSEA analysis. Only pathways with a false discovery rate (FDR) < 0.25 are shown. The bottom heatmap shows individual $\log_2$(Fold-Change) values for custom innate immune genes across the experimental groups: Her2+IAV vs. HER2 + PBS, Her2+IAV vs. WT + IAV, and Her2+IAV+anti-CD4 vs. Her2+IAV. All genes are included, with statistical significance marked by * for adjusted p-value < 0.05 and # for raw p-value < 0.05. Note that for groups with depletion of CD4$^+$ cells using anti-CD4 antibody, the residual cells in CD4$^+$ effector, CD4$^+$ memory and regulatory T-cell clusters expressed minimal CD4, and thus are not analyzed. Flow cytometric detection of Cxcr4 and western blotting for Dusp5 protein (b) (n = 4/group). Heatmap of top 20 differentially expressed genes from scRNAseq comparing CD4$^+$ memory T-cells (c), CD8$^+$ effector T-cells (d), CD8$^+$ memory T-cells (e) in MMTV-Her2+IAV versus WT + IAV mice at 15dpi. Proportion of T-cell subtypes identified in scRNAseq (f). Mean fluorescence intensity of MitoTracker stain in CD4$^+$ and CD8$^+$ cells (g) (n = 4/group) from lungs of 15 dpi IAV infected WT or MMTV-Her2 mice. Significance is determined by two tailed Student's t test. CD4 and CD8 cell populations used to gate for MitoTracker staining (h) (n = 4/group). All box-and-whisker plots are presented as maximum value (top line), median value (middle line), minimum value (bottom line) with all data points shown (dots).

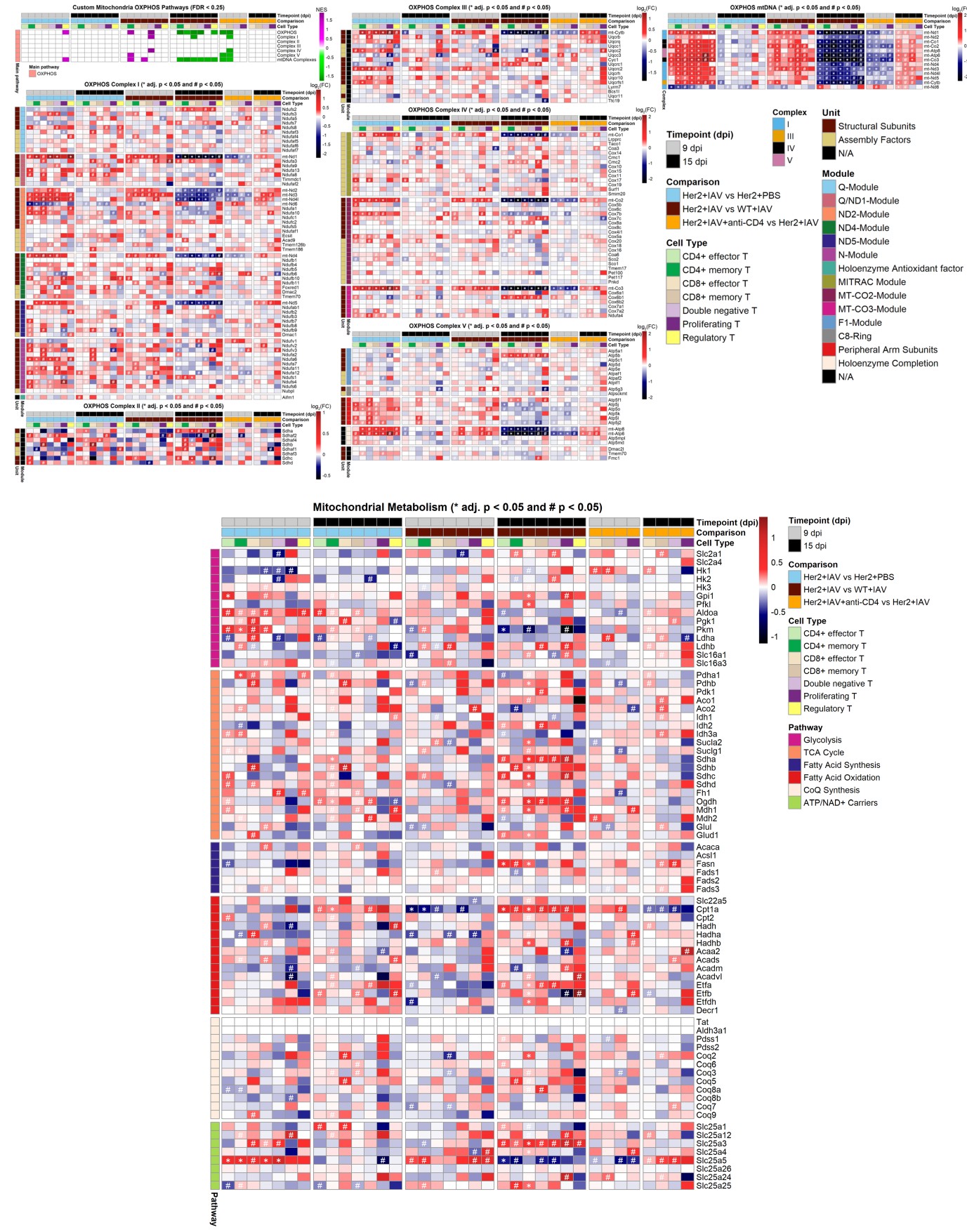

**Extended Data Fig. 8** | See next page for caption.

**Extended Data Fig. 8 | Mitochondrial OXPHOS pathway changes across T-cells following IAV infection with or without DCCs or CD4 depletion.** The top-left heatmap illustrates statistically significant changes in custom mitochondrial OXPHOS pathways, ranked by normalized enrichment score (NES) and determined through fGSEA analysis. Only pathways with a false discovery rate (FDR) < 0.25 are displayed. The remaining heatmaps depict individual log$_2$(Fold-Change) values for mitochondrial OXPHOS complex genes, comparing the experimental groups: Her2+IAV vs. HER2 + PBS, Her2+IAV vs. WT + IAV, and Her2+IAV+anti-CD4 vs. Her2+IAV. All genes are displayed, with statistical significance indicated by * for adjusted p-value < 0.05 and # for raw p-value < 0.05. For groups with depletion of CD4$^+$ cells using anti-CD4 antibody, the residual cells in CD4$^+$ effector, CD4$^+$ memory and regulatory T-cell clusters exhibited minimal detection of CD4, and thus are not analyzed.

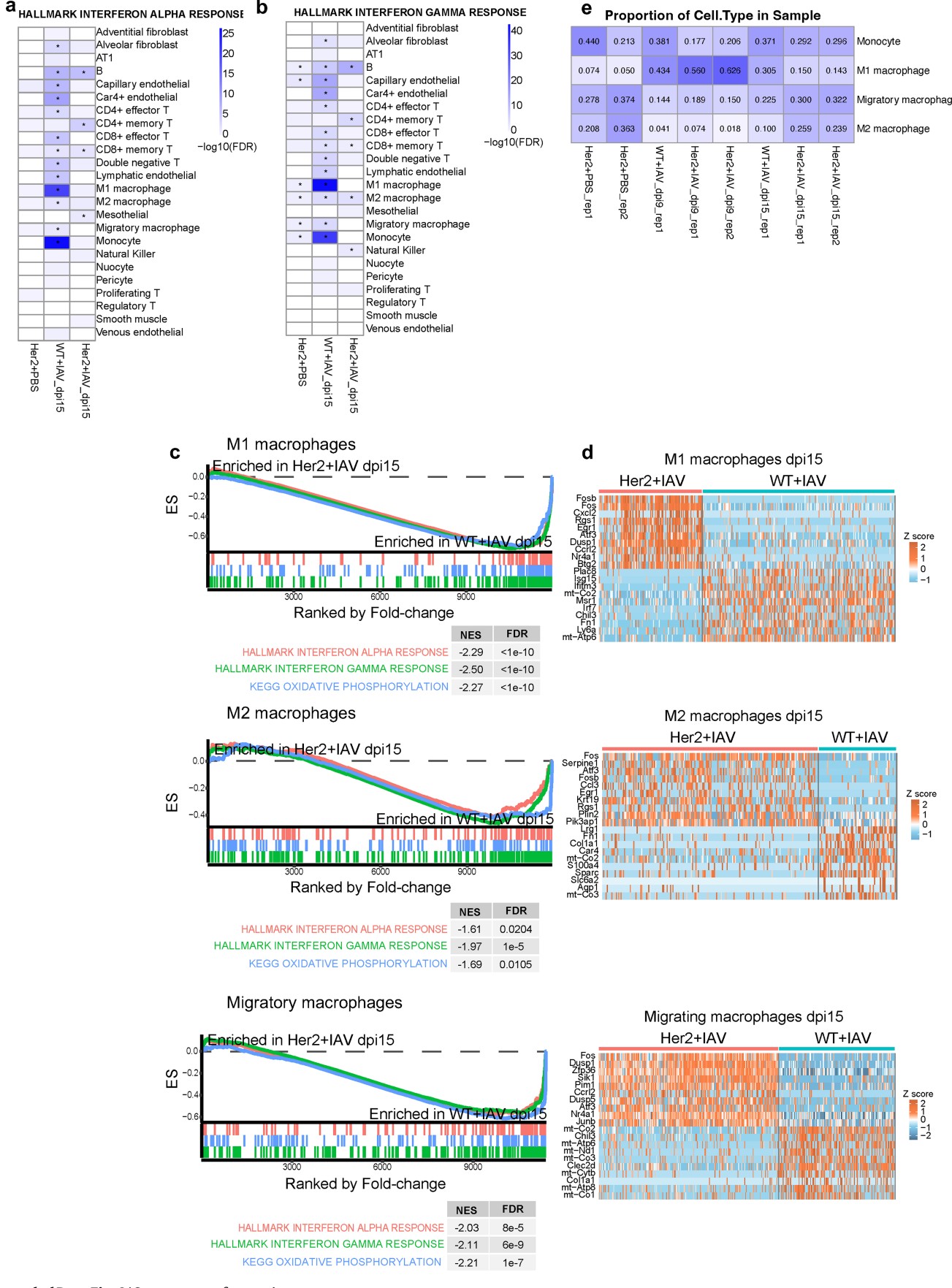

**Extended Data Fig. 9** | See next page for caption.

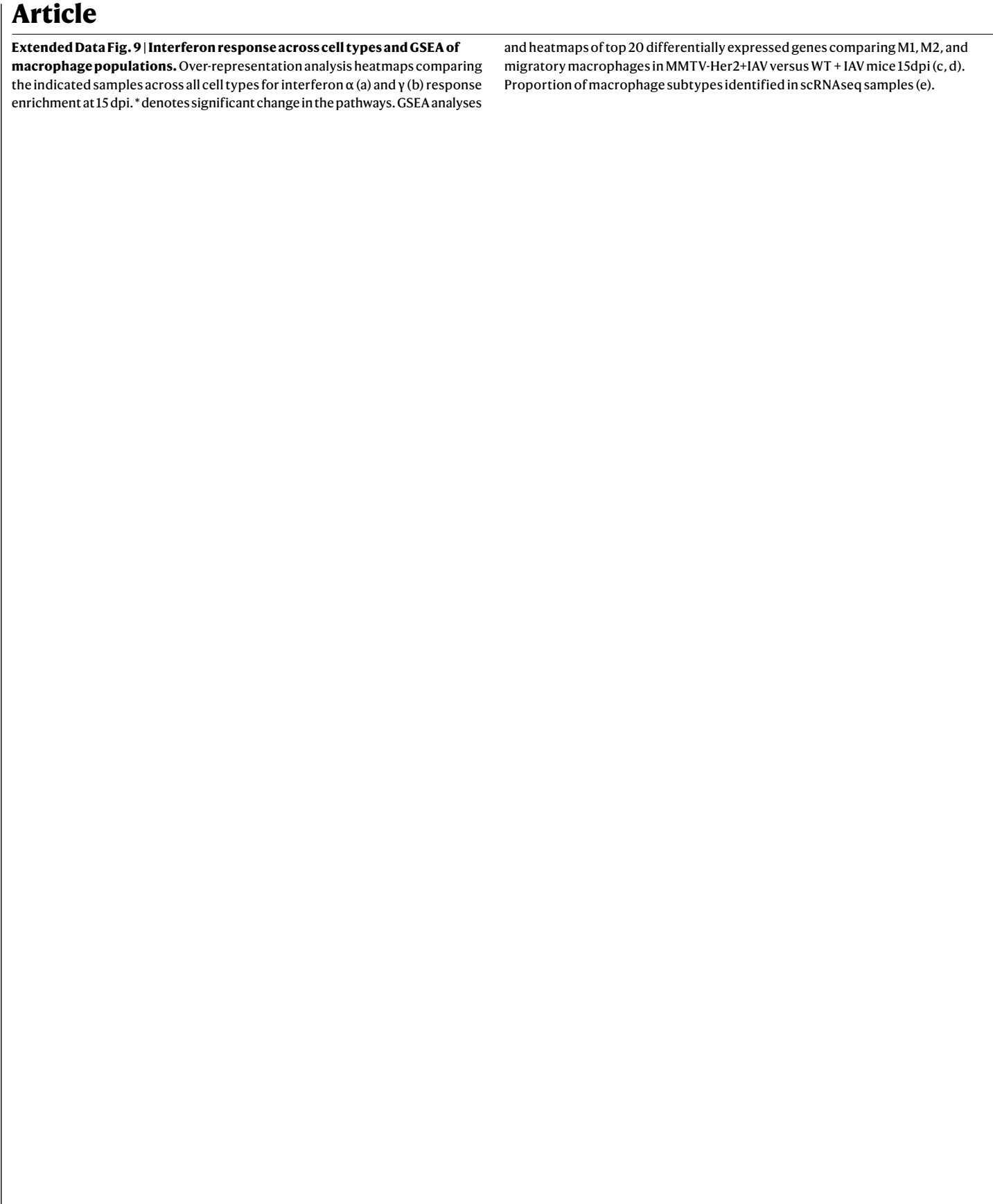

**Extended Data Fig. 9 | Interferon response across cell types and GSEA of macrophage populations.** Over-representation analysis heatmaps comparing the indicated samples across all cell types for interferon α (a) and γ (b) response enrichment at 15 dpi. * denotes significant change in the pathways. GSEA analyses and heatmaps of top 20 differentially expressed genes comparing M1, M2, and migratory macrophages in MMTV-Her2+IAV versus WT + IAV mice 15dpi (c, d). Proportion of macrophage subtypes identified in scRNAseq samples (e).

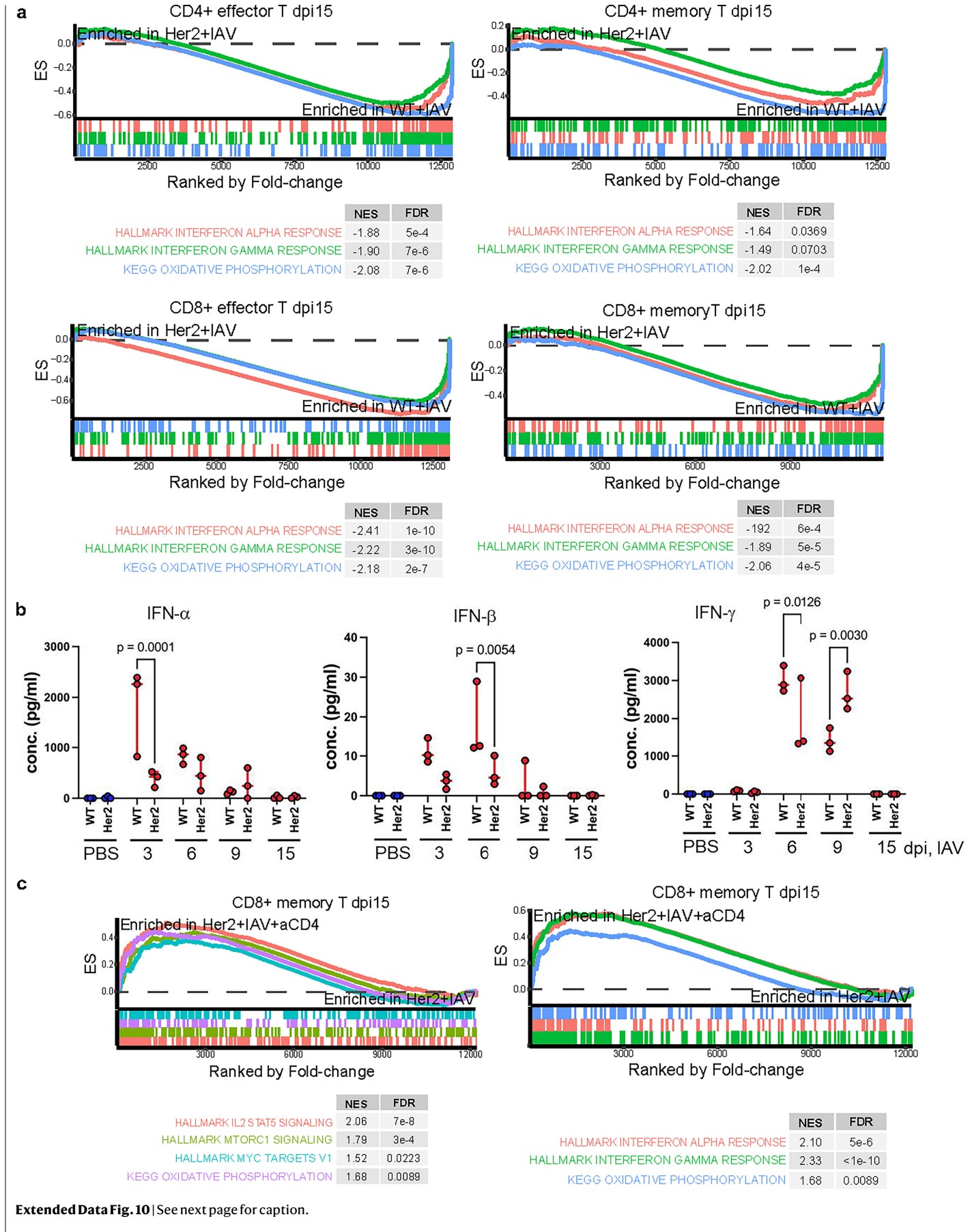

**Extended Data Fig. 10** | See next page for caption.

**Extended Data Fig. 10 | Comparison of interferon concentrations and GSEA analysis CD4 and CD8 T-cells post IAV with or without DCCs.** GSEA analyses comparing CD4[+] effector T-cells, CD4[+] memory T-cells, CD8[+] effector T-cells, and CD8[+] memory T-cells in MMTV-Her2+IAV versus WT + IAV mice at 15dpi (a). Concentrations of IFN-α, IFN-β, and IFN-γ in BAL at the indicated times post-IAV infection (b) (n = 3/group). Significance is determined by two-way ANOVA test (b). GSEA analyses of CD8[+] memory T-cells in CD4-depleted MMTV-Her2+IAV versus MMTV-Her2+IAV mice at 15 dpi (c). All box-and-whisker plots are presented as maximum value (top line), median value (middle line), minimum value (bottom line) with all data points shown (dots).

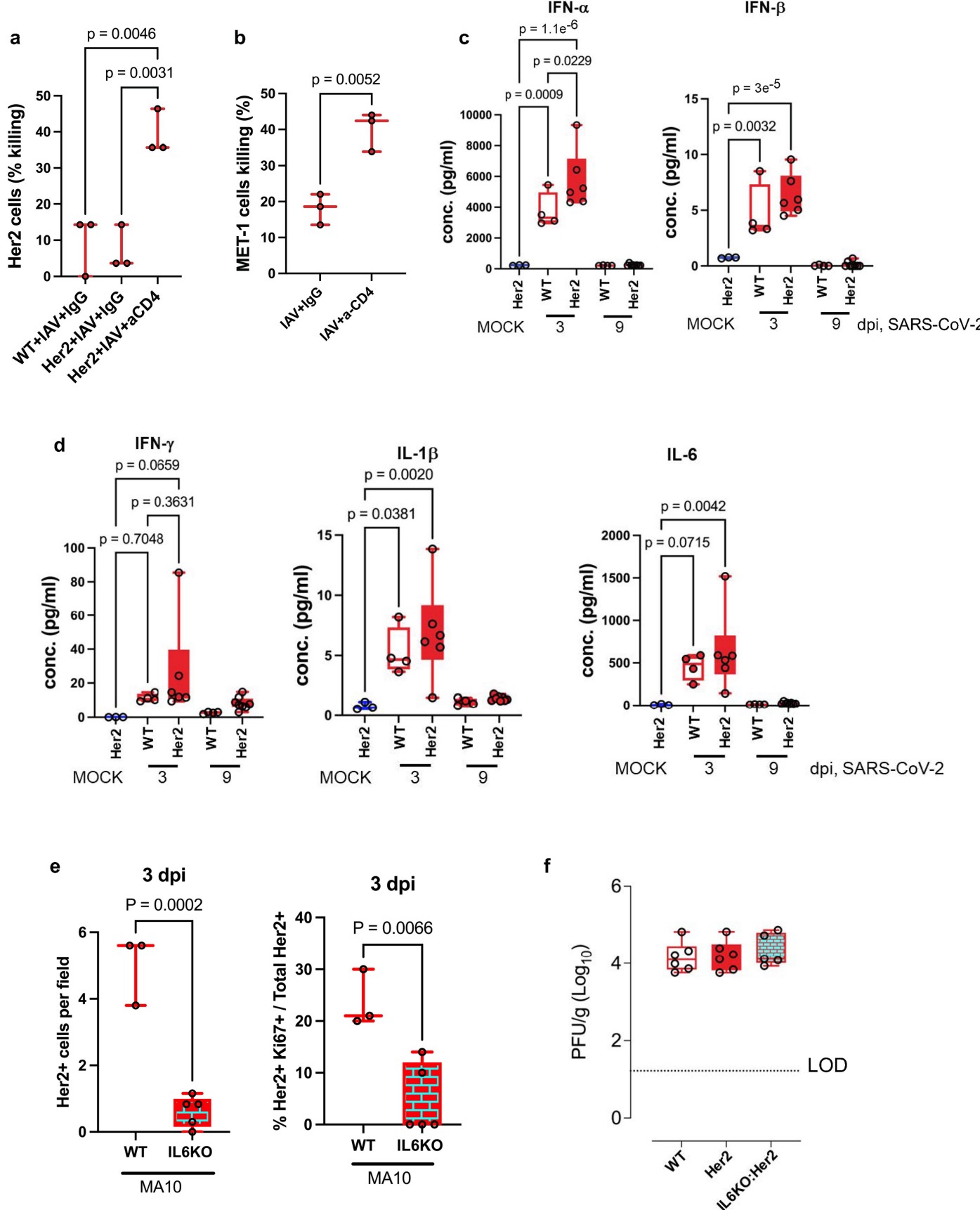

**Extended Data Fig. 11 |** See next page for caption.

**Extended Data Fig. 11 | CD8+ cytotoxic assay and analysis of SARS-CoV-2 MA10 infection.** Ex vivo CD8+ cytotoxic assay where Her2+ mammary tumor cells were incubated with CD8+ cells enriched from lungs of WT, MMTV-Her2 mice, and MMTV-Her2 mice with CD4 depletion at 15 dpi (a) (n = 3/group). Ex vivo CD8+ cytotoxic assay where PyMT-expressing MET1 cells were incubated with CD8+ cells enriched from lungs of MMTV-Her2 mice with or without CD4 depletion 15 dpi (b) (n = 3/group). Concentrations of IFN-α, IFN-β, IFN-γ, IL-1β, and IL-6 in BAL at the indicated times post-MA10 infection (c, d). Quantification of Her2+ cells and percentage of Ki67+/Her2+ cells in lungs of MMTV-Her2 mice without (WT) or with IL-6KO 3 dpi (e) (n = 3 WT, 5 IL6KO). Infectious SARS-CoV-2 MA10 burden 3dpi with MA10 in lungs as quantified by plaque assay (f) (n = 6/group). Significance is determined by one-way ANOVA test (a,c,d) or two tailed Student's t test (b,e). All box-and-whisker plots are presented as maximum value (top line), median value (middle line), minimum value (bottom line) with all data points shown (dots).

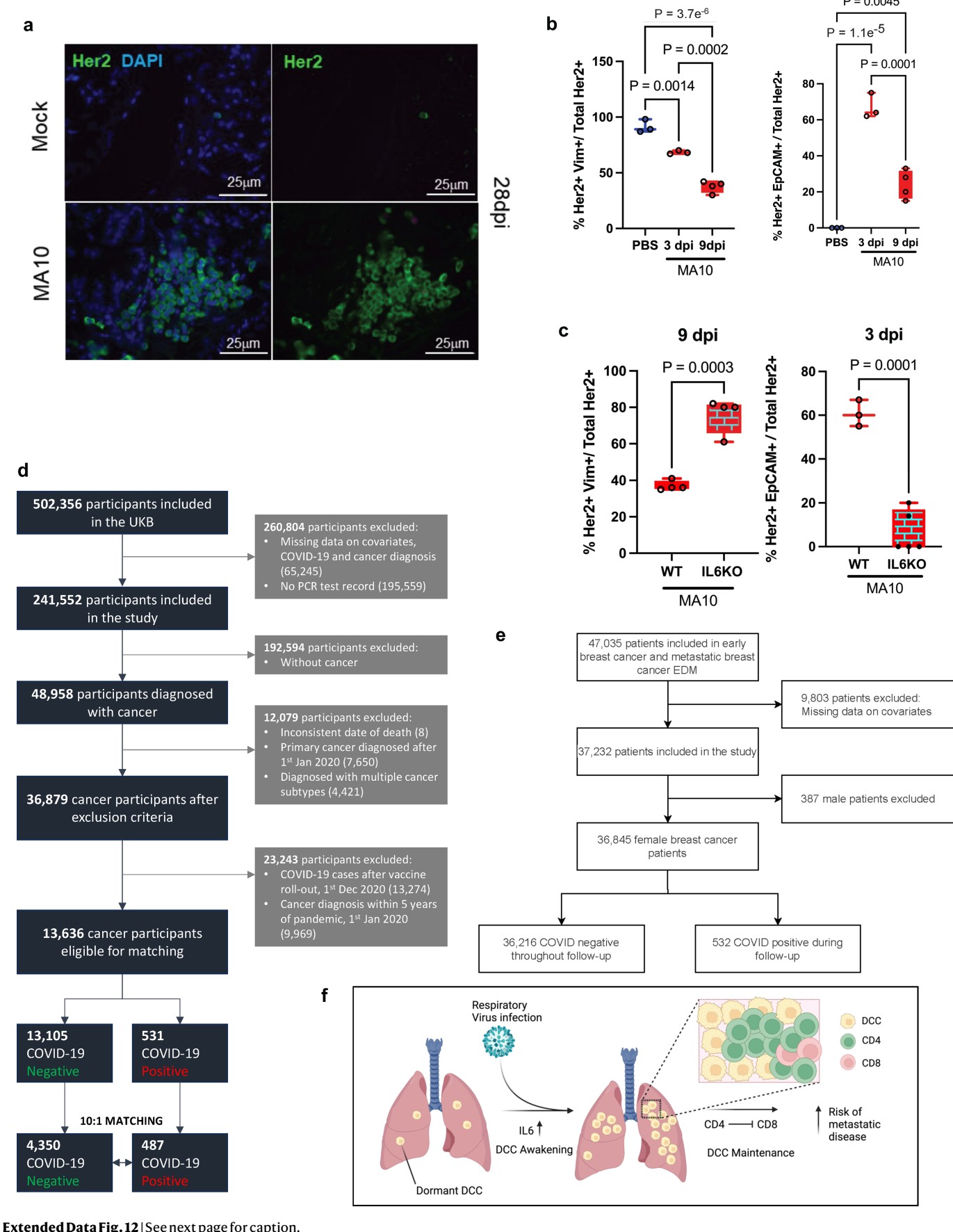

**Extended Data Fig. 12 | See next page for caption.**

**Extended Data Fig. 12 | DCC cell state changes following SARS-CoV-2 MA10 infection and summary representations of epidemiological studies.** IF stain of Her2 at 28 dpi (a) for C57BL6/J MMTV-Her2 mouse lungs at 28 dpi with MA10 SARS-CoV-2 or vehicle (mock). Quantification of vimentin⁺/Her2⁺ and EpCAM⁺/Her2⁺ cells 3- and/or 9-dpi in FVB MMTV-Her2 (b) (n = 3 PBS, 3 dpi, n = 4 at 9 dpi) and without (WT) or with IL6KO (c) (n = 4/group 9 dpi, n = 3 WT at 3 dpi, n = 5 IL6KO at 3 dpi). Significance is determined by one-way ANOVA test (b, left) or two tailed Student's t test (others). Summary representation of the selection, exclusion, and matching criteria implemented for UK Biobank (d). Summary representation of the selection, exclusion, and matching criteria implemented for the Flatiron Health Database (e). Model - pulmonary virus-dependent increases in IL-6 contribute to the awakening and expansion of dormant mesenchymal-like breast cancer cells that switch to a mixed epithelial/mesenchymal-like phenotype in lungs in the early phase of viral infection. CD4⁺ cells maintain the expanded breast cancer cells in late phase of viral infection through suppressing CD8⁺ cells. Virus-dependent awakening and expansion of DCC in the lungs increases the risks of metastatic progression (f). All box-and-whisker plots are presented as maximum value (top line), median value (middle line), minimum value (bottom line) with all data points shown (dots). Illustration in f created using BioRender (De Dominici, M., https://BioRender.com/yxpclmg; 2025).

**Extended Data Table 1 | Cancer related mortality ICD codes**

| Group | ICD10 | ICD9 |
|---|---|---|
| Malignant neoplasms of lip, oral cavity and pharynx | C00-C14 | 140-149 |
| Malignant neoplasms of digestive organs | C15-C26 | 150-159 |
| Malignant neoplasms of respiratory and intrathoracic organs | C30-C39 | 160-165 |
| Malignant neoplasms of bone and articular cartilage | C40-C41 | 170 |
| Melanoma and other malignant neoplasms of skin | C43-C44 | 172-173 |
| Malignant neoplasms of mesothelial and soft tissue | C45-C49 | 171 |
| Malignant neoplasm of breast | C50 | 174-175 |
| Malignant neoplasms of female genital organs | C51-C58 | 179-184 |
| Malignant neoplasms of male genital organs | C60-C63 | 185-187 |
| Malignant neoplasms of urinary tract | C64-C68 | 188-189 |
| Malignant neoplasms of eye, brain and other parts of central nervous system | C69-C72 | 190-192 |
| Malignant neoplasms of thyroid and other endocrine glands | C73-C75 | 193-194 |
| Malignant neoplasms, stated or presumed to be primary, of lymphoid, haematopoietic and related tissue | C81-C96 | 200-208 |

ICD codes considered to define cancer-related mortality in UK Biobank.

**Extended Data Table 2 | UK Biobank Sensitivity analyses**

| | Censoring date | Number of participants | Number of deaths | OR (95% CI) | p |
|---|---|---|---|---|---|
| All Cause mortality | 31/12/2022 | 4,837 | 413 | 4.50 (3.49, 5.81) | 6.88E-31 |
| | 01/06/2022 | | 340 | 4.90 (3.74, 6.40) | 3.49E-31 |
| | 01/12/2021 | | 276 | 6.19 (4.66, 8.21) | 2.06E-36 |
| | 01/06/2021 | | 174 | 10.74 (7.72, 14.94) | 3.21E-45 |
| | 01/12/2020 | | 116 | 14.03 (9.45, 20.84) | 3.48E-39 |
| Non Covid mortality | 31/12/2022 | 3,847 | 128 | 2.56 (1.86, 3.51) | 6.95E-09 |
| | 01/06/2022 | | 107 | 2.31 (1.64, 3.27) | 2.04E-06 |
| | 01/12/2021 | | 80 | 2.78 (1.91, 4.04) | 8.95E-08 |
| | 01/06/2021 | | 47 | 3.52 (2.25, 5.51) | 3.62E-08 |
| | 01/12/2020 | | 23 | 3.34 (1.81, 6.14) | 0.000108 |
| Cancer Mortality | 31/12/2022 | 4,225 | 293 | 1.85 (1.14, 3.02) | 0.013401 |
| | 01/06/2022 | | 244 | 1.92 (1.13, 3.25) | 0.015345 |
| | 01/12/2021 | | 185 | 2.67 (1.52, 4.68) | 0.000588 |
| | 01/06/2021 | | 105 | 4.39 (2.28, 8.46) | 9.78E-06 |
| | 01/12/2020 | | 53 | 8.24 (3.43, 19.77) | 2.31E-06 |

Sensitivity analyses comparing all-cause, non-COVID-19, and cancer mortality in cancer survivors (with cancer diagnosis > 5 years before the start of the COVID-19 pandemic) with a positive vs a negative test. We considered several censoring dates for the death events ranging from 01/12/2021 to 31/12/2022. (all dates as day/month/year).

**Extended Data Table 3 | Flatiron Health Breast Cancer Patients with COVID-19 Diagnosis**

| Variables | Total N | Missing N | levels | Total |
|---|---|---|---|---|
| Age at diagnosis | 46545 (100.0) | 0 | Median (IQR) | 60.0 (50.0 to 69.0) |
| Age at diagnosis group | 46545 (100.0) | 0 | < 65 | 29533 (63.5) |
| | | | >= 65 | 17012 (36.5) |
| Race | 41427 (89.0) | 5118 | Asian | 1120 (2.4) |
| | | | Black or African American | 5190 (11.2) |
| | | | Other Race | 4645 (10.0) |
| | | | White | 30472 (65.5) |
| | | | (Missing) | 5118 (11.0) |
| Ethnicity | 38476 (82.7) | 8069 | Hispanic or Latino | 3586 (7.7) |
| | | | Not Hispanic or Latino | 34890 (75.0) |
| | | | (Missing) | 8069 (17.3) |
| Covid Diagnosis | 46545 (100.0) | 0 | No | 45956 (98.7) |
| | | | Yes | 589 (1.3) |
| Diagnosis year | 46545 (100.0) | 0 | < 2015 | 23566 (50.6) |
| | | | >= 2015 | 22979 (49.4) |
| Commordity | 34960 (75.1) | 11585 | Median (IQR) | 0.0 (0.0 to 1.0) |
| Cancer subgroups | 40947 (88.0) | 5598 | ER+ | 27826 (59.8) |
| | | | ER+ and HER2+ | 4305 (9.2) |
| | | | HER2+ | 2189 (4.7) |
| | | | Triple-negative | 6627 (14.2) |
| | | | (Missing) | 5598 (12.0) |
| Stage: I-III vs IV | 42518 (91.3) | 4027 | I-III | 33175 (71.3) |
| | | | IV | 9343 (20.1) |
| | | | (Missing) | 4027 (8.7) |

Analysis of Flatiron Health Breast Cancer Patients with COVID-19 Diagnosis. Demographic and clinical characteristics.

**Extended Data Table 4 | Cox proportional hazard model of Flatiron Health data**

| Metastasis to the lung | | | HR (univariable) | HR (multivariable) |
|---|---|---|---|---|
| COVID-19 Diagnosis | No | 46545 (98.8) | - | - |
| | Yes | 589 (1.2) | 1.35 (0.96-19.2, p=0.087) | 1.44 (1.01-2.05, p=0.043) |
| Age at Diagnosis | Mean (SD) | 59.1 (13.1) | 1.00 (1.00-1.01, P<0.001) | 1.01 (1.00-1.01, P<0.001) |
| Race | Asian | 1131 (2.7) | - | - |
| | Black or African American | 5274 (12.5) | 1.16 (1.03-1.31, p=0.016) | 1.15 (1.00-1.31, p=0.045) |
| | Hispanic or Latino | 103 (0.2) | 1.45 (1.03-2.03, p=0.032) | 1.14 (0.80-1.63, p=0.474) |
| | Other Race | 4694 (11.2) | 0.92 (0.82-1.05, p=0.216) | 0.84 (0.73-0.97, p=0.018) |
| | White | 30896 (73.4) | 0.68 (0.61-0.76, p<0.001) | 0.67 (0.59-0.76, p<0.001) |
| Ethnicity | Hispanic or Latino | 3618 (9.3) | - | - |
| | Not Hispanic or Latino | 35395 (90.7) | 0.75 (0.70-0.80, p<0.001) | 0.76 (0.70-0.82, p<0.001) |

Cox proportional hazard model of Flatiron Health Breast Cancer Patients with COVID-19 status as time varying covariate, the number of patients included in the multivariable model is 36,748.

**Extended Data Table 5 | Stratified Cox proportional hazard model of Flatiron Health data**

| Metastasis to the lung | | | HR (univariable) | HR (multivariable) |
|---|---|---|---|---|
| Covid Diagnosis | No | 46545 (98.8) | - | - |
| | Yes | 589 (1.2) | 1.35 (0.96-1.92, p=0.087) | 1.46 (0.99-2.15, p=0.057) |
| Age at diagnosis | Mean (SD) | 59.1 (13.1) | 1.00 (1.00-1.01, p<0.001) | 1.00 (1.00-1.00, p=0.675) |
| Race | Asian | 1131 (2.7) | - | - |
| | Black or African American | 5274 (12.6) | 1.16 (1.03-1.31, p=0.016) | 1.03 (0.86-1.24, p=0.735) |
| | Other Race | 4694 (11.2) | 0.92 (0.82-1.05, p=0.216) | 0.96 (0.79-1.15, p=0.638) |
| | White | 30896 (73.6) | 0.68 (0.61-0.76, p<0.001) | 0.78 (0.65-0.92, p=0.003) |
| Ethnicity | Hispanic or Latino | 3635 (9.3) | - | - |
| | Not Hispanic or Latino | 35384 (90.7) | 0.75 (0.70-0.80, p<0.001) | 0.86 (0.77-0.95, p=0.004) |
| Commordity | Mean (SD) | 0.7 (1.1) | 1.02 (0.99-1.04, p=0.180) | 1.04 (1.01-1.07, p=0.003) |

Strata (stage, year diagnosis, age at diagnosis, cancer)

Stratified Cox proportional hazard model of Flatiron Health Breast Cancer Patients with stratification factors stage, year of diagnosis, age group, and cancer subgroup, the number of patients included in the multivariable model is 24,235. Analysis of Flatiron Health Breast Cancer Patients with COVID-19 Diagnosis. Demographic and clinical characteristics (a). Cox proportional hazard model with COVID-19 status as time varying covariate, the number of patients included in the multivariable model is 36,748 (b). Stratified Cox proportional hazard model with stratification factors stage, year of diagnosis, age group, and cancer subgroup, the number of patients included in the multivariable model is 24,235 (c).

# Reporting Summary

## Statistics

For all statistical analyses, confirm that the following items are present in the figure legend, table legend, main text, or Methods section.

| n/a | Confirmed | |
|---|---|---|
| ☐ | ☒ | The exact sample size (*n*) for each experimental group/condition, given as a discrete number and unit of measurement |
| ☐ | ☒ | A statement on whether measurements were taken from distinct samples or whether the same sample was measured repeatedly |
| ☐ | ☒ | The statistical test(s) used AND whether they are one- or two-sided *Only common tests should be described solely by name; describe more complex techniques in the Methods section.* |
| ☐ | ☒ | A description of all covariates tested |
| ☐ | ☒ | A description of any assumptions or corrections, such as tests of normality and adjustment for multiple comparisons |
| ☐ | ☒ | A full description of the statistical parameters including central tendency (e.g. means) or other basic estimates (e.g. regression coefficient) AND variation (e.g. standard deviation) or associated estimates of uncertainty (e.g. confidence intervals) |
| ☐ | ☒ | For null hypothesis testing, the test statistic (e.g. *F*, *t*, *r*) with confidence intervals, effect sizes, degrees of freedom and *P* value noted *Give P values as exact values whenever suitable.* |
| ☒ | ☐ | For Bayesian analysis, information on the choice of priors and Markov chain Monte Carlo settings |
| ☐ | ☒ | For hierarchical and complex designs, identification of the appropriate level for tests and full reporting of outcomes |
| ☐ | ☒ | Estimates of effect sizes (e.g. Cohen's *d*, Pearson's *r*), indicating how they were calculated |

*Our web collection on statistics for biologists contains articles on many of the points above.*

## Software and code

Policy information about availability of computer code

| Data collection | No software or code was used for data collection. |
|---|---|
| Data analysis | Data Processing for single cell RNA-seq analysis and bulk RNA-seq analysis is described in Methods, utilizing R (v 4.1.1) and publicly available software Cell Ranger (v 7.1.0), Seurat R package (v 4.3.0), R package scDblFinder (v 1.6.0) (single cell RNA) and limma (v 3.46.0) with the voom method (bulk RNA-seq). Gene set enrichment analysis (GSEA) was performed using the clusterProfiler R package (v 4.0.5). For mitochondrial specific analysis, custom mitochondrial pathway gene lists (Guarnieri et al. Reference provided in manuscript) were utilized, and fGSEA (needs version, don't believe this was a BBSR analysis) was used for pathway analysis using the custom pathway gene lists. For UK Biobank and Flatiron Health data, statistical analyses were conducted using R version 4.1.0. All analyses methods are described in the Methods section, and code for single cell RNA-seq is available on GitHub (https://github.com/Aeg22/dcc_flu). |

For manuscripts utilizing custom algorithms or software that are central to the research but not yet described in published literature, software must be made available to editors and reviewers. We strongly encourage code deposition in a community repository (e.g. GitHub). See the Nature Portfolio guidelines for submitting code & software for further information.

## Data

Policy information about availability of data

All manuscripts must include a data availability statement. This statement should provide the following information, where applicable:
- Accession codes, unique identifiers, or web links for publicly available datasets
- A description of any restrictions on data availability
- For clinical datasets or third party data, please ensure that the statement adheres to our policy

Data availability - UK Biobank: This study used the UK Biobank data under application number 69328 to MC-H. The UK Biobank received ethical approval from the North West Multi-centre Research Ethics Committee (REC reference: 11/NW/0382) (http://www.ukbiobank.ac.uk/ethics/). UK Biobank data is accessible upon approval from the UK Biobank access committee. Pre-processing/recoding and analytical scripts are available upon request to allow replication of findings by researchers with active UK Biobank access.

Data availability - Flatiron: The data that support the findings of this study were originated by and are the property of Flatiron Health, Inc., which has restrictions prohibiting the authors from making the data set publicly available. Requests for data sharing by license or by permission for the specific purpose of replicating results in this manuscript can be submitted to PublicationsDataAccess@flatiron.com.

Data availability - gene expression: All scRNAseq and bulk RNA-seq data were uploaded to GEO. Raw and processed scRNAseq data is deposited in the Gene Expression Omnibus (GSE264175). For RNA-seq of DCC, raw and processed RNA-seq data are deposited in the Gene Expression Omnibus (GSE282438). Both have been made publicly available, and there will be no restrictions placed on the data.

The bulk RNA-seq was aligned to Ensembl GRCm38, release 102 while the scRNAseq was processed using the Cell Ranger Chromium mouse transcriptome probe set (version 1.0.1).

## Research involving human participants, their data, or biological material

Policy information about studies with human participants or human data. See also policy information about sex, gender (identity/presentation), and sexual orientation and race, ethnicity and racism.

| | |
|---|---|
| Reporting on sex and gender | Electronic health records (EHR) data were analyzed from the UK Biobank and Flatiron Health. For Flatiron Health, as our analyses focused on breast cancer, only biological females were included. For UK Biobank data both sexes were included, of which 53.1% were female. Analyses were sex-matched. |
| Reporting on race, ethnicity, or other socially relevant groupings | Flatiron EHR data: Age, Gender, Race, Ethnicity, were from the US-based, electronic health record-derived deidentified Flatiron Health Research Database[1]<br>1. Flatiron Health. Database Characterization Guide. Flatiron.com. Published March 18, 2025. Accessed [spelled out Month Day, Year]. https://flatiron.com/database-characterization.<br>UK Biobank data included information on ethnicity, education, employment status, household income which were used for propensity score matching. |
| Population characteristics | For Flatiron Health, as our analyses focused on breast cancer, only biological females were included. The median age at diagnosis were 59 with IQR (49 to 69). 2.6% were Asian, 12.7% were Black or African American, 9.7% were Other Race, and 75% were White. 92% were Not Hispanic or Latino, and 8% were Hispanic or Latino.<br>For UK BIOBANK at recruitment, the mean age was 60 with IQR (53 - 64) with an average BMI of 27 (IQR 24-30), 53.1% were female, 22.4% had a university degree, 0.9% were Black or African American, 0.7% were Asian, 0.8% Other Race, and 97.6% were White, 43.7% had a higher income. 52.5% was employed. |
| Recruitment | No active recruitment was carried out. The epidemiological data used for analyses are extracted from existing databases. Inclusion/exclusion criteria for the subjects included in the final analyses were described in the method section. |
| Ethics oversight | For the Flatiron Health analyses, the Colorado Institutional Review Board approval of the protocol was obtained prior to study conduct and included an informed consent waiver (COMIRB#23-1485, Exemption Category 4). The UK Biobank received ethical approval from the North West Multi-centre Research Ethics Committee (REC reference: 11/NW/0382) (http://www.ukbiobank.ac.uk/ethics/). This information is provided in the Methods section of the manuscript. |

Note that full information on the approval of the study protocol must also be provided in the manuscript.

## Field-specific reporting

Please select the one below that is the best fit for your research. If you are not sure, read the appropriate sections before making your selection.

☒ Life sciences  ☐ Behavioural & social sciences  ☐ Ecological, evolutionary & environmental sciences

For a reference copy of the document with all sections, see nature.com/documents/nr-reporting-summary-flat.pdf

## Life sciences study design

All studies must disclose on these points even when the disclosure is negative.

| | |
|---|---|
| Sample size | No sample-size calculation was performed. A sample size of at least 3 samples per group were in each experiment, and each experiment was repeated at least once. Based on previous experience with the metastases models, most experiments were performed with at least samples sizes of 4, as variability in responses to infection were anticipated to necessitate more than the minimal number of 3 for calculations of |

| significance. |
| :-- |

| Data exclusions | No data were excluded from the analyses. |
| :-- | :-- |
| Replication | Each new experiment contains an experimental group for the previous experiment for confirmation of the previous result. All experiments have been successfully replicated at least once in independent experiments. |
| Randomization | Allocations of animals are randomized. For experiments other than those involving animals, samples were allocated into different treatment groups (e.g. mammospheres treated with vehicle or interleukin-6) in an unbiased fashion (e.g. half the wells get treated with IL-6 and half with vehicle). For in vitro CD8 T-cell killing assays, the different CD8 cell groups were based on the treatment of the donor mice, and then these cells were proportionally split into co-cultures with the different target cells.<br><br>For in vitro studies, Her2+ organoids were derived from mammary glands of MMTV-Her2 mice of the indicated ages that were randomly selected from our mouse colony for that age and genotype. For EO771 and MET1 cell line studies, no covariates were controlled as all cells were derived from the same source, and thus are considered relatively clonal. |
| Blinding | Investigators were not blinded in group allocations but were blinded in data collection and data analyses. |

# Reporting for specific materials, systems and methods

We require information from authors about some types of materials, experimental systems and methods used in many studies. Here, indicate whether each material, system or method listed is relevant to your study. If you are not sure if a list item applies to your research, read the appropriate section before selecting a response.

## Materials & experimental systems

| n/a | Involved in the study |
| :-- | :-- |
| ☐ | ☒ Antibodies |
| ☐ | ☒ Eukaryotic cell lines |
| ☒ | ☐ Palaeontology and archaeology |
| ☐ | ☒ Animals and other organisms |
| ☒ | ☐ Clinical data |
| ☒ | ☐ Dual use research of concern |
| ☒ | ☐ Plants |

## Methods

| n/a | Involved in the study |
| :-- | :-- |
| ☒ | ☐ ChIP-seq |
| ☐ | ☒ Flow cytometry |
| ☒ | ☐ MRI-based neuroimaging |

## Antibodies

| Antibodies used | All information relevant to antibodies are included in the Extended Data Table 1: resource table. |
| :-- | :-- |
| Validation | All antibodies were purchased from commercial sources and validated by the manufacturer, whose websites contain validation statement. Her2 antibody validation data are provided in the manuscript (see Fig 4a). We provide a table (Extended Data Table I) that contains all the antibodies used together with information regarding the manufacturer, catalog number, and dilution; validation for each antibody is described on the manufacturer's web-page. In addition antibody stains were done in parallel with secondary only and with negative controls to insure specificity. |

## Eukaryotic cell lines

Policy information about cell lines and Sex and Gender in Research

| Cell line source(s) | EO771 mammary tumor cells are of C57BL/6 origin, and were the gift of Dr. Diana Cittelly.<br>Vero C1008 (clone E6) were obtained directly from ATCC CRL-1586. |
| :-- | :-- |
| Authentication | For EO771, as a mouse line, STR methods are not available, but we confirmed the expected high expression of p53 and their C57BL/6 origin (as the cells were not rejected in immunocompetent C57BL/6 mice). For Vero cells, the ATCC authenticates all cell lines. |
| Mycoplasma contamination | EO771 and Vero cells tested negative for mycoplasma. |
| Commonly misidentified lines (See ICLAC register) | none |

## Animals and other research organisms

Policy information about studies involving animals; ARRIVE guidelines recommended for reporting animal research, and Sex and Gender in Research

| Laboratory animals | Mus musculus FVB female 12-14 weeks old; Mus musculus FVB MMTV-erbB2/neu/HER2 female 12-14 weeks old; Mus musculus FVB MMTV-erbB2/neu/Her2-IL6KO female 12-14 weeks old; Mus musculus C57BL6/J female 14-18 weeks old; Mus musculus C57BL6/J MMTV-erbB2/neu/HER2 female 14-17 weeks old; Mus musculus MMTV-PyMT female 7-9 weeks old; Mus mucsculus MMTV-PyMT-IL6KO female 7-9 weeks old. As described in Methods, all mice were co-housed in specific pathogen free animal facilities, maintained |
| :-- | :-- |

at 21 degrees C (+/- 1 degree C), 35% humidity, and a 14 hours light/10 hours dark cycle (6 am-8 pm). All the mice were backcrossed in the C57Bl/6J background for over 10-12 generations.  Only female mice were used for the studies, as we are studying metastases derived from mammary gland tumors. The average age of the mice was between 12 to 24 weeks. CO2 followed by cervical dislocation as an approved secondary method was used for euthanasia.

| | |
|---|---|
| Wild animals | The study did not involved wild animals. |
| Reporting on sex | Findings from the mouse model applies only to one sex (female). |
| Field-collected samples | No field collected samples were used in the study |
| Ethics oversight | University of Colorado Anschutz Medical Campus Institutional Animal Care and Use Committee approved all experiments. |

Note that full information on the approval of the study protocol must also be provided in the manuscript.

# Plants

| | |
|---|---|
| Seed stocks | *Report on the source of all seed stocks or other plant material used. If applicable, state the seed stock centre and catalogue number. If plant specimens were collected from the field, describe the collection location, date and sampling procedures.* |
| Novel plant genotypes | *Describe the methods by which all novel plant genotypes were produced. This includes those generated by transgenic approaches, gene editing, chemical/radiation-based mutagenesis and hybridization. For transgenic lines, describe the transformation method, the number of independent lines analyzed and the generation upon which experiments were performed. For gene-edited lines, describe the editor used, the endogenous sequence targeted for editing, the targeting guide RNA sequence (if applicable) and how the editor was applied.* |
| Authentication | *Describe any authentication procedures for each seed stock used or novel genotype generated. Describe any experiments used to assess the effect of a mutation and, where applicable, how potential secondary effects (e.g. second site T-DNA insertions, mosiacism, off-target gene editing) were examined.* |

# Flow Cytometry

## Plots

Confirm that:

☒ The axis labels state the marker and fluorochrome used (e.g. CD4-FITC).

☒ The axis scales are clearly visible. Include numbers along axes only for bottom left plot of group (a 'group' is an analysis of identical markers).

☒ All plots are contour plots with outliers or pseudocolor plots.

☒ A numerical value for number of cells or percentage (with statistics) is provided.

## Methodology

| | |
|---|---|
| Sample preparation | Cells were either recovered from the bronchoalveolar lavage fluid (BALF) by resuspending cell pellets after centrifuging BALF at 500g, 4 degrees C, 5 minutes in PBS with 2%FBS, 2mM EDTA or by digesting whole lungs in collagenase A (Sigma Aldrich cat# COLLA-RO; St. Louis, MO) and deoxyribonuclease I (Worthington cat# LS002139; Lakewood, NJ) with final cells resuspended in PBS with 2% FBS and 2mM EDTA. Cells were strained with 50um cell strainers before antibody staining. For Her2+ sorting, whole lungs were digested using Miltenyi's lung dissociation kit. |
| Instrument | BD Biosciences LSRII Flow Cytometer, Astrios EQ Flow Cytometer |
| Software | FlowJo |
| Cell population abundance | Her2+ cells were 100% within Her2+ sorted fraction as determined by the Astrios EQ Flow Cytometer. |
| Gating strategy | FSC-A/SSC-A were used to select cells, FSC-A/FSC-H were used to select single cells, live/dead and CD45 were used to select distinct CD45+ cell population. CD4 vs. CD8 axes were used to select CD4+ or CD8+ cells. B220 were used to select B cells, and Ly6G were used to select neutrophils after CD45 gating. CD44, GzmB, FoxP3 were used to select for respective cell populations. Cell types were selected where there are distinct populations (CD45, CD4, CD8, B220, Ly6G), or fluorescent minus one controls (FMO) were used to distinguish positive and negative populations (CD44, FoxP3, GzmB). For Her2+ sorting, Her2+ cells were gated on Her2+CD45- population. |

☒ Tick this box to confirm that a figure exemplifying the gating strategy is provided in the Supplementary Information.

