## [Peer Review File · Nature]

Respiratory viral infections awaken metastatic breast cancer cells in lungs

Corresponding Author: Professor James DeGregori

Version 1:

Reviewer comments:

Referee #1

(Remarks to the Author)

The manuscript entitled with "Respiratory viral infection promotes the awakening and outgrowth of dormant metastatic breast cancer cells in lungs" presents a comprehensive investigation into the impact of respiratory viral infections, specifically influenza and SARS-CoV-2, on the awakening and proliferation of dormant metastatic breast cancer cells in the lungs. The study integrates clinical data analysis, mouse models, and advanced molecular techniques such as single-cell RNA sequencing (scRNAseq) to provide evidence that these infections can disrupt the quiescent state of disseminated cancer cells (DCCs), leading to their rapid multiplication and formation of metastatic lesions. The study is of significant importance due to its potential implications for understanding the mechanisms behind cancer recurrence following respiratory viral infections. The interdisciplinary approach, combining epidemiological analysis with experimental biology, strengthens the robustness of the findings. The use of multiple models, including the influenza PR8 strain and a SARS-CoV-2 adapted mouse model, adds depth to the study's conclusions. Overall, the study offers valuable insights into the interplay between respiratory viral infections and cancer progression, but the mechanistic studies were largely missing; therefore, addressing the below-raised issues will further fortify its scientific rigor and translational potential.

Major issues:

1. Clarification of Molecular Mechanisms Triggering Dormant Cell Awakening: While the study implicates respiratory viral infections in the reactivation of dormant breast cancer cells, the exact molecular pathways and mechanisms by which these viruses induce this awakening are not fully elucidated. It is crucial to provide a more detailed explanation of how viral components interact with cancer cells' signaling pathways, such as the involvement of inflammatory cytokines (e.g., IL-6, TNF-alpha), immune cell modulation, or direct virus-induced epigenetic changes. Further in vitro and in vivo experiments pinpointing the key molecular switches would significantly strengthen the study.
2. Dissection of the Role of Specific Immune Cells: The manuscript alludes to the involvement of CD4+ T cells and macrophages in regulating the protumor microenvironment, yet the specific roles of these immune subsets and their interaction with viral infection and cancer cell reactivation remain vague. A deeper exploration into the functional polarization of these cells (e.g., Th1, Th2, M1, M2) and their cytokine profiles could offer insights into the immune-mediated mechanisms underlying the awakening of dormant cells. Additionally, the impact of regulatory T cells (Tregs) and myeloid-derived suppressor cells (MDSCs) on this process should be considered, as they are known to modulate immune responses and may play a role in viral-induced cancer progression. CD4+ and CD8+ T are clearly involved, but again the in depth mechanisms are lacking.
3. Mechanistic Insights into the possible Virus-Specific Effects: The study involves both influenza and SARS-CoV-2 infections, but does not extensively differentiate the unique molecular mechanisms each virus employs to reactivate dormant cancer cells. Given the distinct pathogenic mechanisms of these viruses, it is important to elucidate whether they share common pathways or exert differential effects on cancer cell biology. Comparative analysis of the viral proteomes, host-pathogen interactions, and the immune response elicited by each virus could clarify virus-specific mechanisms of cancer cell awakening and inform potential therapeutic strategies tailored to different respiratory viral infections. What is the major common mechanism and differences?
3. The authors implicate interleukin-6 (IL-6) and CD4 T cells in the observed phenomena, but the mechanistic underpinnings

of how these factors mediate DCC awakening need further elucidation. Additional experiments exploring the direct role of IL-6 signaling or specific CD4 T cell subsets could strengthen the study's mechanistic claims.

4. While the study adjusts for matching factors in the UK Biobank analysis and utilizes a large sample size, it is crucial to ensure that the matching process effectively controls for potential confounders. The exclusion criteria for cancer diagnosis timing should be thoroughly justified, and the impact of this exclusion on the study's power and representativeness should be discussed.

5. The study focuses on breast cancer and primarily uses mouse models. It would be beneficial to discuss the potential for these findings to extend to other cancer types and human populations. Additionally, the use of mouse models necessitates consideration of species-specific differences in immune response and cancer biology.

Minor issues:

1. Although the manuscript mentions the use of UK Biobank data and Flatiron Health's EHR-derived database, it would be helpful to clarify if and how other researchers can access the processed datasets or replication materials.

2. Some aspects of the methodology, particularly the specifics of the single-cell RNA-seq analysis pipeline and how data from mouse models were compared to COVID-19 patient data, could benefit from more detailed descriptions to enhance reproducibility.

3. While Figure Legends are included, ensure they are concise yet informative enough for readers to understand the main findings without referring back to the text. Clear interpretation of the figures should be provided, especially regarding any trends or anomalies observed.

4. To maintain precision and consistency, please affix "+" to denote positivity when referring to CD4+ and CD8+ cells throughout the document.

5. Please ensure the proper implementation of hyphens (-), en-dashes (–), and em-dashes (—) throughout the text according to standard grammatical rules.

6. Full acronyms must be spelled out upon their first mention, accompanied by the abbreviation in parentheses. For instance, in line 106, specify "knockout (KO)" upon its initial appearance. e.g. in line 106, KO.

7. Uniformity in units is crucial; please adopt a consistent format. In line 109, "µl" is used, while in line 140, "uL" appears. Standardize to microliters (µL) throughout.

8. In line 218, why should P in processing capitalized? Be uniform. Similar issue can also be found in line 267.

9. In line 542, gene name should be italic.

Referee #2

(Remarks to the Author)

Summary – The manuscript by Chia and colleagues describes the effect of acute respiratory viral infection on dormant metastatic breast cancer cells in the lung. Using mouse models of metastatic breast cancer, the authors show that influenza infection induces an outgrowth of disseminated cancer cells in the lung. This effect is shown to require IL-6 (acutely) and CD4 T cells (progressively). The authors then examine large human cancer databases and show an association between SARS-CoV-2 infection and lung metastases/death in cancer survivors. Data from mouse models are compelling and intriguing regarding the potential translational impact. However, there are instances of over-interpretation without demonstration of mechanism.

Major Comments –

1) Mortality Data – It would be useful to include mortality data for the influenza MMTV-Her2 model to show that reactivation of these dormant metastatic cells results in a worsened outcome than in the Her2 alone. Further, mortality data in IL-6ko mice in the influenza model would also be helpful for interpretation.

2) T cell Mechanism – The authors infer a lot about T cell function from scSeq gene expression. There are no functional studies to support these data. The only protein data is by flow cytometry in EF10 showing reduced granzyme B in CD8 T cells in Her2 mice versus WT and this finding is not discussed in the manuscript. Conceptually, the proposed mechanism that CD4 T cells impair CD8 T cells post-influenza is counter intuitive and against the general understanding of how helper T cells support CD8 memory formation. What is the antigenic specificity of the T cells examined? Are they influenza specific, Her2, self antigen?

3) IL-6 Mechanism – the findings in IL-6ko are compelling, but no data are provided to infer mechanism by which IL-6 is regulating dormant metastatic cells.

Minor Comments –

1) Figure 1C – At what time point is the PBS group collected in the Her2 mice?

2) Figure 3A/3B/3F/EF4 – Quantification of iBALT, colocalization of CD4 T cells and Her2 cells, and CD8 is necessary to support the statements made.

3) Figure 4 – MA10 model details are missing regarding severity of infection, viral burden, and lung inflammation. This is needed to compare with previous influenza model.

4) Methods – Details are needed for how the lungs were homogenized for the scSeq study. Also, were lungs perfused for flow cytometry?

Referee #3

(Remarks to the Author)

Chia et al. propose that viral infection such as influenza A or SARS-CoV-2 promotes the awakening of dormant disseminated mammary cancer cells in mice by elevated IL-6 in the lung upon infection. Important for sustaining the newly proliferative cancer cells are CD4 T cells, that block CD8 T cells from killing the micrometastases.

Overall, the manuscript is interesting but lacks many controls and the link to human disease is correlative at best. Moreover, that IL6 awaken dormant cells has been reported previously (Werner-Kelin et al, Nat Comm 2020).

Major comments:

- What happens with primary tumor progression upon viral infection in these models. How can we be sure metastatic progression over time is not compromised by effects in the primary tumor site? Also, it's unclear if mammary glands (so non-cancerous) were used for downstream analysis throughout the paper (as also described in Figure 1a), as MMTV-models develop several tumors simultaneously.
- Can the authors exclude increased dissemination as a mechanism?
- Figure 2: Analysis of dormant or proliferating cells is only done by cell number or ki67 staining. Even though authors present relative and absolute quantification in their histological analysis (Figure 2b), it will be more informative to present cell number per metastatic cluster instead of total Her2+ cells. Additional use of dormant cells tracking approaches is recommended to support key findings (lack of EdU incorporation or p27, p21, etc.), especially in the PyMT model. Several dormancy signatures have been reported and could be used to discriminate between dormant cells and micrometastasis. Moreover, how do the authors explain that there is almost a complete ablation of Ki67+ cancer cells at later timepoints (Figure 2b)? Especially whilst in the total lung, there are still over 100 Ki67+ Her2+ cancer cells; but they represent almost 0% of the total Her2+ population? I would expect that these metastases must, if they're not expanding, at least undergo tumor mass dormancy. So how come the proportion of proliferating cancer cells is so low?
- It is important to show which cell type in the lung microenvironment increases IL-6, IL- β secretion upon viral infection. Did authors also evaluate IL- β KO? were the results similar? IL6 stimulation was shown to promote awakening and cell proliferation before, the more important aspect here – how it gets maintained?
- A CD4+ T-cells aspect is presented in the paper, but there is not sufficient evidence on how changes in this population affect cancer cells. Addition of the transcriptomic data from the cancer cells is important to support authors' statements. The authors hint towards a reduced mitochondrial content of CD4+ T cells, yet provide no data. A flow cytometry experiment using mitochondrial staining would help support this claim.
- Data presented in the manuscript mainly focuses on an IAV infection model, and is quite scarce on the COVID model. Given that the clinical dataset is focused exclusively on COVID, the authors should confirm their main mechanistic findings in MA10 model, and present these apart from the later change in the HER2+ cells after infection in the Figure 4.
- In Extended Data Figure 1b: The number of cells found in the BAL on day 15 are almost double in the Her2 mice compared to the WT (although statistics are lacking). Can the authors comment on how this finding might skew the analysis of the scRNAseq of the T cells (esp. regarding the CD4 T cell proliferation hint the authors state in line 565)?
- Where are the Her2+ cancer cells in the scRNAseq? It would be interesting to see the transcriptomic changes occurring in this compartment. Additionally, the authors do not seem to characterize alveolar macrophages. It would be interesting for the reader to know in which other cell subsets they are divided or included, as they are of crucial importance in lung viral infections.
- Generally, the scRNAseq dataset hits were nowhere validated on protein level, which should be done (either by stainings or functional assays).
- There is a bias in the Flatiron health database where the authors stratify for COVID-19 infected women. As the SARS-CoV-2 pandemic is recent (4 years approximately), the risk of death by metastases depends on breast cancer subtype (if ER+ or TNBC). Thus, it would be of high importance to stratify the dataset accordingly. Or at least provide additional data on the percentages of subtypes for all the analyses. Moreover, the time of disease detection and their treatment categories matter. Additionally, other potential confounding factors should be included, such as BMI and smoking.

Minor:

- The source of the MMTV-HER2 and PyMT mouse models is not provided.
- Methods section on image analysis lacks key information on how many sections were evaluated per sample (only in figure legends), regions selection, steps in the analysis process which led to multiple key graphs presented as «Total cell number» or «Cell number per field».
- Figure 1: To improve figure reading, include 15 dpi label to (e) graph. (f) Add information on dpi in the PyMT model. Why quantification of the lesions was limited to certain size? Apart from providing the number of metastatic lesions (Fig. 1f), it would be good to have the total metastatic burden (as % of cancerous area per total lung area). This is crucial to see how

much under control those virally induced metastases are after the infection is cleared.

- Figure 1: Which statistical test was used for 2 group comparisons? the multiple test correction for One-way ANOVA should be indicated. Moreover, statistical tests are missing in some of the figures (Extended Data Figure 1,2, 3a).
- Extended Figure 1d: if the authors could circle the area they consider as metastases, that would aid in composition recognition. Could it also be, that the metastases just have a different immune infiltration or grow out in different environments (alveolar vs. intra epithelial)?
- Why do the inoculation methods differ between influenza and SARS-CoV-2?
- Please refrain from the word "FACS" buffer, as it is a trademark by BD. Any other appropriate label will do.
- Line 192: were the lungs incubated shaking or still? And was the leftover tissue manually homogenized? How long was the hemolytic buffer applied?
- Line 246: Amount of RNA reversely transcribed is missing.
- Line 402: Please capitalize the V from FVB.
- Figure 2: unclear which cells the arrows indicate in the figure legend. Also, why is dpi 60 missing in the graph (e)?
- Extended Data Figure 1b+c: Unclear about the number of mice used. Taken into account that a lavage does not always yield the same recovery of buffer injected, and additionally that the lavage was only performed once with 1 mL, this data needs to include single mouse datapoints instead of just a bar graph.
- In the methods section the processing of the BALF is only described by centrifuging and resuspending. Was any erythrolysis performed on these samples? Which cell types are included in the analysis? The numbers are very high, especially comparing to the immune cell counts from Extended Data Figure 4, which are much lower.
- Extended Data Figure 1d: Please increase the labeling size of the scale bar.
- Figure 2: The authors conclude that «a hybrid state allows for dormant DCC awakening» based on their results. They show that % of vimentin+ cells are not significantly altered upon IAV infection in IL6 KO mice. While focusing further on the mechanistic part is not a goal of this manuscript, could you elaborate on this connection and place your findings into the scope of current knowledge in the dormancy field in the discussion?
- Extended Figure 2: Please add a "n.d." or equivalent to samples where you were not able to measure the cytokines.
- The authors state (line 519+520), that CD8+ T cells are increased upon CD4+ T cell depletion, however the only data provided is one microscopy image. Thus, the authors should provide flow cytometry analysis of whole lung digest for a proper data representation. Alternatively, as the authors subsequently provide CD8+ T cell depletion data, they could simply remove this statement.
- Lines 533-536: The sentence references to a wrong figure. Unclear about the message the authors are trying to make.
- Extended Figure 3a: Please indicate in the figure legend or methods how many tumors the mice developed on average. Were any mice excluded from these analyses? Also, it is unclear how many mice were used for the Kaplan-Meier plot.
- Figure 3b: The authors claim that they see almost no Her2+ cells when there's no CD4+ T cells, but all the data provided is just each one image, with one lacking a DAPI staining. Therefore, the authors should provide more data. If the intention was to show the same image as the upper panel of Fig. 3b, then that should be stated. Also, are there any/more CD8 T cells present in those areas?
- Whole figure 3: Please enlarge the scale bars on all microscopy pictures.
- Extended Data Figure 3: The figure legend mentions that the criteria for endpoint was tumor size in «a». Please clearly label it in the Figure «Days to maximum tumor size» instead «Probability of Survival».
- Extended Data Figure 3b: Why are the authors using a mix between bar graph and SD lines? Please unify.
- Extended Data Figure 4a-d: Please add how many animals were analyzed.
- Extended Data Figure 5: The authors are asked to provide previous gating strategy above the T cell gates for the samples. Additionally, what message should the third column be conveying? CD4 and CD8 gates are already included in the second column. Arrows pointing from one gate to the subsequently gated should be added. Please capitalize "Lung", and unify that fluorochromes are labelled everywhere on the axes, as well as that the axes are aligned.
- Figure 3/Ext. Figure 6: Some images have a double scalebar and too small labeling of its length. Please correct for consistency. Addition of an experimental set-up schematic will help readers understand results in «c-i».
- Extended Data Figure 6: Why only data on ki67+ Her2+ cells for aCD4, but not aLy6G presented? Scalebar information of the images is lacking in the figure legend.
- Extended Data Figure 7c: Unclear which sample is shown in the UMAP. The same UMAP should be shown for all sample types (WT vs MMTV).
- Any reason that only human observational data on Covid and not on influenza are provided?
- Authors should discuss limitations of their study.

Referee #4

(Remarks to the Author)

A) The manuscript "Respiratory viral infection promotes the awakening and outgrowth of dormant metastatic breast cancer cells in lungs" demonstrates a mechanism by which influenza A virus infection causes progression of dormant metastatic breast cancer. A linked set of observational epidemiologic analyses show increased risk of progression among cancer survivors following SARS-CoV-2 diagnosis in the UK Biobank and a US EHR dataset. The degree to which these findings strengthen and support the epidemiologic relevance of the primary finding, however, depends upon the strength of the evidence provided by the authors that SARS-CoV-2, in the mouse model, is causing a similar phenomenon to IAV. More on that below.

B) To my knowledge the findings are highly novel and of great importance across numerous fields of research, thus meriting the attention of the wide scientific community that reads Nature.

C/D) My area of expertise is epidemiology and I focus on the analysis of the UK Biobank and US EHR data as I do not have the necessary knowledge to review the mechanistic experiments.

UK biobank study: Confirm if the matching procedure is matching those who test positive within 2020 pre-vaccine with those who have at least one negative test over this period? The authors refer to test positive and test negative individuals but I do not see an explicit criterion for the controls as having needed to test negative; presumably this is being done to match on healthcare seeking behavior? The concern that arises in such studies is the possibility that sicker people are being tested more often and thus have more chance for a positive SARS-CoV-2 test to occur (or to be documented) than those who are relatively healthy and at lower risk of experiencing the outcome. To further alleviate concerns about missed infections, it would be of value to determine if any control subjects are available who definitely (based on results of several tests at a minimum interval like 60 or 90 days) remained uninfected throughout the period of interest.

Relatedly, people seem to be followed up until 2022 for cancer outcomes in this analysis. By mid-2022 over half of the US and UK populations had experienced SARS-CoV-2 infection so it does not make sense to use their infection status pre-vaccine (2021) only as the exposure; this is likely misclassified by 2022 for many or most people. The infection and cancer outcomes should be evaluated over the same period of time, or a shorter maximum length of time after COVID vaccination (<<<1.5 years) should be used to reduce this problem.

The Flatiron EHR analysis does not make any such requirement for a negative test. Because these people were participating in cancer cohorts is it fair to assume all received regular testing? For the cox proportional hazards model, what was the unit of time? Did observation windows continue until a COVID-19 diagnosis was made or was there a specified unit? The adjustment set makes no reference to comorbid conditions or healthcare-seeking behavior and thus appears woefully inadequate for EHR-based real world evidence studies. I note that adjustment for comorbidities was also missing in the UK Biobank studies although at least these matched on cancer type and included more covariates in the adjustment set.

E) The significance of the COVID-19 epidemiologic study linkage is dependent upon whether a similar mechanism to that seen with IAV occurs with SARS-CoV-2. At present the applicability/interchangeability of the IAV and SARS-CoV-2 findings seems limited; a single sentence states that "Infection of MMTV-Her2 (C57BL6/J) mice with MA10- SARS-CoV-2 resulted in a striking increase in Her2+ 595 cells by 28 dpi", and I am unsure of whether this biomarker is sufficient to demonstrate that IAV and SARS-CoV-2 are doing the same thing. If not, I am less convinced that the epidemiologic study is adding something of importance to the manuscript.

F) My suggestions for revision are contained in my statistical review (C/D).

G) I believe appropriate references to previous work are made.

H) The manuscript is clearly written.

Version 2:

Reviewer comments:

Referee #1

(Remarks to the Author)

After reading the revised version, this reviewer feels that the authors have largely satisfied reviewers. In the revised version, the authors provided mechanistic analysis (to be honest, not in-depth enough, mostly at the level of phenomenological description) and suggest that the IL6 (together with others) probably played major role in awakening and expanding disseminated cancer cells.

This reviewer has a final request, that is: could the authors provide a therapeutic strategy to prevent (breast) tumor progression caused by respiratory virus infection, and can they conduct experiments at the animal level to verify it?

Referee #2

(Remarks to the Author)

The authors have adequately addressed my prior concerns and have added new data and analyses to strengthen the manuscript.

Referee #3

(Remarks to the Author)

The revised version includes additional data that have improved some aspects of the manuscript. In particular, the cell line, MA10 COVID infection model, RNAseq of cancer cells post infection and analysis of the patient data. Nonetheless, the link between IL-6 and dormant cancer cell exit from dormancy is not fully addressed in the revised version. Moreover, the data to support the effect of COVID-19 on increased lung metastasis of breast cancer patients are weak.

• Figure 3g and Extended Data Figure 4 new graphs show a mammosphere assay results. This assay only implicates that

established organoids acquire faster proliferating rate upon stimulation with IL-6, but not a direct switch from dormant state to proliferation. An in vitro model including a co-culture between cancer cells and potential cell type(s) promoting IL-6 burst after viral infection can better mimic in vivo observations of the authors. Additional stainings for proliferation and dormancy markers in a time course fashion, as well as treatment conditions blocking IL-6 signalling are necessary to clarify the mechanism.

The authors can further explore the exact mechanism by changes in the cancer cells observed from RNAseq — changes in epithelial and mesenchymal markers, upregulation of extracellular matrix proteins like collagens, and acquisition of an immune escape phenotype (upregulation of Cd274, which encodes PDL1) and decreases in B2m (required for antigen presentation by MHC Class 1).

- The authors propose IL-6 blocking strategies for preventing DDC awakening in cancer survivors, yet do not show pre-clinical data that suggest this would be beneficial in vivo.

Minor:

- Please increase the mouse number per group in data presented at least to n=3 (new Figure 1g).
- In Figure 6a please add «years» to the graph labeling with «5,10».
- In Extended Data Fig. 1 is missing indication of n mouse numbers, as well as a statistical test indicated «one-way ANOVA» can't be implemented for 2 groups comparisons (also applies to Figure 1). Please correct.
- Regarding Mitotracker staining, adding flow cytometry antibodies for 5 min at the last incubation would very likely not yield enough staining intensity (generally 30 min are recommended unless properly assessed). What are the reasons for this short incubation? Could the authors show that the MFI of the CD4 and CD8 epitopes are not altered upon IAV infection?
- Discussion is not containing the study limitations, as the authors state in the point-by-point.

Referee #5

(Remarks to the Author)

I am reviewing the observational studies of UK Biobank and Flatiron health populations, how responsive the reviewers were to comments and any remaining concerns.

- Test-negative “controls” are not necessarily individuals who have never experienced a SARS-CoV-2 infection. This is acknowledged in the Flatiron analysis but not in the UK Biobank analysis. I think any bias introduced by this is towards the null.
- The increased effect size for shorter follow-up periods could be due to time-varying effects on mortality risk, but could also be due to increased bias by making the infection profile of the test-positives and test-negatives more similar, as the probability of a test-negative accruing an infection that was not tested is higher. The authors introduced this sensitivity analysis to respond to a concern of a previous reviewer about misclassification of the exposure, but are interpreting the results of this sensitivity analysis as representing a biological effect. It should be acknowledged that misclassification of the outcome (which is differential between test-positives and test-negatives) is likely contributing to the attenuation of the ORs.
- In the methods, the distinction between the two analyses performed on the Flatiron population (i.e. lines 534-540 and 541-562) is not entirely clear to me, and the second analysis is not well justified. Why not only present the analysis adjusted for additional potential confounders, if you are concerned about these variables introducing confounding? Why does the analysis switch from multivariable Cox PH to stratified Cox PH?
- In the Flatiron analysis it is stated that the analysis included 36,216 COVID-19 positive patients and 532 COVID-19 negative patients (with similar numbers for the second analysis). I have a couple of comments/concerns about this.
- The follow-up period is never stated, but I assume the last follow-up date is in 2022. These numbers are a little surprising (the high proportion who tested positive) but not outside the realm of possibility if this is a highly tested population. But the small number of individuals in the COVID-19 negative cohort does raise questions about how representative this group is in term of metastasis risk (although see my comment below). It's not 100% clear what this population is though – are they individuals who never tested positive throughout the follow-up period, or does it also include individuals who had metastasis (or were otherwise censored) before testing positive?
- As COVID-19 diagnosis status was treated as a time-varying covariate, the numbers given for positive and negative individuals is less relevant than the person-time contributed by test-positive and negative individuals. If I'm understanding correctly, diagnosis was entered into the Cox model as a time-varying covariate, so presumably most of the positive individuals contributed “negative” person-time before their COVID-19 diagnosis. This should also alleviate concerns about the impact of the fully negative cohort, which are mentioned in the results. I think it would make more sense and be more informative to give the test-negative and test-positive person-time.
- How does the risk of metastasis evolve over time in a breast cancer patient? If it increases over time since diagnosis/remission, there may be some confounding by time here, as individuals can only go from negative to positive COVID-19 diagnosis, so COVID-19 positive person-time will be later in time (on average) than negative person-time. Stratifying by year of diagnosis, as they have done, may help alleviate this bias.

Version 3:

Reviewer comments:

Referee #1

(Remarks to the Author)

If the authors feel that it is difficult to propose and verify treatments in a short period of time, the reviewers feel that they should at least propose some potentially effective methods in the discussion.

Referee #3

(Remarks to the Author)

The authors addressed all my minor comments and discussed why it is not possible to perform the experiments related to the major points. Thus, the weaknesses mentioned previously were not addressed.

For example:

- The data in Figure R1 show that IL-6 is essential for the development of the virus-induced lung metastases in this model; which addresses a question that is different from the one I asked. It still does not establish its implication in the exit from dormancy.
- The authors modified the text to highlight the limitations of the mamosphere assay. But this has not addressed the implication in the switch from a dormant to a proliferative state.

Referee #5

(Remarks to the Author)

The authors have responded to all my comments in a satisfactory way. One typographical issue - line 1429-1430 should be deleted as the authors have replaced this erroneous information with the correct information on the number of COVID-19 positive and negative patients.

November 23, 2024

Dear Reviewers,

We greatly appreciate the thorough critiques of our manuscript entitled **Respiratory viral infection promotes the awakening and outgrowth of dormant metastatic breast cancer cells in lungs** (2024-02-04019) submitted for publication in Nature. We have spent the last ~5 months addressing the reviewers' suggestions and identified deficiencies. To address the reviewers' concerns, we performed numerous additional experiments and analyses, including -

- 1) Further characterization of the murine SARS-CoV-2 model, showing that awakening, expansion, and phenotypic shifts of disseminated cancer cells (DCC) in the lungs following SARS-CoV-2 (MA10) infection are similar to what is observed for influenza virus infections, and are dependent on IL-6.
- 2) Detailed characterization of the gene expression changes in different immune subsets following influenza virus infection and the profound influence of Her2+ DCC on these changes, using single cell (sc) RNA-seq. These analyses also show how depleting CD4⁺ T cells restores an activated phenotype to CD8⁺ T cells.
- 3) Demonstration that depletion of CD4⁺ T cells in influenza infected MMTV-Her2 mice unleashes the killing activity of CD8⁺ T cells towards mammary tumor cells (showing direct killing of these cells in vitro).
- 4) Demonstration that IL-6 promotes the growth of mammary tumor cells as mammospheres, revealing that IL-6 can directly affect the expansion of these cells.
- 5) RNA-seq on isolated DCC from influenza infected Her2 mice, demonstrating strong upregulation of the IL-6-Jak-Stat pathway, dramatic changes in epithelial and mesenchymal markers, and increased expression of immune modulatory factors (like collagens and PDL1), providing new insight into the mechanisms of DCC awakening and immune avoidance.
- 6) Reanalysis of the breast cancer metastases data from Flatiron Health, showing that the effect of COVID on metastatic progression to the lungs is robust to corrections for co-morbidities, breast cancer subtype (e.g. ER status), and other potential confounding factors.
- 7) Reanalysis of data from the UK Biobank, limiting analyses to individuals with either positive or negative test results for SARS-CoV-2. In addition, we conducted an

additional sensitivity analysis, censoring deaths at different time points, which indicates that excess cancer-related mortality is greater close to SARS-CoV-2 infection, in keeping with experimental observations.

In all, addressing the reviewer critiques has led to a greatly improved body of work, with substantially more insight into the mechanisms of immune avoidance, IL-6 dependent awakening of DCC, and similarities for the influenza and SARS-CoV-2 contexts, as well as more thorough and better controlled epidemiological analyses of associations between SARS-CoV-2 infections and cancer metastases and mortality. We will elaborate on each of these advances, together with other analyses and experiments performed, in the response to each critique below. Our response will be in blue, and all figures referred to below present new data unless italicized. We also provide a version of the manuscript where major changes are in blue.

Referee #1

The manuscript entitled with “Respiratory viral infection promotes the awakening and outgrowth of dormant metastatic breast cancer cells in lungs” presents a comprehensive investigation into the impact of respiratory viral infections, specifically influenza and SARS-CoV-2, on the awakening and proliferation of dormant metastatic breast cancer cells in the lungs. The study integrates clinical data analysis, mouse models, and advanced molecular techniques such as single-cell RNA sequencing (scRNAseq) to provide evidence that these infections can disrupt the quiescent state of disseminated cancer cells (DCCs), leading to their rapid multiplication and formation of metastatic lesions. The study is of significant importance due to its potential implications for understanding the mechanisms behind cancer recurrence following respiratory viral infections. The interdisciplinary approach, combining epidemiological analysis with experimental biology, strengthens the robustness of the findings. The use of multiple models, including the influenza PR8 strain and a SARS-CoV-2 adapted mouse model, adds depth to the study's conclusions. Overall, the study offers valuable insights into the interplay between respiratory viral infections and cancer progression, but the mechanistic studies were largely missing; therefore, addressing the below-raised issues will further fortify its scientific rigor and translational potential.

We thank the reviewer for their very positive assessment of our study. We have added substantial new data to bolster our understanding of underlying mechanisms.

Major issues:

1. Clarification of Molecular Mechanisms Triggering Dormant Cell Awakening: While the study implicates respiratory viral infections in the reactivation of dormant breast cancer cells, the exact molecular pathways and mechanisms by which these viruses induce this awakening are not fully elucidated. It is crucial to provide a more detailed explanation of how viral components interact with cancer cells' signaling pathways, such as the involvement of inflammatory

cytokines (e.g., IL-6, TNF-alpha), immune cell modulation, or direct virus-induced epigenetic changes. Further in vitro and in vivo experiments pinpointing the key molecular switches would significantly strengthen the study.

We have now performed a number of new experiments to increase our understanding for how pulmonary virus infections promote metastatic outgrowth including:

- 1) Treatment of Her2+ mammary tumor organoids (mammospheres) with IL-6 results in increased organoid size, demonstrating a direct role for IL-6 in the awakening and expansion of DCC. See **new Figure 3g**; **new Extended Data Figure (EDF) 4d and 4e**.
- 2) We now show that lung epithelial cells produce IL-6 in response to influenza infection (**new EDF 3c**).
- 3) We show that IL-6 is required for the mesenchymal/epithelial state changes in DCC that occur following either influenza or SARS-CoV-2 infection (**new Figure 3f** and **new Figure 5e-g**).
- 4) We have now performed RNA-seq on isolated Her2+ DCC from mock and influenza infected mice (day 9), revealing activation of the IL-6-Jak-Stat signaling pathway, dramatic changes in epithelial and mesenchymal markers, upregulation of extracellular matrix proteins like collagens, and acquisition of an immune escape phenotype (upregulation of *Cd274*, which encodes PDL1) and decreases in *B2m* (required for antigen presentation by MHC Class 1)). Epithelial markers include *Cdh1*, *Cldn2&5*, *Krt19*, *Klf4*, and *Ovol2*, and mesenchymal markers include *Cdh2*, *Cdh11*, *, *Eng*, and *Vim*. It is notable that in contrast to previous studies that show a clearer transition from a dormant mesenchymal state to an awakened epithelial state, we observe an interesting hybrid phenotype. The upregulation of many mesenchymal markers is consistent with our previous demonstration that awakened DCC exhibit a distributed pattern in the lungs (instead of a more uniform clump), suggestive of a more motile/invasive phenotype. In addition to the upregulation of genes encoding PDL1 and β 2-microglobulin, the dramatic increases in the expression of collagens (validated using trichrome staining) provides further insight into DCC outgrowth and immune escape, as collagens have been shown to limit T cell infiltration and activity (PMID: 34956223), and to promote survival and growth of breast cancer cells (many reports, including PMID: 35121989). These new data are presented in the **new Figure 2e-h** and the associated **new EDF 2**.*

2. Dissection of the Role of Specific Immune Cells: The manuscript alludes to the involvement of CD4+ T cells and macrophages in regulating the protumor microenvironment, yet the specific roles of these immune subsets and their interaction with viral infection and cancer cell reactivation remain vague. A deeper exploration into the functional polarization of these cells (e.g., Th1, Th2, M1, M2) and their cytokine profiles could offer insights into the immune-mediated mechanisms underlying the awakening of dormant cells.

Thanks for these suggestions.

- 1) We performed additional analyses of our scRNAseq data. The results show that the ratio of memory to effector CD4 and CD8 cells increases in the lungs of Her2 mice relative to WT mice, as does the ratio of M2-like to M1-like macrophages, consistent with a more immune tolerant state. Moreover, gene expression within each of these immune subsets is dramatically changed by the presence of Her2+ DCC to a more immune suppressed pattern (**new Figures 4g-h, EDF 10-14**). Notably, depleting CD4 cells restores a more activated phenotype to other immune cells.
- 2) CD8 T cells purified from the lungs of influenza infected Her2+ (but not WT) mice selectively kill mammary tumor cells *ex vivo*, but *only* if CD4 cells had been depleted *in vivo* during the infection (**new Figures 4i-j and EDF 17**).

Thus, we have established that DCCs reprogram CD4 cells during influenza virus infection, and these reprogrammed CD4 cells impair the anti-tumor response of CD8 cells. In all, these new data provide novel insight into how expanded Her2+ DCC avoid immune elimination through modulation of macrophage phenotype and CD4 cell phenotype that restrain CD8 cell killing of DCC.

Additionally, the impact of regulatory T cells (Tregs) and myeloid-derived suppressor cells (MDSCs) on this process should be considered, as they are known to modulate immune responses and may play a role in viral-induced cancer progression. CD4+ and CD8+ T are clearly involved, but again the in-depth mechanisms are lacking.

We thank the reviewer for the comment. We used the FoxP3-DTR mouse to specifically deplete FoxP3+ regulatory T (Treg) cells. In three experiments, we did not observe any reduction in the burden of DCC post-influenza infection, however we were unable to achieve more than 50% depletion of Treg cells. Given the partial depletion of Treg cells, we feel that we cannot definitely conclude that Treg cells do not play a role, thereby we did not include these data in the revised version. However, if the reviewers or the editor would like us to include these results (albeit negative), we can do so. Nonetheless, we note that we do not observe any significant effect of Her2 DCC on Treg cell numbers post-infection (which are a small fraction of total CD4 cells in the lungs). Given that we observe a clear shift of the *entire* CD4 population towards an immune suppressive phenotype, and that DCC can express negative regulators of the T cell response post-infection (e.g. PDL1 and collagens), we have focused on these mechanisms instead. Indeed, our demonstration that depleting CD4 cells leads to CD8 cell influx into the lungs post-IAV, and that these CD8 cells have a more effector/cytotoxic activity (both by scRNA-seq and by *in vitro* killing assays), provides a clear mechanism to explain how DCC reprogramming of CD4 cells promotes immune suppression. Finally, we show that depletion with anti-Ly6g, which should deplete MDSC, does not alter the ability of influenza virus infection to awaken DCC (**EDF 8c-e**).

3. Mechanistic Insights into the possible Virus-Specific Effects: The study involves both influenza and SARS-CoV-2 infections, but does not extensively differentiate the unique molecular mechanisms each virus employs to reactivate dormant cancer cells. Given the

distinct pathogenic mechanisms of these viruses, it is important to elucidate whether they share common pathways or exert differential effects on cancer cell biology.

Comparative analysis of the viral proteomes, host-pathogen interactions, and the immune response elicited by each virus could clarify virus-specific mechanisms of cancer cell awakening and inform potential therapeutic strategies tailored to different respiratory viral infections. What is the major common mechanism and differences?

The reviewer raises good points, since in our original manuscript we provided limited data regarding SARS-CoV2. In response, we have now infected MMTV-Her2 and IL6 KO MMTV-Her2 mice with mouse adapted SARS-CoV-2 MA10 and analyzed their lungs 3- and 9 dpi. Similar to influenza (IAV) infection, we observed dramatic increases in Her2⁺ DCC cell cycling and expansion post-infection in MMTV-Her2 mice, which were largely prevented in IL-6 KO MMTV-Her2 mice (**new Figure 5c-f**). Immunofluorescence experiments further demonstrate loss of vimentin and transient gain of EpCAM (3 dpi) in DCC post-MA10 infection, and these changes required IL-6 (**new Figure 5e-f**). We also analyzed cytokine production in the lungs post-MA10 infection, demonstrating increased levels of IL-6, IL-1 β and interferons α , β and γ post-infection (**new EDF 18**). All of these results are similar to those observed with IAV infection, demonstrating common responses for very different respiratory viruses. Given the public relevance of understanding how COVID-19 contributes to cancer progression, these results significantly increase the impact of the manuscript, providing mechanistic insights for our described epidemiological associations.

4. The authors implicate interleukin-6 (IL-6) and CD4 T cells in the observed phenomena, but the mechanistic underpinnings of how these factors mediate DCC awakening need further elucidation. Additional experiments exploring the direct role of IL-6 signaling or specific CD4 T cell subsets could strengthen the study's mechanistic claims.

As described above (see Point 1), we have: 1) used organoids to demonstrate a direct role for IL-6 in the expansion of mammary tumor cells, 2) analyzed different CD4 and CD8 subsets in scRNAseq data to elucidate how expanding DCC and CD4 cells influence immune cell phenotypes to promote immune escape, and 3) analyzed gene expression in DCC post-IAV infection to understand how infection promotes DCC awakening and immune tolerance. Please, see Point 1 for specific new figures.

5. While the study adjusts for matching factors in the UK Biobank analysis and utilizes a large sample size, it is crucial to ensure that the matching process effectively controls for potential confounders. The exclusion criteria for cancer diagnosis timing should be thoroughly justified, and the impact of this exclusion on the study's power and representativeness should be discussed.

We thank the reviewer for raising this important issue. Matching in the UK biobank was done both through exact matching and propensity score matching. We first performed exact matching on sex, cancer type, and diagnosis date (allowing for 5 years difference; mean time difference between diagnosis of test positives and matched negatives is 0.19 year). This

ensured that cases and controls had the same sex, cancer-type, and similar survival. We then did propensity score matching for age, ethnicity, smoking, alcohol, education, employment, BMI and income. As illustrated below, the standardized mean difference (SMD) for each matching variables indicates a very good balance between test positives and test negatives (SMD<0.1, p>0.44 across variables).

		Total	Test negatives	Test negatives	p	SMD
	n	4837	4350	487		
sex (%)	Female	2569.4 (53.1)	2314.4 (53.2)	255.0 (52.4)	0.727	0.017
	Male	2267.6 (46.9)	2035.6 (46.8)	232.0 (47.6)		
Household income (%)	Lower	2484.1 (51.4)	2229.1 (51.2)	255.0 (52.4)	0.901	0.022
	Higher	2111.6 (43.7)	1903.6 (43.8)	208.0 (42.7)		
	Other	241.3 (5.0)	217.3 (5.0)	24.0 (4.9)		
education (%)	None	1126.2 (23.3)	1007.2 (23.2)	119.0 (24.4)	0.94	0.03
	A Level	493.6 (10.2)	444.6 (10.2)	49.0 (10.1)		
	Other	2132.4 (44.1)	1920.4 (44.1)	212.0 (43.5)		
	University	1084.8 (22.4)	977.8 (22.5)	107.0 (22.0)		
employment (%)	Unemployed	398.3 (8.2)	351.3 (8.1)	47.0 (9.7)	0.505	0.056
	Employed	2541.0 (52.5)	2288.0 (52.6)	253.0 (52.0)		
	Retired	1897.7 (39.2)	1710.7 (39.3)	187.0 (38.4)		
Smoking status (%)	Never	2417.9 (50.0)	2176.9 (50.0)	241.0 (49.5)	0.743	0.036
	Current	402.3 (8.3)	357.3 (8.2)	45.0 (9.2)		
	Former	2016.8 (41.7)	1815.8 (41.7)	201.0 (41.3)		
Alcohol consumption status (%)	Never	175.4 (3.6)	157.4 (3.6)	18.0 (3.7)	0.754	0.036
	Current	4474.1 (92.5)	4027.1 (92.6)	447.0 (91.8)		
	Former	187.6 (3.9)	165.6 (3.8)	22.0 (4.5)		
Ethnicity (%)	White	4723.0 (97.6)	4246.0 (97.6)	477.0 (97.9)	0.806	0.057
	Black	41.1 (0.9)	37.1 (0.9)	4.0 (0.8)		
	Other	38.7 (0.8)	36.7 (0.8)	2.0 (0.4)		
	SouthAsian	34.2 (0.7)	30.2 (0.7)	4.0 (0.8)		
BMI (mean (SD))		27.79 (4.87)	27.77 (4.90)	27.95 (4.63)	0.448	0.037

Age (mean (SD))	58.19 (7.54)	58.18 (7.48)	58.31 (8.06)	0.73	0.017
--------------	--------------	--------------	------	-------

Nevertheless, to address the potential for residual confounding, as detailed in the methods section (starting on p20), our models were adjusted for the matching variables.

To further investigate the similarity of the test positives and matched test negatives, we considered several comorbidities as potential additional confounders: ischaemic stroke, CHD, diabetes, hypercholesterolaemia, and hypertension. As illustrated below, although slightly attenuated, the effect size estimates were similar and our conclusions were not affected by the additional adjustment for comorbidities, hence suggesting limited residual confounding.

	Original model		Model adjusted for co-morbidities	
	OR (95% CI)	p	OR	p
All-cause mortality	4.50 (3.49, 5.81)	6.88E-31	4.37 (3.37, 5.67)	1.13E-28
Cancer-specific mortality	1.85 (1.14, 3.02)	0.013401	1.80 (1.10, 2.94)	0.018778
Non-COVID-19 mortality	2.56 (1.86, 3.51)	6.95E-09	2.42 (1.75, 3.35)	9.03E-08

6. The study focuses on breast cancer and primarily uses mouse models. It would be beneficial to discuss the potential for these findings to extend to other cancer types and human populations.

UK-Biobank analyses did focus on all cancers. As a population-based study the number of cancers cases by subtype is limited (see below). We therefore are not powered enough to do cancer-specific analyses in UK-Biobank.

Cancer types	Frequency
Breast	1032
Digestive organs	340
Female genital	199
Lymphoid & hematopoietic	250
Male genital	742
Respiratory, intrathoracic	70
Skin	2102
Urinary tract	102
Total	4837

Within Flatiron, only breast cancer is represented by a well-annotated dataset of sufficient size for the analyses that we performed.

Additionally, the use of mouse models necessitates consideration of species-specific differences in immune response and cancer biology. The reviewer raises a salient point, and we have added a statement to the Discussion acknowledging that immune and cancer changes in response to respiratory virus infection could differ between these species. Of course, the concordance of our results in mouse models with the associations derived from UK Biobank and Flatiron databases does not mean that all details are conserved.

Minor issues:

1. Although the manuscript mentions the use of UK Biobank data and Flatiron Health's EHR-derived database, it would be helpful to clarify if and how other researchers can access the processed datasets or replication materials.

UK Biobank data arises from application number 69328 (to MC-H). We are not allowed to render UK Biobank data available. However, UK Biobank is accessible upon successful application to the UK Biobank access committee. Pre-processing/recoding and analytical scripts are available upon request to ensure that any researcher with active access to UK Biobank data could replicate our work. This has been clarified in the methods section (under *Data Availability*, p23).

For Flatiron Health results, the data that support the findings reported here were originated by and are the property of Flatiron Health, Inc., which has restrictions prohibiting the authors from making the data set publicly available. Information on submitting requests for data sharing by license or by permission for the specific purpose of replicating results in this manuscript is provided in the Methods.

2. Some aspects of the methodology, particularly the specifics of the single-cell RNA-seq analysis pipeline and how data from mouse models were compared to COVID-19 patient data, could benefit from more detailed descriptions to enhance reproducibility.

We have expanded the methods section describing the analyses of scRNAseq data. We do not compare our mouse scRNAseq data to COVID-19 patient data.

3. While Figure Legends are included, ensure they are concise yet informative enough for readers to understand the main findings without referring back to the text. Clear interpretation of the figures should be provided, especially regarding any trends or anomalies observed.

These corrections have been made.

4. To maintain precision and consistency, please affix "+" to denote positivity when referring to CD4+ and CD8+ cells throughout the document.

“+” has been added to all CD4+ and CD8+ cells.

5. Please ensure the proper implementation of hyphens (-), en-dashes (–), and em-dashes (—) throughout the text according to standard grammatical rules.

Thanks for catching this, and these corrections have been made.

6. Full acronyms must be spelled out upon their first mention, accompanied by the abbreviation in parentheses. For instance, in line 106, specify "knockout (KO)" upon its initial appearance. e.g. in line 106, KO.

Acronyms have been spelled out upon their first mention accompanied by the abbreviation in parentheses.

7. Uniformity in units is crucial; please adopt a consistent format. In line 109, "µl" is used, while in line 140, "uL" appears. Standardize to microliters (µL) throughout.

All "ul"s have been replaced with "µL".

8. In line 218, why should P in processing capitalized? Be uniform. Similar issue can also be found in line 267.

The incorrectly capitalized words have been corrected.

9. In line 542, gene name should be italic.

Gene names have been italicized. We appreciate the reviewer's catching these errors!

Referee #2

Summary – The manuscript by Chia and colleagues describes the effect of acute respiratory viral infection on dormant metastatic breast cancer cells in the lung. Using mouse models of metastatic breast cancer, the authors show that influenza infection induces an outgrowth of disseminated cancer cells in the lung. This affect is shown to require IL-6 (acutely) and CD4 T cells (progressively). The authors then examine large human cancer databases and show an association between SARS-CoV-2 infection and lung metastases/death in cancer survivors. Data from mouse models are compelling and intriguing regarding the potential translational impact. However, there are instances of over-interpretation without demonstration of mechanism. We thank the reviewer for their positive assessment of our work, and we have addressed their concerns below.

Major Comments –

1) Mortality Data – It would be useful to include mortality data for the influenza MMTV-Her2 model to show that reactivation of these dormant metastatic cells results in a worsened outcome than in the Her2 alone. Further, mortality data in IL-6ko mice in the influenza model would also be helpful for interpretation.

We are unable to assess the consequences of metastatic outgrowth in the lungs on mouse survival, as the growth of the primary tumor in the mammary gland will limit mouse survival before metastatic disease would otherwise lead to mouse morbidity. Although surgical

removal of the primary tumors would seem to be an option, all glands would need to be removed and surgery is itself known to promote systemic inflammation and metastatic progression (PMID: 29643230). We hope that the reviewer will agree that a >100-fold expansion of DCC in the lungs following respiratory virus infection should substantially increase the risk of life-threatening cancer progression (as is further substantiated by our epidemiological studies).

2) T cell Mechanism – The authors infer a lot about T cell function from scSeq gene expression. There are no functional studies to support these data. The only protein data is by flow cytometry in EF10 showing reduced granzyme B in CD8 T cells in Her2 mice versus WT and this finding is not discussed in the manuscript. Conceptually, the proposed mechanism that CD4 T cells impair CD8 T cells post-influenza is counter intuitive and against the general understanding of how helper T cells support CD8 memory formation. What is the antigenic specificity of the T cells examined? Are they influenza specific, Her2, self antigen?

As described above in response to Reviewer 1 Point 2, we have provided new results that substantially bolster our understanding for how the expanding DCC avoid immune elimination, and how depleting CD4 cells promotes DCC clearance. First, we purified CD8 T cells from the lungs of influenza infected MMTV-Her2 mice, and show that these CD8 cells from Her2 mice (but *not* WT) selectively kill mammary tumor cells *ex vivo*, but *only* if CD4 cells had been depleted *in vivo* during the infection (**new Figure 4i-j; new EDF 17**). These results are concordant with our scRNAseq data that show how DCC can reprogram CD4 and CD8 cells towards an impaired effector phenotype (with CD8 cell reprogramming reversed when CD4 cells are depleted). We agree that these results are counterintuitive, with DCC-reprogrammed CD4 cells restricting CD8 cell anti-tumor immune response instead of providing help. These results are consistent with other observations of pro-tumor functions of CD4 cells.

3) IL-6 Mechanism – the findings in IL-6ko are compelling, but no data are provided to infer mechanism by which IL-6 is regulating dormant metastatic cells.

As described in the response to Reviewer 1 (point 1), we now show that treatment of Her2+ mammary tumor organoids with IL-6 promotes organoid growth (**new Figure 3g and EDF 4d and 4e**), demonstrating a direct role for IL-6 in the awakening and expansion of DCC. Also as described above, IL-6 deficiency prevents the phenotypic transitions (increasing EpCam and decreasing vimentin), proliferation and expansion of DCC induced by both influenza and SARS-Co-V2 viruses (**new Figures 3f and 5c-f**).

Minor Comments –

1) Figure 1C – At what time point is the PBS group collected in the Her2 mice?

We collect the PBS group at each timepoint, but since we observe no differences in DCC expansion or phenotype for PBS treated mice at the different timepoints (as expected), we pool the results for the PBS treated mice. We have included this information in the Methods.

2) Figure 3A/3B/3F/EF4 – Quantification of iBALT, colocalization of CD4 T cells and Her2 cells, and CD8 is necessary to support the statements made.

The reviewer raises a pertinent issue. We have now quantified the number of iBALT per lung section, as well as areas stained with antibodies to CD4, Her2, and CD8 within iBALTs and outside iBALTs in **new EDF 5g-h**. Results show that almost all expanding Her2⁺ cells are found within iBALT, with Her2 cells colocalizing with CD4⁺ cells and B cells. There are no differences in CD8⁺ cell numbers within and outside iBALT as CD8 cells are scarce post infection (unless CD4⁺ cells are depleted), as shown in *Fig. 4f*. We also show that iBALT numbers and area are similar for influenza infected MMTV-Her2 and WT mice (**new EDF 5k**).

3) Figure 4 – MA10 model details are missing regarding severity of infection, viral burden, and lung inflammation. This is needed to compare with previous influenza model.

As described in our response to Reviewer 1 (Point 3), we add new results where we infected MMTV-Her2 and IL6 KO MMTV-Her2 mice with mouse adapted SARS-CoV-2 MA10 and analyzed their lungs 3- and 9 dpi. Similar to influenza (IAV) infection, we observed dramatic increases in Her2⁺ DCC cell cycling and expansion post-infection, which were largely prevented in the IL6 KO mice (**new Figure 5**). IF analyses further demonstrate loss of vimentin and transient gain of EpCAM (3 dpi) in DCC post-MA10 infection, and these changes required IL-6. We also analyzed cytokine production in the lungs post-MA10 infection, demonstrating increased levels of IL-6, IL-1 β and interferons α , β and γ on 3 dpi which returned to baseline by 9 dpi (**new EDF 18**). Finally, we analyzed MA10 viral burden post-infection, which was similar for Her2, Her2 IL6 KO, and WT mice (**new EDF 18c**). All of these results are similar to those observed with IAV infection. Given the public relevance of understanding how COVID contributes to cancer progression, these results should increase the impact of the manuscript, providing mechanistic and cause-and-effect insight for our described epidemiological associations.

4) Methods – Details are needed for how the lungs were homogenized for the scSeq study. Also, were lungs perfused for flow cytometry?

Thanks for catching this. Lungs were perfused for flow cytometry, and relevant information has been added to the Methods section.

Referee #3

Chia et al. propose that viral infection such as influenza A or SARS-CoV-2 promotes the awakening of dormant disseminated mammary cancer cells in mice by elevated IL-6 in the lung upon infection. Important for sustaining the newly proliferative cancer cells are CD4 T cells, that block CD8 T cells from killing the micrometastases.

Overall, the manuscript is interesting but lacks many controls and the link to human disease is correlative at best.

We believe the robustness of these findings make the work more than correlative, particularly when the epidemiological associations are paired with cause-and-effect results

from our mouse models. We used two large human datasets (UK Biobank and Flatiron), which independently show that there is an increased the risk of cancer-related mortality and lung metastasis, respectively among cancer survivors after a SARS-CoV-2 infection.

Moreover, that IL6 awaken dormant cells has been reported previously (Werner-Kelin et al, Nat Comm 2020).

We do not believe that this paper (<https://pubmed.ncbi.nlm.nih.gov/33020483/>) shows that IL6 awakens DCC. They show that the IL6 pathway is active in bone marrow DCC, that it promotes sphere-forming ability, that it promotes stemness, and that BM stromal cells regulate gp130 on the breast cancer cells. Regardless, we are not claiming that we are the first to show that IL6 is important for metastasis, only that IL6 is critical for awakening of dormant DCC in the context of a respiratory virus infection.

Major comments:

- What happens with primary tumor progression upon viral infection in these models. How can we be sure metastatic progression over time is not compromised by effects in the primary tumor site? Also, it's unclear if mammary glands (so non-cancerous) were used for downstream analysis throughout the paper (as also described in Figure 1a), as MMTV-models develop several tumors simultaneously.

We thank the reviewer for the comment. We have analyzed mammary glands of IAV-infected MMTV-Her2 mice. Immunofluorescent stain of mammary glands with Her2 and Ki67 showed that there is infection-induced proliferation of Her2⁺ cells in mammary glands (**new EDF 1e and 1f**). Thus, we observe no apparent effect of IAV infection on Her2⁺ cells in their primary site in the mammary glands.

- Can the authors exclude increased dissemination as a mechanism?

The reviewer raises a good point. In response, we have now quantified circulating Her2⁺ tumor cells in isolated CD45^{neg} cells from blood before and after IAV infection using qRT-PCR for transgene expressed rat *ErbB2*, and we observed no differences (**new EDF 1g**). These data indicate that there is no further increase in circulating Her2⁺ cells post-IAV infection.

- Figure 2: Analysis of dormant or proliferating cells is only done by cell number or ki67 staining. Even though authors present relative and absolute quantification in their histological analysis (Figure 2b), it will be more informative to present cell number per metastatic cluster instead of total Her2+ cells.

The reviewer raises an interesting point, but counting cells per cluster is not feasible as we cannot be confident about what delineates a cluster (i.e. derived from an original dormant DCC). As shown in EDF 1d, the DCC exhibit a much more scattered distribution relative to the typical semi-spherical clusters observed with spontaneous awakening of DCC. This distributed pattern post-virus infection is consistent with our new RNA-seq results from DCC, showing how virus-awakened DCC exhibit a hybrid phenotype, and retain and adopt many mesenchymal markers (e.g. MMPs which could contribute to invasive phenotypes) (**new Figure**

2e-h and the associated **new EDF 2**). We do quantify total Her2⁺ cell numbers (see *Fig. 1*), and we now also provide EdU incorporation detection as orthogonal evidence for increased proliferation post-influenza infection (**new Fig. 2b**).

Additional use of dormant cells tracking approaches is recommended to support key findings (lack of EdU incorporation or p27, p21, etc.), especially in the PyMT model. Several dormancy signatures have been reported and could be used to discriminate between dormant cells and micrometastasis.

We have now also done EdU staining for Her2⁺ DCC post-IAV infection, showing very low cycling in dormant DCC in uninfected mice, with increases in EdU⁺ cells post-infection mirroring our observations using Ki67 IF (**revised Fig. 2b**). In our analyses in the PyMT model, we focused on small tumor lesions (<5 PyMT⁺ cells), in order to distinguish IAV-awakened DCC from those that had previously expanded spontaneously. To further support the dormancy model, we have now performed experiments using another independent model (orthotopic model) where EO771 cells were injected into the mammary fat pad. In this model, EO771 cells metastasize to lungs where the cells stay largely dormant for the experimental period (PMID: 37553342). Using this model, we now demonstrate that IAV infection results in dramatic expansion of tumor cell mass in the lung (**new Figure 1g**). Finally, as described above in response to Reviewer 1, we performed RNA-seq on DCC post-IAV infection, revealing the dramatic reprogramming of gene expression from the dormant state towards an actively proliferating awakened state (**new Figure 2e-h** and the associated **new EDF 2**).

Moreover, how do the authors explain that there is almost a complete ablation of Ki67⁺ cancer cells at later timepoints (Figure 2b)? Especially whilst in the total lung, there are still over 100 Ki67⁺ Her2⁺ cancer cells; but they represent almost 0% of the total Her2⁺ population? I would expect that these metastases must, if they're not expanding, at least undergo tumor mass dormancy. So how come the proportion of proliferating cancer cells is so low?

We realize this may have been misleading. There is >100-fold more Ki67⁺ DCC for many weeks after infection than in uninfected Her2 mice. Once infection is cleared, IL-6 levels plummet, thus explaining why most DCC return to quiescence (low *percentage* of Ki67⁺). As shown in *Fig. 3*, the infection induced expansion of DCC is largely IL-6 dependent. But since there are >100x more DCC, even the same low percentage of Her2⁺ Ki67⁺ cells still represents >100x increase in Ki67⁺ DCC *number*. As we show, tumor mass is maintained for many months post-infection. These points are clarified in the revised manuscript.

- It is important to show which cell type in the lung microenvironment increases IL-6, IL-6 secretion upon viral infection.

We agree, and we now show that lung epithelial cells produce IL-6 following IAV infection (**new EDF 3c**). We now show heatmaps for altered mediators of innate immunity in the **new EDF 10a**, including a number of mediators of interferon and TNF α pathways. While we can observe changes in IL-1 α , we did not observe any changes in IL-1 β , likely due to very low expression levels.

Did authors also evaluate IL-6 KO? were the results similar?

In contrast to the high levels of IL-6, very low levels of IL-1 β are produced during the infection with influenza virus (see *EDF 3b*). Therefore, we have not pursued examining the contribution of this cytokine to the awakening of DCC.

- A CD4⁺ T-cells aspect is presented in the paper, but there is not sufficient evidence on how changes in this population affect cancer cells. Addition of the transcriptomic data from the cancer cells is important to support authors' statements.

We agree with the reviewer. It is indeed important to obtain and analyze the transcriptomic data of the cancer cells. We have now enriched Her2⁺ cells by flow cytometric sorting and performed RNA-seq (**new Figure 2e-h** and the associated **new EDF 2**). The results showed proliferating cancer cells with activated IL-6 pathway, anti-viral pathways (e.g. interferon gamma response pathway), dramatic changes in the epithelial to mesenchymal transition pathway (with increases in many epithelial *and* mesenchymal markers post IAV infection, suggesting a hybrid population), increased angiogenesis pathways, and increased expression of collagen/metalloproteinases/ lysyl oxidases, among others. These data show that post IAV infection that DCC are awakened in association with activation of IL-6 pathway, increased angiogenesis, and extracellular matrix alterations. In addition, these analyses reveal potential mechanisms for immune escape by DCC, as we observe upregulation of the gene encoding PDL1 and various collagens (known to promote immune escape).

The authors hint towards a reduced mitochondrial content of CD4⁺ T cells, yet provide no data. A flow cytometry experiment using mitochondrial staining would help support this claim.

Good point. In response, we have performed new experiments using Mitotracker staining and flow cytometry to show that mitochondrial mass is decreased by day 15 post-infection in lung CD4⁺ and CD8⁺ cells from infected Her2 mice relative to lung CD4 cells from WT mice (**new EDF 10g**).

- Data presented in the manuscript mainly focuses on an IAV infection model, and is quite scarce on the COVID model. Given that the clinical dataset is focused exclusively on COVID, the authors should confirm their main mechanistic findings in MA10 model, and present these apart from the later change in the HER2⁺ cells after infection in the Figure 4.

As described above, we have now extensively characterized the SARS-CoV-2 MA10 model, showing similar mesenchymal/epithelial phenotypic shifts, increased proliferation, and increased DCC burden post-infection. Furthermore, we also now show that all of these changes are dependent on IL-6 (**new Figure 5e-g**). We also show increased production of cytokines in the lungs post MA10 (**see new EDF 18**), and we show that transgenic Her2 expression or IL-6 deficiency do not impact viral production (**new Figure EDF 18c**).

- In Extended Data Figure 1b: The number of cells found in the BAL on day 15 are almost double in the Her2 mice compared to the WT (although statistics are lacking). Can the authors

comment on how this finding might skew the analysis of the scRNAseq of the T cells (esp. regarding the CD4 T cell proliferation hint the authors state in line 565)?

The difference is not statistically significant and not reproducible, which we now make clear, and should not affect the phenotypes of lung cells observed by scRNA-seq.

- Where are the Her2+ cancer cells in the scRNAseq? It would be interesting to see the transcriptomic changes occurring in this compartment.

Please see results from RNA-seq for DCC described above (**new Figure 2e-h** and the associated **new EDF 2**). We note that we did not detect sufficient DCC in the scRNAseq data, and in fact we detected few lung epithelial cells, as the isolation protocol selected for mostly immune cell types.

Additionally, the authors do not seem to characterize alveolar macrophages. It would be interesting for the reader to know in which other cell subsets they are divided or included, as they are of crucial importance in lung viral infections.

We have now performed additional analyses of macrophages in the scRNA-seq data, showing that the presence of Her2+ DCC leads to increases in the M2:M1 ratio and dramatically changes gene expression profiles to what appears to be a more tumor supportive phenotype (**new EDF 13**).

- Generally, the scRNAseq dataset hits were nowhere validated on protein level, which should be done (either by stainings or functional assays).

We now validate the differences in mitochondrial mass per cells, the differential expression of Cxcl4 in CD4 cells post-IAV infection in Her2 relative to WT mice, and the differences in Dusp5 protein expression in CD4 cells (**new EDF 10b**). In addition, we now provide functional CD8 cell killing assays described above (**new Fig. 4i-j** and **EDF 17**)

- There is a bias in the Flatiron health database where the authors stratify for COVID-19 infected women. As the SARS-CoV-2 pandemic is recent (4 years approximately), the risk of death by metastases depends on breast cancer subtype (if ER+ or TNBC). Thus, it would be of high importance to stratify the dataset accordingly. Or at least provide additional data on the percentages of subtypes for all the analyses. Moreover, the time of disease detection and their treatment categories matter. Additionally, other potential confounding factors should be included, such as BMI and smoking.

We provide updated analyses from the Flatiron Health database, with multivariate analysis (MVA) stratified with breast cancer subgroups, and regrouping the stage to I-III vs IV to address the missingness. The results were consistent. Thus, correction and stratification of results did not substantially alter the outcomes. These results are reported in **revised Fig. 5b** and **Extended Data Table 4a-d**.

Minor:

- The source of the MMTV-HER2 and PyMT mouse models is not provided.

We now make clear the sources for these models in the methods.

- Methods section on image analysis lacks key information on how many sections were evaluated per sample (only in figure legends), regions selection, steps in the analysis process which led to multiple key graphs presented as «Total cell number» or «Cell number per field». This information has been added to the quantification and statistics section of the Methods.

- Figure 1: To improve figure reading, include 15 dpi label to (e) graph. The changes have been made. (f) Add information on dpi in the PyMT model. Why quantification of the lesions was limited to certain size? Apart from providing the number of metastatic lesions (Fig. 1f), it would be good to have the total metastatic burden (as % of cancerous area per total lung area). This is crucial to see how much under control those virally induced metastases are after the infection is cleared.

The PyMT model exhibits substantial spontaneous metastatic outgrowth, and thus we do not expect a change in total metastatic area. As such, we analyze small (micrometastases, defined as lesions with an area $<0.03 \text{ mm}^2$) lesions for measurement of awakening in the short period following viral infection. We have clarified the specifics of this model in the Methods. To further address this concern, we introduced a new model, using orthotopic implantation of EO771 cells into mammary glands, a model of metastatic dormancy in the lungs. Using the EO771 model, we show that influenza infection promotes a dramatic increase in DCC proliferation and a substantial increase in the total lung tumor burden (new Fig. 1g and EDF 1h-i).

- Figure 1: Which statistical test was used for 2 group comparisons? the multiple test correction for One-way ANOVA should be indicated. Moreover, statistical tests are missing in some of the figures (Extended Data Figure 1,2, 3a).

Thanks for catching this. Statistical tests are indicated in all figures and EDFs.

- Extended Figure 1d: if the authors could circle the area they consider as metastases, that would aid in composition recognition. Could it also be, that the metastases just have a different immune infiltration or grow out in different environments (alveolar vs. intra epithelial)?

We now show that expanded Her2 DCC are specifically localized in iBALT post-IAV infection (new EDF 5k). Note that expanded Her2 DCC do not form a tight cluster, but are distributed throughout the iBALT (see EDF 2d), and thus it is not easy to identify a particular cluster of common origin (i.e. from an original dormant DCC). But DCC are easily identified by IF for anti-Her2, as shown in multiple figures.

- Why do the inoculation methods differ between influenza and SARS-CoV-2?

We are following published protocols for each. Since the MA10 model was already established in the Morrison lab, we decided to stick with their inoculation method as the kinetics and character of infection were already established. Nevertheless, both protocols use intranasal delivery as the inoculation method.

- Please refrain from the word “FACS” buffer, as it is a trademark by BD. Any other appropriate label will do.

“FACS” buffer has been changed to “flow” buffer.

- Line 192: were the lungs incubated shaking or still? And was the leftover tissue manually homogenized? How long was the hemolytic buffer applied?

Lungs were incubated with shaking, and residual tissue went through vigorous vortexing to break up visible pieces. Hemolytic buffer was applied for 3 minutes at room temperature. Relevant information has been added to the Methods section.

- Line 246: Amount of RNA reversely transcribed is missing.

Thanks for catching this. 1µg RNA was reverse transcribed. This information has been added to the methods section.

- Line 402: Please capitalize the V from FVB.

“V” in FVB has been capitalized throughout.

- Figure 2: unclear which cells the arrows indicate in the figure legend. Also, why is dpi 60 missing in the graph (e)?

The arrows indicate the cells expressing Her2, so that expression of EpCam or vimentin in the same cells is evident. 60dpi has been added to graph (e), now (Fig 2d).

- Extended Data Figure 1b+c: Unclear about the number of mice used. Taken into account that a lavage does not always yield the same recovery of buffer injected, and additionally that the lavage was only performed once with 1mL, this data needs to include single mouse datapoints instead of just a bar graph.

The graphs have been changed to show single mouse datapoints.

- In the methods section the processing of the BALF is only described by centrifuging and resuspending. Was any erythrolysis performed on these samples?

Erythrolysis was performed on these samples, and this information has been added to the Methods section.

Which cell types are included in the analysis? The numbers are very high, especially comparing to the immune cell counts from Extended Data Figure 4, which are much lower.

The prior *EDF 1b* included all cells from the BAL, and the prior *EDF 4a-d* analyzes only the indicated cell types. We likely experience cell loss during processing for antibody staining and flow cytometry.

- Extended Data Figure 1d: Please increase the labeling size of the scale bar. The size bar is now labeled.

- Figure 2: The authors conclude that «a hybrid state allows for dormant DCC awakening» based on their results. They show that % of vimentin+ cells are not significantly altered upon IAV infection in IL6 KO mice. While focusing further on the mechanistic part is not a goal of this manuscript, could you elaborate on this connection and place your findings into the scope of current knowledge in the dormancy field in the discussion?

As described above, we now have substantial additional characterization of phenotypic transitions in DCC following respiratory virus infection (influenza and SARS-CoV-2). In addition

to IF characterizations of the loss of vimentin and transient gain of EpCam expression post-infection, and the dependence of these changes on IL-6, RNA-seq analysis of DCC demonstrates a truly hybrid epithelial/mesenchymal state post-infection. We add discussion for how this hybrid state is quite different from that observed in previous studies of spontaneous DCC awakening (e.g. PMID: 36050483).

- Extended Figure 2: Please add a “n.d.” or equivalent to samples where you were not able to measure the cytokines.

“n.d.” has been added to samples below detection limit.

- The authors state (line 519+520), that CD8+ T cells are increased upon CD4+ T cell depletion, however the only data provided is one microscopy image. Thus, the authors should provide flow cytometry analysis of whole lung digest for a proper data representation. Alternatively, as the authors subsequently provide CD8+ T cell depletion data, they could simply remove this statement.

We now quantify CD4+, CD8+ and B cells within iBALT and outside of iBALT post-IAV infection (**new EDF 5g-j**).

- Lines 533-536: The sentence references to a wrong figure. Unclear about the message the authors are trying to make.

Thanks for the catch. This has been corrected.

- Extended Figure 3a: Please indicate in the figure legend or methods how many tumors the mice developed on average. Were any mice excluded from these analyses? Also, it is unclear how many mice were used for the Kaplan-Meier plot.

No mice were excluded. 4 mice per group were used for the Kaplan-Meier plot; these mice usually develop 2-4 large mammary gland masses and are euthanized per IACUC protocol (at 2 cm diameter).

- Figure 3b: The authors claim that they see almost no Her2+ cells when there’s no CD4+ T cells, but all the data provided is just each one image, with one lacking a DAPI staining. Therefore, the authors should provide more data. If the intention was to show the same image as the upper panel of Fig. 3b, then that should be stated. Also, are there any/more CD8 T cells present in those areas?

Quantitation of Her2+ cells in the lungs of mice with depletion of CD4+ cells is shown in *Fig. 4d,e*, showing a ~100X reduction relative to control MMTV-Her2 mice (also infected with influenza). We also quantified CD8+ cells in iBALT relative to the lung parenchyma, showing similarly low numbers in both (**new EDF 5i**). In addition, we have now quantified Her2+ DCC within and outside of iBALT, showing that the vast majority of DCC are localized to iBALT post-infection (**new EDF 5j**).

- Whole figure 3: Please enlarge the scale bars on all microscopy pictures. **Requested changes have been made.**

- Extended Data Figure 3: The figure legend mentions that the criteria for endpoint was tumor size in «a». Please clearly label it in the Figure «Days to maximum tumor size» instead

«Probability of Survival». Requested changes have been made.

- Extended Data Figure 3b: Why are the authors using a mix between bar graph and SD lines? Please unify.

The graph has been changed to Box-and-Whisker plot with individual points shown, consistent with other graphs.

- Extended Data Figure 4a-d: Please add how many animals were analyzed.

3 mice per group were analyzed, and relevant information has been added to the figure legend.

- Extended Data Figure 5: The authors are asked to provide previous gating strategy above the T cell gates for the samples. Additionally, what message should the third column be conveying? CD4 and CD8 gates are already included in the second column. Arrows pointing from one gate to the subsequently gated should be added. Please capitalize "Lung", and unify that fluorochromes are labelled everywhere on the axes, as well as that the axes are aligned.

This is now *EDF 7*. The second column gated on CD4 (x-axis) and CD8 (y-axis); the third column gated on CD3 (x-axis) and CD8 (y-axis), showing that few CD3⁺ cells that are negative for both CD4 and CD8 persist (in case the antibody was just preventing detection of CD4). This explanation was added to the figure legend. Arrows have been added to graphs indicating the order of gating strategies.

- Figure 3/Ext. Figure 6: Some images have a double scalebar and too small labeling of its length. Please correct for consistency. Addition of an experimental set-up schematic will help readers understand results in «c-i». Requested changes have been made.

- Extended Data Figure 6: Why only data on ki67⁺ Her2⁺ cells for aCD4, but not aLy6G presented? Scalebar information of the images is lacking in the figure legend.

We now include the Ki67 staining results for the anti-Ly6g depletion, showing that depleting neutrophils does not impact Her2⁺ cell proliferation post-infection.

- Extended Data Figure 7c: Unclear which sample is shown in the UMAP. The same UMAP should be shown for all sample types (WT vs MMTV).

The UMAP shown in *EDF 9a* (left) represents the aggregate for all samples, but individual UMAPs for different samples and replicates are now shown in **revised EDF 9a**, right.

- Any reason that only human observational data on Covid and not on influenza are provided?

Influenza infection and vaccination information are not available through linkage in the UK Biobank study nor in Flatiron. This holds true for many studies as most influenza cases are not recorded in hospital records and there is no systematic Flu testing.

- Authors should discuss limitations of their study.

We now more fully discuss the limitations of our study in the Discussion.

Referee #4 (Remarks to the Author):

A) The manuscript "Respiratory viral infection promotes the awakening and outgrowth of dormant metastatic breast cancer cells in lungs" demonstrates a mechanism by which

influenza A virus infection causes progression of dormant metastatic breast cancer. A linked set of observational epidemiologic analyses show increased risk of progression among cancer survivors following SARS-CoV-2 diagnosis in the UK Biobank and a US EHR dataset. The degree to which these findings strengthen and support the epidemiologic relevance of the primary finding, however, depends upon the strength of the evidence provided by the authors that SARS-CoV-2, in the mouse model, is causing a similar phenomenon to IAV. More on that below.

B) To my knowledge the findings are highly novel and of great importance across numerous fields of research, thus meriting the attention of the wide scientific community that reads Nature.

We thank the reviewer for recognizing the value and impact of our studies.

C/D) My area of expertise is epidemiology and I focus on the analysis of the UK Biobank and US EHR data as I do not have the necessary knowledge to review the mechanistic experiments.

UK biobank study: Confirm if the matching procedure is matching those who test positive within 2020 pre-vaccine with those who have at least one negative test over this period?

We thank the reviewer for this important comment. As detailed above, for our original submission, we matched test positive cases with any participant with no positive test, as test negative information was not available back then. In the current revision, we use the new UK-Biobank data release from February 2024, which now includes test negative results. We are now matching test positive with test negative participants using the matching strategy described in the Methods. Results based on test positives matched to test negatives (new analyses) vs. those from positives matched to non-positives (ie test negatives and non-tested, as in our previous submission) are very comparable (see below). We now only report the former in the present revision. The new data and matching procedure are described in the Methods (see p20-23).

	Positives vs non-positives (i.e including non-tested)		Positive vs negatives	
	OR (95% CI)	p	OR (95% CI)	p
All-cause mortality	4.38 (3.59, 5.34)	5.6E-48	4.50 (3.49, 5.81)	6.88E-31
Cancer-specific mortality	1.72 (1.25, 2.37)	9.28E-4	1.85 (1.14, 3.02)	0.013
Non-COVID-19 mortality	2.21 (1.73, 2.82)	2.7E-10	2.56 (1.86, 3.51)	6.95E-9

The authors refer to test positive and test negative individuals but I do not see an explicit criterion for the controls as having needed to test negative; presumably this is being done to match on healthcare seeking behavior?

We recognize that the language of test positive and test negative may have been misleading. We now compare test negatives and test positives (we no longer consider non-tested participants), which should clarify our statements throughout.

The concern that arises in such studies is the possibility that sicker people are being tested more often and thus have more chance for a positive SARS-CoV-2 test to occur (or to be documented) than those who are relatively healthy and at lower risk of experiencing the outcome. To further alleviate concerns about missed infections, it would be of value to determine if any control subjects are available who definitely (based on results of several tests at a minimum interval like 60 or 90 days) remained uninfected throughout the period of interest.

As clarified above, we now compare test positives and test negatives. While effect size estimates are slightly attenuated in these new analyses, conclusions remain unchanged. We showed marked similarities in the distribution of key risk factors in test positives and test negatives (see response to point #5 of Reviewer #1). In addition, we adjusted our analyses for comorbidities. This should capture the concern about differential health care seeking behavior (both in testing for SARS-CoV-2, and for cancer). Our conclusions remained unchanged, still identifying an increase in mortality in test positive cancer survivors.

Relatedly, people seem to be followed up until 2022 for cancer outcomes in this analysis. By mid-2022 over half of the US and UK populations had experienced SARS-CoV-2 infection so it does not make sense to use their infection status pre-vaccine (2021) only as the exposure; this is likely misclassified by 2022 for many or most people. The infection and cancer outcomes should be evaluated over the same period of time, or a shorter maximum length of time after COVID vaccination (<<<1.5 years) should be used to reduce this problem.

The reviewer raises an important point. We purposely limited the infection status to pre-vaccination because: (i) until vaccination there were only a limited number of strains circulating in the UK population (wild-type, alpha), and (ii) vaccination may alter the primary response to the virus and thus may also influence the virus-cancer association. As is common in epidemiological analyses of cancer, we then followed up the group until 2022. This date was chosen as the latest date with full follow-up data available and ensuring the longest possible follow-up period. Individual infection status is likely to have changed after 2021. We assume that this would be non-differential across the two groups and therefore would not have affected our analyses. However, to address this issue, we present below an extended analysis considering shorter follow-up times. For the three outcomes of interest (all-cause, non-COVID and cancer mortality), we find that restricting the follow-up time gradually by 6 months increases the strength of association with risk of cancer mortality ranging from 1.85 (95%CI 1.14, 3.02) for the original follow-up time (censoring date of Dec 31, 2022) to 8.24 (95% CI 3.43, 19.77) for a censoring date of Dec 1, 2020. These results are now reported in the **revised Figure 6a** and **Extended Data Table 3**. These new results show that excess cancer-related mortality is greater close to SARS-CoV-2 infection, which parallels the experimental results.

Fig. 6a. Sensitivity analyses comparing cancer mortality in cancer survivors (with cancer diagnosis > 5 years before the start of the COVID-19 pandemic) with a positive vs a negative test for SARS-CoV2 from Jan 1, 2020 to Dec 1, 2020. We considered several censoring dates for the death events ranging from 01/12/2021 to 31/12/2022.

The Flatiron EHR analysis does not make any such requirement for a negative test. Because these people were participating in cancer cohorts is it fair to assume all received regular testing?

Information on negative test data is not available from Flatiron Health, which is a limitation that we now report in the last section of Results. Although we agree that cancer survivors likely received regular testing, we do not have confirmation of this. We are only provided with the COVID-19 diagnosis from the diagnosis code.

For the cox proportional hazards model, what was the unit of time?

The unit of time is in months, which we now make clear. Please note that the hazard ratio is invariant from unit of time.

Did observation windows continue until a COVID-19 diagnosis was made or was there a specified unit?

The observation window for the COVID-19 test is for as long as there is activity records in the Flatiron database. The same applies for our survival outcome.

The adjustment set makes no reference to comorbid conditions or healthcare-seeking behavior and thus appears woefully inadequate for EHR-based real world evidence studies. I note that adjustment for comorbidities was also missing in the UK Biobank studies although at least these matched on cancer type and included more covariates in the adjustment set.

The UK Biobank study is matched on comorbidities, as described above.

We provide updated analyses from the Flatiron Health database, with multivariate analysis (MVA) stratified with breast cancer subgroups, and regrouping the stage to I-III vs IV to address the missingness. The results were consistent. Thus, correction and stratification of

results did not alter the outcomes. These results are reported in the **revised Figure 5b** and **Extended Data Table 4a-d**.

We also addressed the potential for differences in comorbidity between groups could impact outcomes. We computed the comorbidity score using Elixhauser comorbidity index. Cause specific analysis was conducted (death was censored). Stratified Cox proportional hazard model with stratification factors stage, year of diagnosis, and age group were used to evaluate the effect of COVID-19 diagnosis on the risk of metastases to the lungs, while adjusting important covariates at initial diagnosis. The adjusted hazard ratio (HR) with the corresponding 2-sided 95% confidence interval was reported. Although we lost some power due to reduced cohort sizes (given censoring of individuals with insufficient data), the hazard ratio (HR) remained similar although significance was reduced: HR (multivariate) of 1.46 (0.99-2.15, $p=0.057$). These results are reported in the **revised Fig 6b**.

E) The significance of the COVID-19 epidemiologic study linkage is dependent upon whether a similar mechanism to that seen with IAV occurs with SARS-CoV-2. At present the applicability/interchangeability of the IAV and SARS-CoV-2 findings seems limited; a single sentence states that “Infection of MMTV-Her2 (C57BL6/J) mice with MA10- SARS-CoV-2 resulted in a striking increase in Her2+ 595 cells by 28 dpi”, and I am unsure of whether this biomarker is sufficient to demonstrate that IAV and SARS-CoV-2 are doing the same thing. If not, I am less convinced that the epidemiologic study is adding something of importance to the manuscript.

As described above in response to other reviewers, we have performed extensive additional analyses of the MA10 SARS-CoV-2 mouse model, showing that MA10 infection results in similar increases in proliferation, state changes, and metastatic outgrowth, each dependent on IL-6, as previously observed for influenza infection. We also measured cytokine levels post-MA10 infection, showing the expected increases in IL-6, IL-1 β and interferons. These data are presented in the **revised Figure 5** and **EDF 18**.

F) My suggestions for revision are contained in my statistical review (C/D).

G) I believe appropriate references to previous work are made.

H) The manuscript is clearly written.

We sincerely thank the reviewers for their thoughtful critiques, which have led to a substantially more robust demonstration of links between respiratory virus infections and metastatic progression, with a more thorough understanding of mechanistic underpinnings. Considering the millions of cancer survivors who have or will experience respiratory virus infections such as influenza or SARS-CoV-2, and thus the huge potential impact of this work, a rigorous demonstration of links and their underlying mechanisms is essential, which we believe that this revised study provides. We thank you for consideration of our revised manuscript.

We have provided responses to the critiques below. A few points that we would like to emphasize are that: 1) the impact of these studies is huge, as we demonstrate a clear ability of respiratory virus infections (including COVID19) to promote rapid metastatic awakening and progression of disseminated cancer cells (DCC); 2) these discoveries are rigorous and reproducible across multiple mouse models and striking epidemiological data; 3) we provide extensive mechanistic insight into this virus-induced awakening, showing that awakening, phenotypic changes, and expansion of DCC require IL-6, and that DCC reprogram CD4 T cells to a suppressive state that prevents CD8 T cell mediated DCC elimination; 4) these studies have a large impact on public health policy including cancer screening of vulnerable groups and the development of interventions, which is particularly timely given the current denigration of the value of vaccines and viral research in general. Finally, the magnitude of these effects is also worth stressing, with 100-1000-fold increases in metastatic burden post-infections in the mouse model, and ~8-fold increase in risk of death from cancer in the first year of a COVID positive result for the UK Biobank analyses. Such effects are really unprecedented.

We believe that we were highly responsive to the previous round of reviews, thoroughly addressing the points raised with extensive new data (essentially doubling the results presented). We further posit that addressing some of the points raised by this latest round of review (not brought up earlier) would unnecessarily delay the publication of a study with great public, medical and scientific importance. At this stage, we believe the most constructive next step is for other research groups to independently replicate and build upon our findings. Our specific responses are below in black, with Reviewer comments in blue. All new text (excluding minor changes) in the marked-up version of the manuscript are highlighted in yellow. Thanks so much for your shepherding of this manuscript over the last year.

Referee #1

After reading the revised version, this reviewer feels that the authors **have largely satisfied reviewers**. In the revised version, the authors provided mechanistic analysis (to be honest, not in-depth enough, mostly at the level of phenomenological description) and suggest that the IL6 (together with others) probably played major role in awakening and expanding disseminated cancer cells. We agree with the statement (emphasis added by us) that we have largely satisfied the reviewers comments, but would argue strongly that what we are showing is not phenomenological. We show that respiratory viruses *promote* DCC awakening, and we provide clear dependencies on IL-6 and T cells at multiple levels (proliferation, survival and phenotypic transitions) and in multiple contexts (different mammary tumors and different viruses). Regarding IL-6,

while the reviewer says that we "**suggest** that IL-6 **probably** played a major role in awakening", our data clearly **demonstrate** that IL-6 is essential for awakening and expansion. The number of Her2+ cells in IL-6 KO MMTV-Her2 mice following influenza virus infection is about 5-fold lower than in WT mice (**Fig. 3a-d**). In addition, we show that as early as day 3 post-infection, WT MMTV-Her2 mice show induction of HER2+/EpCAM expression on ~70% of DCC; this induction is reduced to almost zero in DCCs in the IL-6 KO MMTV-Her2 mice (**Fig. 3f**). We show a similar dependency for reductions in vimentin post-infection on IL-6 (**Fig. 3e**). Thus, we **directly** demonstrate that IL-6 is essential for the awakening and acquisition of the proliferative epithelial phenotype of dormant DCC post infection with influenza.

Important for the findings in humans, in the revised version we also **demonstrate** that IL-6 is **essential** for the awakening of dormant DCC (and their phenotypic transitions) in the lung following the infection with mouse SARS-CoV2 (MA10) virus (**new Fig. 5d and 5f**). These data were added to the revised version of the manuscript.

This reviewer has a final request, that is: could the authors provide a therapeutic strategy to prevent (breast) tumor progression caused by respiratory virus infection, and can they conduct experiments at the animal level to verify it? While certainly of interest, we note that such a request was not made in the original reviews, and that this task would take at least a year given realities. While in humans, tocilizumab (IL-6R blocking antibody) works well and is approved for treatment of COVID and several autoimmune diseases (e.g. rheumatoid arthritis), there are no effective anti-IL-6R antibodies for mouse available (in part due to IP rights). In the past for other projects, we have tested a number of these antibodies and anti-IL-6 Abs without success even by ELISA. Moreover, developing interventions that work in animal models is a massive and uncertain undertaking, and successful results could fill a manuscript by itself. We would need to not only discover such an intervention, but generate the animals to test it, figure out a dosing schedule, determine efficacy, assess undesirable effects (toxicities, reduced recovery from virus infection, etc.), and more. This is in our view beyond the scope of our work.

Referee #2

The authors have adequately addressed my prior concerns and have added new data and analyses to strengthen the manuscript. We thank the reviewer for considering our revisions adequately responsive.

Referee #3

The revised version includes additional data that have improved some aspects of the manuscript. In particular, the cell line, MA10 COVID infection model, RNAseq of cancer

cells post infection and analysis of the patient data. Nonetheless, the link between IL-6 and dormant cancer cell exit from dormancy is not fully addressed in the revised version. We have shown that IL-6 is *required* for mesenchymal/epithelial phenotypic transitions in DCC post-infection, for cell cycle entry, and for DCC expansion, and we've shown this in multiple models (MMTV-Her2, MMTV-PyMT) and for both influenza and SARS-CoV2 (the latter in the revised manuscript). In addition, recently (after resubmission of the study) we have performed additional experiments with the EO771 mammary tumor line implanted in WT and IL-6 KO mice, again showing that IL-6 is essential for the development of the virus-induced lung metastases in this model. We have added these new data to the manuscript (Extended Data Figure 3d,e), which are also shown below in Figure R1.

1.

Figure R1. (a) 2×10^5 EO771 mammary tumor cells were implanted into the mammary fat pads of WT or IL-6 KO C57Bl/6 mice, and infected with IAV (or PBS control). Lungs were harvested 12 days after infection, stained with H&E, and tumor area and the numbers of lesions quantified.

Moreover, the data to support the effect of COVID-19 on increased lung metastasis of breast cancer patients are weak. We would strongly disagree that these results are "weak". The epidemiological data from two independent cohorts present striking and consistent findings. If the concern is related to statistical significance, it is important to recognize that significance depends on both effect size and sample size. The FLATIRON data indicate a 40% increased risk of developing lung metastases in breast cancer patients, which aligns closely with the risk observed in the UK Biobank for all cancers (RR = 1.85 [1.14, 3.02]). Moreover, with greater statistical power in UK-B, we were able to analyze the etiological time window, revealing that this risk ratio increases to 8.5 when restricting follow-up to the first year after infection. An 850% increase risk can hardly be regarded as weak.

- Figure 3g and Extended Data Figure 4 new graphs show a mammosphere assay results. This assay only implicates that established organoids acquire faster proliferating rate upon stimulation with IL-6, but not a direct switch from dormant state to proliferation. Organoids do not accurately model the dormant state of DCC as the *in vitro* conditions promote fast growth. Further, we argue that in the field there are currently no sufficiently accurate models of dormancy *in vitro*. We thus used treatments of mammospheres starting 24 hrs after seeding when cells are still solitary, like DCCs. This assay has been reproducibly used to test predictions of signaling and

transcriptional mechanisms that were informed by or subsequently validates *in vivo* (Harper and Sosa et al., Nature, 2016; Hosseini et al., Nature 2016; Linde et al., Nat Commun 2018; Nobre et al., Nat Cancer 2022; Dalla et al., Cell 2024). Here, we used this mammosphere assay to demonstrate that the effects of IL-6/sIL-6R on Her2+ mammary tumor cells can be direct, and that these cells are responsive in a solitary state, recapitulating at least one aspect of the IL-6 dependent response following virus infection that we observed *in vivo* (increased proliferation). We have added the following new text to Results:

Although *in vivo* solitary DCC dormancy in the lung alveoli^{15,61} cannot be fully replicated *in vitro*, mammosphere initiation assays from single cells to proliferative clusters have been used to study how gene perturbations or treatments affect single cell growth initiation or growth arrest^{54,61}. Despite its limitations this assay confirmed that IL-6/sIL-6R directly affects Her2+ mammary tumor cells in a solitary state, which reproduced the IL-6 response observed *in vivo*.

An *in vitro* model including a co-culture between cancer cells and potential cell type(s) promoting IL-6 burst after viral infection can better mimic *in vivo* observations of the authors. Such an *in vitro* model that could incorporate influenza virus or SARS-CoV2 infection, different lung cells (MSCs, AT1, AT2, endothelial or alveolar macrophage cells), and tumor cells does not exist as far as we know. It would require obtaining multiple cell types at different times post-infection and this would be a separate paper by itself. However, our mammosphere assay from single cell transition to colonies has been reproducibly used and rigorously benchmarked *in vivo* (Harper and Sosa et al., Nature, 2016; Hosseini et al., Nature 2016; Linde et al., Nat Commun 2018; Nobre et al., Nat Cancer 2022; Dalla et al., Cell 2024) as in the current study. Importantly, our *in vivo* results already demonstrate the complex interplay between viral infection, DCC, and the immune system, and how each of these affect the others (virus induced DCC awakening through IL-6 produced from lung epithelial cells, the ability of DCC to reprogram CD4 cells which then suppress CD8 cells, etc.). The *in vivo* model allowed us to manipulate each of these factors (presence or absence of virus, DCC, IL-6, CD4s and CD8s) under a highly physiological context, providing a powerful means to dissect and validate the underlying mechanism that is only partially possible with an *in vitro* model. Importantly, a similar awakening role for IL6 was reported (PMID: 33020483) using *in vitro* assays, further supporting that our choice of assays is reproducible and rigorous.

Additional stainings for proliferation and dormancy markers in a time course fashion, as well as treatment conditions blocking IL-6 signalling are necessary to clarify the mechanism. The authors can further explore the exact mechanism by changes in the cancer cells observed from RNAseq — changes in epithelial and mesenchymal markers, upregulation of extracellular matrix proteins like collagens, and acquisition of

an immune escape phenotype (upregulation of Cd274, which encodes PDL1) and decreases in B2m (required for antigen presentation by MHC Class 1). While performing more immunofluorescence and analyses of both the in vitro and in vivo models could be done, we argue that such additional studies will not significantly change the overall conclusions or impact of this manuscript, and will only cause additional delay. Furthermore, we show that awakening of DCC occurs early post-infection (at least day 3), and it is well established that T cells are not recruited to the lungs at least until day 7-8 post-infection for influenza virus (see our Ext Data Figure 4b-c). Thus, expression of PDL1, MHC-I is likely not relevant for awakening triggered by IL-6.

- The authors propose IL-6 blocking strategies for preventing DDC awakening in cancer survivors, yet do not show pre-clinical data that suggest this would be beneficial in vivo. As noted above, this would take years, and was not requested in the first round.

Minor:

- Please increase the mouse number per group in data presented at least to n=3 (new Figure 1g). We present two experiments each with an n=2, so a combined n=4 per group. We ran statistics comparing the fold change for PBS versus influenza across the two experiments, and the differences are significant. See Fig R2 below, which has been incorporated into Fig 1g in the manuscript. In addition, we now include a new experiment showing that the increase in EO771 burden following influenza infection is IL-6 dependent (Fig R1 above).

Figure R2. EO771 mammary tumor cells were implanted into the mammary fat pads of C57Bl/6 mice, and infected with IAV (or PBS control) after 31 days (Exp 1) and 20 days (Exp 2). The mice were implanted with 2×10^5 (Exp 1) or 1×10^6 EO771 (Exp 2) cells across two experiments, and combined results are shown. Lungs were harvested 18 days (Exp 1) and 17 days (Exp 2) after infection, stained with H&E, and tumor area and the numbers of lesions quantified. For each experiment, the average of the quantitation of the PBS treated mouse lungs was set to 1, so that a fold change could be calculated.

- In Figure 6a please add «years» to the graph labeling with «5,10». Added.
- In Extended Data Fig. 1 is missing indication of n mouse numbers, as well as a statistical test indicated «one-way ANOVA» can't be implemented for 2 groups comparisons (also applies to Figure 1). Please correct. Corrected.
- Regarding Mitotracker staining, adding flow cytometry antibodies for 5 min at the last incubation would very likely not yield enough staining intensity (generally 30 min are

recommended unless properly assessed). What are the reasons for this short incubation? Could the authors show that the MFI of the CD4 and CD8 epitopes are not altered upon IAV infection? The reviewer is correct that the normal staining for CD4 and CD8 (or any cell surface staining) is performed for 30 min at 4 C. However, in this case Mitotracker staining needs to be done at 37 C to properly detect mitochondria. Thus, our standard protocol is to add the anti-CD4 and anti-CD8 Ab during the last 5 min at 37 C. As expected, 5 min is more than enough for detecting CD4 and CD8 at 37 C (faster on-rate at higher temperatures). See Figure R3 below, which has been incorporated into Ext Data Fig 7h in the manuscript, which demonstrates that the 5 min staining protocol at 37 C with anti-CD4 and anti-CD8 results in clearly defined populations of these T cells from both WT and Her2 mice. For clarification, we also provide additional details regarding the staining with Mitotracker plus antibodies in the Methods.

Figure R3. Flow cytometric analysis of CD4 and CD8 expression of cells isolated from the in lungs of 15 dpi IAV infected WT or MMTV-Her2 mice and stained for Mitotracker and anti-CD4 plus anti-CD8. The Mitotracker intensity shown in Ext Data Figure 10g is for CD4 and CD8 cells gated as shown here.

• Discussion is not containing the study limitations, as the authors state in the point-by-point. Our apologies, as study limitations are in both Results and Discussion. In the Results section on the analysis of the UK Biobank data, we state: “Stratification of results based on primary tumor type and metastatic disease was not possible due to an insufficient number of observations”. In the Results section on the analyses of the Flatiron datasets, we state “We note that we lack information on COVID test negativity, and thus the COVID-19 negative cohort likely contain individuals who contracted COVID-19, which should contribute to underestimation of the HR.” Finally, in the Discussion, we refer to limitations in the mouse models: “While differences in immune responses and other factors suggest caution when interpreting results from mouse models...”. We also make it clear that the epidemiological data alone would be insufficient to identify the cause: “The cause of the increased cancer mortality is not known, but based on the mice experimental data, a re-activation of dormant cancer cells may play a role.” We have

also added the new text indicated above concerning the limitations of mammospheres (to Results), and additional limitations of the epidemiological studies in response to Reviewer #5 below.

Reviewer #5

Test-negative “controls” are not necessarily individuals who have never experienced a SARS-CoV-2 infection. This is acknowledged in the Flatiron analysis but not in the UK Biobank analysis. I think any bias introduced by this is towards the null.

In the UK Biobank data, we only selected participants with valid PCR test results through (consented) linkage to NHS records. As such, all included participants would have had at least one valid test during targeted exposure period from 01/Jan/2020 (assumed start of the pandemic in the UK) to 01/Dec/2020 (assumed start of the vaccination roll-out in the UK). Individuals classified as test-negative were those with a documented negative SARS-CoV-2 test prior to the introduction of COVID-19 vaccination. In line with our primary research objective to investigate the effect of the first exposure to SARS-CoV-2 virus on subsequent health, we compared these test negatives to matched test positives as defined by cancer survivors having been infected during the exposure period.

The reviewer is right that between the end of the exposure period (01/12/2020) and the event occurrence or censorship (death or end of follow-up at 31/12/2022), test negatives may have been exposed to SARS-CoV-2. The resulting exposure misclassification would likely bias the estimated associations towards the null, resulting in an overall underestimation of the true effect of SARS-CoV-2 infection on cancer/metastasis risk. This has now been clarified in the results section; see page 35:

“Some individuals in the test-negative group may have become infected during follow-up. These evolving differences in infection risk over time may have contributed to a weakening of the observed association. Still, our findings suggest that the increased risk of cancer mortality is greatest in the first few months after SARS-CoV-2 infection.”

- The increased effect size for shorter follow-up periods could be due to time-varying effects on mortality risk, but could also be due to increased bias by making the infection profile of the test-positives and test-negatives more similar, as the probability of a test-negative accruing an infection that was not tested is higher. The authors introduced this

sensitivity analysis to respond to a concern of a previous reviewer about misclassification of the exposure but are interpreting the results of this sensitivity analysis as representing a biological effect. It should be acknowledged that misclassification of the outcome (which is differential between test-positives and test-negatives) is likely contributing to the attenuation of the ORs.

We thank the reviewer for raising this subtle point. We agree that the attenuation of the effect size estimates while increasing follow-up time may not solely reflect a biological effect, but could also result from differential misclassification of exposure. As described in the point above, test-negative individuals are more likely to acquire unrecorded SARS-CoV-2 infections during the follow-up period compared to test-positives, who are less likely to be re-infected, notably due to post-infection short-term immunity. This differential misclassification, which may increase with the length of the follow-up period, will contribute, along with possible biological processes to the observed attenuation of the odds ratios.

In response to the reviewer’s concern regarding exposure misclassification, we conducted a sensitivity analysis using nested 6-month incremental follow-up windows, as reported in our previous submission (Figure 5d and Extended Data Table 3). To further disentangle potential effects of time-varying exposure misclassification from biological plausibility, we repeated the analyses using discrete, non-overlapping 6-month follow-up intervals (see Table below).

	01/01/2020 to 01/12/2020	01/12/2020 to 01/06/2021	01/06/2021 to 01/12/2021	01/12/2021 to 01/06/2022	01/06/2022 to 31/12/2022
All cause	14 (9.5-20.8)*	5.7 (3.1-10.4)*	1.6 (0.8-3)	1.2 (0.5-2.8)	2.6 (1.3-4.9)*
Cancer	8.2 (3.4-19.8)*	2 (0.7-6)	1.1 (0.3-3.8)	0.3 (0-2.5)	1.5 (0.4-5.2)
Non-covid19	3.3 (1.8-6.1)*	3.6 (1.9-6.8)*	1.8 (0.9-3.4)	1.1 (0.5-2.6)	3.4 (1.7-6.7)*

Although more limited by case numbers relative to our previous analyses, we observed a clear gradient in effect size estimates, with the highest odds ratios seen in the initial period (2020) and earliest follow-up windows across all outcomes. For example, the odds ratio for all-cause mortality was 14.0 (95% CI: 9.5–20.8) during the first 11 month period, decreasing to 5.7 (95% CI: 3.1–10.4) in the subsequent six months. A similar trend was seen for cancer mortality, with an OR of 8.2 (95% CI: 3.4–19.8) initially, dropping to a non-significant OR of 2.0 (95% CI: 0.7–6.0) in the subsequent six months. This gradient is more apparent than in our previous analysis using overlapping follow-up windows, suggesting that the observed association between SARS-CoV-2 test positivity

and increased mortality may be driven by events occurring within the first year post-infection. Longer-term estimates appear attenuated, potentially reflecting both a waning biological effect and increasing exposure misclassification over time.

Based on these findings, we acknowledge that while time-varying biological effects may contribute to the pattern, differential exposure misclassification—particularly among test-negative individuals—likely plays a role in the attenuation of associations over time. As detailed above, we have revised the manuscript to incorporate this more nuanced interpretation and are grateful to the reviewer for prompting this important clarification.

- In the methods, the distinction between the two analyses performed on the Flatiron population (i.e. lines 534-540 and 541-562) is not entirely clear to me, and the second analysis is not well justified. Why not only present the analysis adjusted for additional potential confounders, if you are concerned about these variables introducing confounding? Why does the analysis switch from multivariable Cox PH to stratified Cox PH?

We thank the reviewer for this insightful comment. We agree that the distinction between the two analyses in the Flatiron population could be made clearer.

To clarify: we conducted two complementary analyses to assess the robustness of our findings. The first used a parsimonious multivariable Cox proportional hazards (PH) model, adjusting for a core set of covariates (n=36748). The second analysis incorporated additional variables of interest using a stratified Cox PH model. These variables: stage, year of diagnosis, age at diagnosis, and cancer type, were included as stratification factors to allow for different baseline hazard functions across strata, as they were suspected to violate the proportional hazards assumption. Schoenfeld residual diagnostics and model fit indices supported this modeling decision.

We opted to present both analyses to transparently demonstrate the robustness of our findings to different modeling strategies. While we recognize that the second model introduces more complexity and due to a loss of subjects in the analyses (n=25640) statistical power was reduced (as reflected in the marginal p-value of 0.057). However, the direction and magnitude of the effect estimates were consistent across both models. This consistency strengthens our confidence in the findings from the primary, parsimonious model, and suggests that any residual confounding from the additional covariates is likely minimal.

We have revised the Methods section to clearly delineate the purpose and structure of each model and to better justify the use of stratification in the second analysis. We

appreciate the reviewer's comment, which helped us clarify this important methodological point.

- In the Flatiron analysis it is stated that the analysis included 36,216 COVID-19 positive patients and 532 COVID-19 negative patients (with similar numbers for the second analysis). I have a couple of comments/concerns about this.

We sincerely thank the reviewer for their careful reading and for bringing this to our attention. We apologize for the typographical error in the reported numbers in the Methods. The correct figures are 36,216 COVID-19 negative patients and 532 COVID-19 positive patients included in the Flatiron analysis (this was correctly indicated in the Extended Data Figure).

To clarify further: all 532 patients classified as COVID-19 positive were COVID-19 negative at the index date and subsequently tested positive during follow-up. This ensures that exposure status was correctly assigned relative to the time to event.

We have corrected the error in the Methods section of the manuscript and revised the text to clearly describe the patient counts and timing of COVID-19 status assignment.

- The follow-up period is never stated, but I assume the last follow-up date is in 2022. These numbers are a little surprising (the high proportion who tested positive) but not outside the realm of possibility if this is a highly tested population. But the small number of individuals in the COVID-19 negative cohort does raise questions about how representative this group is in term of metastasis risk (although see my comment below). It's not 100% clear what this population is though – are they individuals who never tested positive throughout the follow-up period, or does it also include individuals who had metastasis (or were otherwise censored) before testing positive?

We sincerely thank the reviewer for these thoughtful comments. We apologize for the earlier misstatement in the Methods regarding cohort sizes. To clarify: the analysis included 36,216 COVID-19 negative patients and 532 COVID-19 positive patients, with all positive individuals confirmed to be COVID-19 negative at the index date.

The data cutoff date was August 31, 2023, which is now explicitly stated in the Methods section. Follow-up for each patient continued until the occurrence of metastasis, death, loss to follow-up, or the last confirmed clinical activity, whichever came first.

Regarding cohort classification: individuals in the COVID-19 negative group were defined as those with no recorded positive COVID-19 test result during the entire follow-up period. However, we acknowledge a key limitation - the Flatiron dataset includes only structured data from participating in-network oncology clinics, which means COVID-19

testing performed outside these settings may not be captured. As a result, some individuals classified as COVID-19 negative may have experienced undocumented infections. This potential misclassification could bias our estimates toward the null and contribute to an underestimation of the true hazard ratio.

We have now added clarification of the cohort definition to the Methods and have expanded the discussion of this limitation in the Results section, in line with the reviewer's concern.

- As COVID-19 diagnosis status was treated as a time-varying covariate, the numbers given for positive and negative individuals is less relevant than the person-time contributed by test-positive and negative individuals. If I'm understanding correctly, diagnosis was entered into the Cox model as a time-varying covariate, so presumably most of the positive individuals contributed "negative" person-time before their COVID-19 diagnosis. This should also alleviate concerns about the impact of the fully negative cohort, which are mentioned in the results. I think it would make more sense and be more informative to give the test-negative and test-positive person-time.

In response to the reviewer's comment, we have now reported cumulative person-years of follow-up, as well as the median and interquartile range (IQR) for follow-up time, stratified by COVID-19 status. This information has been added to the Methods section to provide greater transparency regarding the follow-up duration for each group.

Table 2. follow-up time in year by covid status

Covid	Cumulative person-year	Median (year)	25% Quantile	75% Quantile	IQR
No	277115	4.345	1.915	8.129	6.214
Yes	673	0.984	0.504	1.586	1.082

- How does the risk of metastasis evolve over time in a breast cancer patient? If it increases over time since diagnosis/remission, there may be some confounding by time here, as individuals can only go from negative to positive COVID-19 diagnosis, so COVID-19 positive person-time will be later in time (on average) than negative person-time. Stratifying by year of diagnosis, as they have done, may help alleviate this bias.

We agree that the risk of metastasis may vary over time since diagnosis or remission and could be influenced by factors such as cancer stage, cancer type, year of diagnosis, and age at diagnosis. Furthermore, as individuals can only transition from

COVID-19 negative to positive, COVID-19 positive person-time tends to occur later in follow-up, potentially introducing time-related bias.

To account for this, we included stage, year of diagnosis, age at diagnosis, and cancer type as stratification factors in our adjusted model. We agree that stratifying by year of diagnosis, in particular, helps to control for temporal trends in treatment, metastasis risk, and COVID-19 exposure. We have updated the Methods section to clarify this rationale and appreciate the reviewer's thoughtful suggestion, which helped us strengthen the methodological transparency of our study.

Referee comments are in black, our responses are in blue, and revised text included in the manuscript is in red.

Referee #1 (Remarks to the Author):

If the authors feel that it is difficult to propose and verify treatments in a short period of time, the reviewers feel that they should at least propose some potentially effective methods in the discussion.

We agree with the reviewer's suggestion, and have included a mostly new paragraph in the Discussion:

Line 481 - Our studies highlight the importance of developing interventions to minimize the risk of lung DCC awakening and metastatic disease in the millions of cancer survivors who experience respiratory virus infections. In addition to primary prevention strategies, FDA-approved treatments for managing severe COVID-19 include antagonistic antibodies against IL-6R⁴⁷ and orally-available JAK1/2 inhibitors⁴⁸, raising the prospect of interventions that could reduce the risk of infection-induced metastatic cancer progression. The effectiveness and safety of these interventions, and the timing of their application to avoid impeding the resolution of the infection, will need to first be tested in pre-clinical and clinical studies.

Referee #3 (Remarks to the Author):

The authors addressed all my minor comments and discussed why it is not possible to perform the experiments related to the major points. Thus, the weaknesses mentioned previously were not addressed.

For example:

- The data in Figure R1 show that IL-6 is essential for the development of the virus-induced lung metastases in this model; which addresses a question that is different from the one I asked. It still does not establish its implication in the exit from dormancy.
- The authors modified the text to highlight the limitations of the mamosphere assay. But this has not addressed the implication in the switch from a dormant to a proliferative state.

We had provided the new data in Figure R1 as further demonstration of the requirement for IL-6 in the development of virus-induced lung metastases, by showing this result in an entirely different mouse model (transplanted E0771 cells). As referenced, this is a model of breast cancer dormancy in the lungs, but the reviewer is correct that we did not directly assess "awakening" in this model. However, we do directly demonstrate the requirement for IL-6 in dormant DCC awakening using the MMTV-Her2 mouse model. Previous studies from the Aguirre-Ghiso Lab have provided a detailed mapping in space and time of the switch mechanisms between the dormant mesenchymal state and

epithelial proliferative state (PMID: 36050483; reference 16 in the manuscript). While acknowledging the limitations of the mammosphere assays, the markers used *in vivo*, HER2/vimentin/Ki67 vs. HER2/EpCAM/Ki67 to map dormant vs. proliferative state, respectively, clearly showed that these markers track with the dormant to proliferative state induced by the infection (as early as 3 days post-infection) (see Fig. 2a-d). Thus, we used proper markers (requested in the first round of reviews) to track the dormant vs. proliferative state *in vivo*. Importantly, IL-6 knockout almost completely prevents these phenotypic conversions (increases in EpCAM, decreases in vimentin, and increases in Ki67), providing a clear demonstration that DCC awakening was abrogated in the absence of IL-6 (see Fig. 3a-f). Thus, these marker-based data strongly implicate the switch from a dormant to a proliferative state dependent on IL6 signaling. We have revised this section of the Results to clearly indicate how these results support the conclusion that IL-6 is required for the infection-induced awakening of dormant DCC. As now written (new text is italicized):

Line 217 - Strikingly, the number of Her2⁺ cells in lungs of IAV-infected IL-6 KO:MMTV-Her2 mice was drastically decreased compared with infected MMTV-Her2 mice at both 9 and 28 dpi (**Fig. 3a-c**), with substantial reductions in Ki67⁺Her2⁺ cells, *indicative of maintained DCC dormancy (Fig. 3d)*.

Line 224 - Staining for vimentin and EpCAM demonstrated that most Her2⁺ cells in lungs of IL-6 KO:MMTV-Her2 mice retain vimentin expression and maintain EpCAM^{neg} status, *which together with the failure to enter the cell cycle as shown through IF for Ki67, supports an IL-6 requirement for infection-induced DCC conversion from dormancy to awakening (Fig. 3e,f)*.

Referee #5 (Remarks to the Author):

The authors have responded to all my comments in a satisfactory way. One typographical issue - line 1429-1430 should be deleted as the authors have replaced this erroneous information with the correct information on the number of COVID-19 positive and negative patients.

Thanks so much for catching this. We have deleted the erroneous information.